# Multimodal decoding of human liver regeneration

K. P. Matchett[1], J. R. Wilson-Kanamori[1,26], J. R. Portman[1,26], C. A. Kapourani[1,2,3,26], F. Fercoq[4], S. May[4], E. Zajdel[1], M. Beltran[1], E. F. Sutherland[1], J. B. G. Mackey[4], M. Brice[1], G. C. Wilson[1], S. J. Wallace[1], L. Kitto[1], N. T. Younger[1], R. Dobie[1], D. J. Mole[1,5], G. C. Oniscu[6,7], S. J. Wigmore[1,5], P. Ramachandran[1], C. A. Vallejos[2,8], N. O. Carragher[9], M. M. Saeidinejad[10], A. Quaglia[11,12], R. Jalan[10,13], K. J. Simpson[14], T. J. Kendall[1], J. A. Rule[15], W. M. Lee[15], M. Hoare[16,17], C. J. Weston[18,19], J. C. Marioni[20,21,22], S. A. Teichmann[21,22,23], T. G. Bird[1,4], L. M. Carlin[4,24] & N. C. Henderson[1,25] ✉

The liver has a unique ability to regenerate[1,2]; however, in the setting of acute liver failure (ALF), this regenerative capacity is often overwhelmed, leaving emergency liver transplantation as the only curative option[3–5]. Here, to advance understanding of human liver regeneration, we use paired single-nucleus RNA sequencing combined with spatial profiling of healthy and ALF explant human livers to generate a single-cell, pan-lineage atlas of human liver regeneration. We uncover a novel ANXA2+ migratory hepatocyte subpopulation, which emerges during human liver regeneration, and a corollary subpopulation in a mouse model of acetaminophen (APAP)-induced liver regeneration. Interrogation of necrotic wound closure and hepatocyte proliferation across multiple timepoints following APAP-induced liver injury in mice demonstrates that wound closure precedes hepatocyte proliferation. Four-dimensional intravital imaging of APAP-induced mouse liver injury identifies motile hepatocytes at the edge of the necrotic area, enabling collective migration of the hepatocyte sheet to effect wound closure. Depletion of hepatocyte ANXA2 reduces hepatocyte growth factor-induced human and mouse hepatocyte migration in vitro, and abrogates necrotic wound closure following APAP-induced mouse liver injury. Together, our work dissects unanticipated aspects of liver regeneration, demonstrating an uncoupling of wound closure and hepatocyte proliferation and uncovering a novel migratory hepatocyte subpopulation that mediates wound closure following liver injury. Therapies designed to promote rapid reconstitution of normal hepatic microarchitecture and reparation of the gut–liver barrier may advance new areas of therapeutic discovery in regenerative medicine.

ALF is a syndrome of severe liver injury in the absence of chronic liver disease[3–5]. ALF is often unexpected, affecting previously healthy individuals, and has a rapid onset with a frequently fatal outcome (30% mortality)[3]. The major causes of ALF in the UK and the USA are APAP (paracetamol) toxicity, non-A–E hepatitis, ischaemia, drug-induced liver injury, hepatitis B virus and autoimmunity, with APAP toxicity representing the most common cause in the UK (65.4% of cases) and the USA (45.7%)[3]. By contrast, viral hepatitis A, B and E are the main causes of ALF in Asia[6]. In severe cases of ALF, emergency liver transplantation remains the only curative option. Therefore, effective pro-regenerative therapies, designed to harness and augment the inherent regenerative and reparative capacity of the liver, are urgently required.

Recent single-cell human liver studies have largely focused on chronic liver disease rather than ALF, with emphasis on non-parenchymal cell populations[7–9]. Hepatocytes, the major parenchymal (epithelial) component of the liver accounting for approximately 80% of its mass,

[1]Centre for Inflammation Research, Institute for Regeneration and Repair, University of Edinburgh, Edinburgh, UK. [2]MRC Institute of Genetics and Cancer, University of Edinburgh, Edinburgh, UK. [3]School of Informatics, University of Edinburgh, Edinburgh, UK. [4]Cancer Research UK Beatson Institute, Glasgow, UK. [5]University Department of Clinical Surgery, University of Edinburgh, Edinburgh, UK. [6]Edinburgh Transplant Centre, Royal Infirmary of Edinburgh, Edinburgh, UK. [7]Division of Transplant Surgery, CLINTEC, Karolinska Institutet, Stockholm, Sweden. [8]The Alan Turing Institute, London, UK. [9]Cancer Research UK Edinburgh Centre, Institute of Genetics and Cancer, University of Edinburgh, Edinburgh, UK. [10]Institute for Liver and Digestive Health, University College London, London, UK. [11]Department of Cellular Pathology, Royal Free London NHS Foundation Trust, London, UK. [12]UCL Cancer Institute, University College London, London, UK. [13]European Foundation for the Study of Chronic Liver Failure, Barcelona, Spain. [14]Department of Hepatology, University of Edinburgh and Scottish Liver Transplant Unit, Royal Infirmary of Edinburgh, Edinburgh, UK. [15]Department of Internal Medicine, University of Texas, Southwestern Medical Center, Dallas, TX, USA. [16]Early Cancer Institute, University of Cambridge, Cambridge, UK. [17]Department of Medicine, University of Cambridge, Cambridge, UK. [18]NIHR Birmingham Biomedical Research Centre, University Hospitals Birmingham NHS Foundation Trust and University of Birmingham, Birmingham, UK. [19]Institute of Immunology and Immunotherapy, University of Birmingham, Birmingham, UK. [20]Cancer Research UK Cambridge Institute, University of Cambridge, Cambridge, UK. [21]European Molecular Biology Laboratory, European Bioinformatics Institute, Cambridge, UK. [22]Wellcome Genome Campus, Wellcome Sanger Institute, Cambridge, UK. [23]Department of Physics, Cavendish Laboratory, Cambridge, UK. [24]School of Cancer Sciences, University of Glasgow, Glasgow, UK. [25]MRC Human Genetics Unit, Institute of Genetics and Cancer, University of Edinburgh, Edinburgh, UK. [26]These authors contributed equally: J. R. Wilson-Kanamori, J. R. Portman, C. A. Kapourani. ✉e-mail: Neil.Henderson@ed.ac.uk

perform a vast array of vital metabolic and synthetic functions and are therefore fundamental in the maintenance of normal liver function[10,11]. Hepatocyte replenishment can occur via conversion of cholangiocytes during severe liver injury[12,13]; however, recent studies have demonstrated that hepatocytes are primarily maintained by the proliferation of pre-existing hepatocytes during liver homeostasis and regeneration[14–19]. Despite major advances in our understanding of the modes of hepatocyte replenishment during liver regeneration, a key question remains regarding how the liver restores normal microarchitecture, and hence gut–liver barrier function, following necro-inflammatory liver injury.

Here, using a cross-species, integrative multimodal approach, we investigated the cellular and molecular mechanisms regulating liver regeneration. Our data define: (1) a single-cell, pan-lineage atlas of human liver regeneration; (2) a novel ANXA2[+] migratory hepatocyte subpopulation that emerges during human liver regeneration; (3) a corollary migratory hepatocyte subpopulation in APAP-induced mouse liver injury; (4) that wound closure precedes hepatocyte proliferation during APAP-induced mouse liver injury; (5) motile hepatocytes (assessed using 4D intravital imaging) at the edge of the necrotic area of APAP-induced mouse liver injury enable collective migration of the hepatocyte sheet to effect wound closure; and (6) that depletion of hepatocyte ANXA2 expression reduces hepatocyte growth factor (HGF)-induced human and mouse hepatocyte migration in vitro, and abrogates necrotic wound closure following APAP-induced mouse liver injury.

Our work dissects unanticipated aspects of liver regeneration, demonstrating an uncoupling of wound closure and hepatocyte proliferation and uncovering a novel migratory hepatocyte subpopulation that mediates wound closure following liver injury.

## Deconstructing human liver regeneration

Initially, we screened human liver explant samples from patients transplanted for multiple aetiologies of ALF (APAP-induced and non-A–E hepatitis) and chronic liver disease including metabolic-associated steatotic liver disease; non-alcoholic fatty liver disease; alcohol-induced, primary biliary cholangitis; primary sclerosing cholangitis; and alcoholic liver disease to investigate which human liver diseases exhibit a substantial hepatocyte proliferative response. Non-lesional liver resected from patients undergoing surgical liver resection for solitary colorectal metastasis without exposure to chemotherapy was used as healthy control tissue. APAP-induced ALF (APAP-ALF) and non-A–E hepatitis ALF (NAE-ALF) demonstrated markedly increased hepatocyte proliferation compared with healthy human liver and all chronic human liver diseases examined (Fig. 1a); this increase in hepatocyte proliferation showed no correlation with patient age (Extended Data Fig. 1a). We therefore focused on APAP-ALF and NAE-ALF in this study of human liver regeneration.

Structurally, the healthy liver is divided into lobules, with each lobule consisting of a central vein surrounded by portal tracts (each containing a portal vein, hepatic artery and bile duct; Fig. 1b, top). NAE-ALF, a disease of unknown cause, results in massive necrosis of hepatocytes across the lobule. By contrast, severe APAP poisoning can result in confluent necrosis of hepatocytes (Extended Data Fig. 1b), extending out from the peri-central vein region of the liver lobule (Fig. 1b, bottom). Hepatocyte proliferation in human APAP-ALF was increased in the peri-necrotic region (PNR) compared with the residual viable region (RVR; Extended Data Fig. 1c).

To deconstruct human liver regeneration, we applied a multimodal approach (Fig. 1c) including single-nucleus RNA sequencing (snRNA-seq), spatial transcriptomics and multiplex single-molecule fluorescence in situ hybridization (multiplex smFISH). As transplantation for ALF is much less frequent than transplantation for chronic liver disease, we sourced biobanked, frozen liver explant tissue samples from

multiple liver transplant centres in the UK (Edinburgh, Birmingham and Cambridge) and the USA (the Acute Liver Failure Study Group).

To investigate potential disruption of zonation in human liver regeneration, we spatially profiled liver samples from healthy participants and patients with APAP-ALF (Supplementary Table 1). As expected, spatial transcriptomics recapitulated hepatocyte and myofibroblast topography in APAP-ALF versus controls (Extended Data Fig. 1d). Spatial transcriptomics also showed increased cell cycling in APAP-ALF versus controls (Extended Data Fig. 1d), consistent with the previously observed topography of hepatocyte proliferation in APAP-ALF using immunofluorescence staining (Fig. 1a and Extended Data Fig. 1c). Guided by immunofluorescence staining (Fig. 1d) and multiplex smFISH (Fig. 1e) of established hepatocyte zonation markers, we drew spatial trajectories in spatial transcriptomics using the SPATA framework to identify gene modules differentially expressed across the lobule (Fig. 1f and Supplementary Table 2). Gene Ontology (GO) trajectory analysis in SPATA confirmed known peri-central and peri-portal hepatocyte biological processes (Extended Data Fig. 1e and Supplementary Table 4). Applying these zonation gene modules to APAP-ALF samples demonstrated loss of hepatocyte portal–central polarity and emergence of hepatocytes with both portal and central characteristics (Fig. 1g); this disruption in zonation was confirmed using multiplex smFISH (Fig. 1h and Extended Data Fig. 1f,g) and immunofluorescence staining (Fig. 1i). Further spatial trajectory analysis revealed mixed portal-associated and central-associated GO terms in the residual viable region of APAP-ALF samples, distinct from those present in PNRs and necrotic regions (Fig. 1j and Supplementary Table 4). These data demonstrate that remnant human hepatocytes display functional plasticity to compensate for substantial loss of peri-central hepatocytes following APAP-ALF.

To further decode human liver regeneration, we performed snRNA-seq on healthy ($n = 9$), APAP-ALF explant ($n = 10$) and NAE-ALF explant ($n = 12$) human liver tissue, including paired snRNA-seq and spatial transcriptomics datasets from select patients (Supplementary Table 1). This combined snRNA-seq dataset (72,262 nuclei) was annotated using signatures of known lineage markers (Fig. 1k, Extended Data Fig. 1h–n and Supplementary Tables 2–4). Isolating and clustering the healthy hepatocytes from this dataset, and applying SPATA-derived zonation signatures, we found consensus between the spatial transcriptomics and snRNA-seq approaches (Extended Data Fig. 2e–g).

We provide an open-access, interactive browser to allow assessment and visualization of gene expression in multiple hepatic cell lineages in our healthy, APAP-ALF and NAE-ALF spatial transcriptomics and snRNA-seq datasets (https://liverregenerationatlas.hendersonlab. mvm.ed.ac.uk).

## Migratory hepatocytes in regeneration

Hepatocyte replenishment is a key process during liver regeneration. Given this, we focused on hepatocytes from the annotated human snRNA-seq dataset (Fig. 2a). In line with multiplex smFISH, spatial transcriptomics and immunofluorescence-based quantitation (Fig. 1a,g–j), we observed disruption of zonation and a robust proliferative transcriptomic response in hepatocytes following APAP-ALF and NAE-ALF compared with controls (Fig. 2b and Extended Data Fig. 3a). Clustering these human hepatocytes uncovered a distinct ANXA2[+] subpopulation in APAP-ALF and NAE-ALF compared with controls (Fig. 2a,b, Extended Data Fig. 3b,c and Supplementary Table 3). *ANXA2* was not expressed uniquely in this hepatocyte subpopulation compared with other cell lineages (Extended Data Fig. 3d,e), nor was the emergence of this subpopulation influenced by patient age and sex (Extended Data Fig. 4a–c).

GO analysis of the ANXA2[+] hepatocytes highlighted ameboidal-type cell migration, regulation of cell morphogenesis, epithelial cell migration and regulation of cell shape, suggesting a migratory cell phenotype (Fig. 2c and Supplementary Table 4). Differentially expressed genes in the ANXA2[+] hepatocyte subpopulation defined a migratory

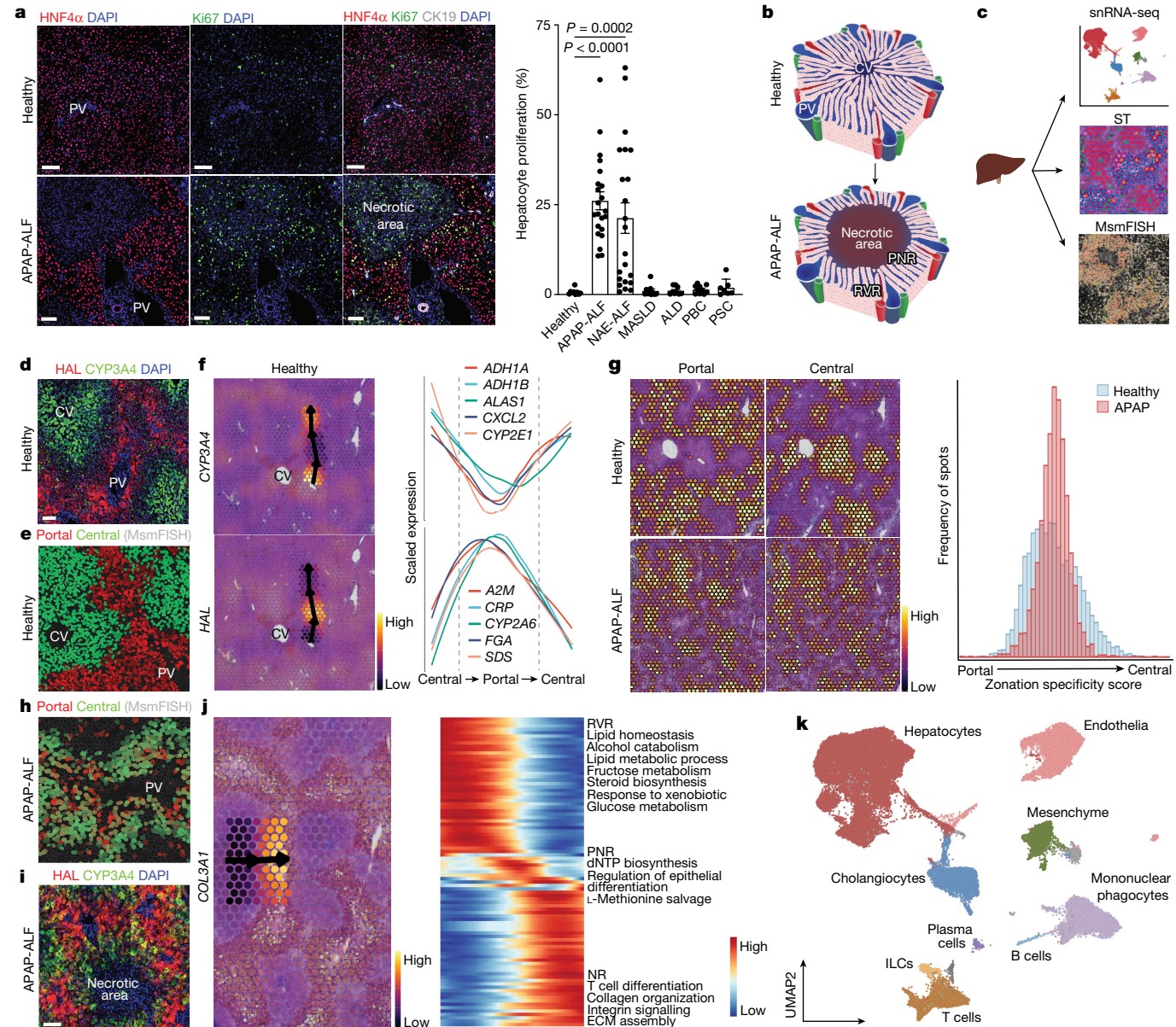

**Fig. 1 | Deconstructing human liver regeneration. a**, Representative immunofluorescence images of HNF4α (hepatocytes, red), Ki67 (green), CK19 (cholangiocytes, white) and DAPI (nuclear stain, blue) in human healthy and APAP-ALF liver tissue (left). Scale bars, 100 μm. Hepatocyte proliferation in human healthy and diseased explant livers across multiple aetiologies (healthy $n = 9$, APAP-ALF $n = 22$, NAE-ALF $n = 22$, metabolic dysfunction-associated steatotic liver disease (MASLD) $n = 10$, alcohol-induced liver disease (ALD) $n = 9$, primary biliary cholangitis (PBC) $n = 10$ and primary sclerosing cholangitis (PSC) $n = 7$ (right). One-way ANOVA, $F = 11.46$, d.f. = 6.82. Data are mean ± s.e.m. PV, portal vein. **b**, Schematic of the healthy liver lobule (top). APAP poisoning, left untreated, can result in massive, confluent necrosis of hepatocytes in the peri-central vein region of the liver lobule (bottom). PNR, peri-necrotic region; RVR, remnant viable region. **c**, Schematic of human liver explant tissue processing for snRNA-seq, spatial transcriptomics (ST) and multiplex smFISH (MsmFISH). Part **c** adapted from ref. 7, Springer Nature. **d**, Representative immunofluorescence image of HAL (portal hepatocytes, red), CYP3A4 (central hepatocytes, green) and DAPI (blue) in healthy human liver tissue. $n = 3$. Scale bar, 100 μm. **e**, Spatial expression (MsmFISH) of hepatocyte zonation gene modules (Supplementary Table 2) in healthy human liver tissue. $n = 2$. **f**, Representative spatial trajectory analysis, identifying differentially expressed gene modules across the healthy human liver lobule. **g**, Spatial expression (ST) of healthy human liver-derived zonation gene modules in healthy and APAP-ALF liver tissue (left). Distribution of zonation specificity score in healthy and APAP-ALF liver tissue (right). **h**, Spatial expression (MsmFISH) of known hepatocyte zonation gene modules (Supplementary Table 2) in human APAP-ALF liver tissue. $n = 2$. **i**, Representative immunofluorescence image of HAL (portal hepatocytes, red), CYP3A4 (central hepatocytes, green) and DAPI (blue) in human APAP-ALF liver tissue. $n = 3$. Scale bar, 100 μm. **j**, Representative spatial trajectory analysis (left) and differential GO terms (Supplementary Table 4) across the human APAP-ALF liver lobule (right). ECM, extracellular matrix; NR, necrotic region. **k**, UMAP of cell lineage inferred using signatures of known lineage markers (Supplementary Table 2). ILC, innate lymphoid cell.

hepatocyte signature (Extended Data Fig. 3f and Supplementary Table 2), which, when applied to spatial transcriptomics, was observed in and around the necrotic region in APAP-ALF and NAE-ALF (Extended Data Fig. 3g). Multiplex smFISH enabled delineation of multiple cell lineages (Extended Data Fig. 3h) and demonstrated expression of the migratory hepatocyte signature in a subpopulation of hepatocytes adjacent to the necrotic region in APAP-ALF, which was absent in healthy liver tissue (Fig. 2d, Extended Data Fig. 3i and Supplementary Table 2). Immunofluorescence staining confirmed significantly increased numbers of ANXA2+ hepatocytes in APAP-ALF and NAE-ALF livers compared

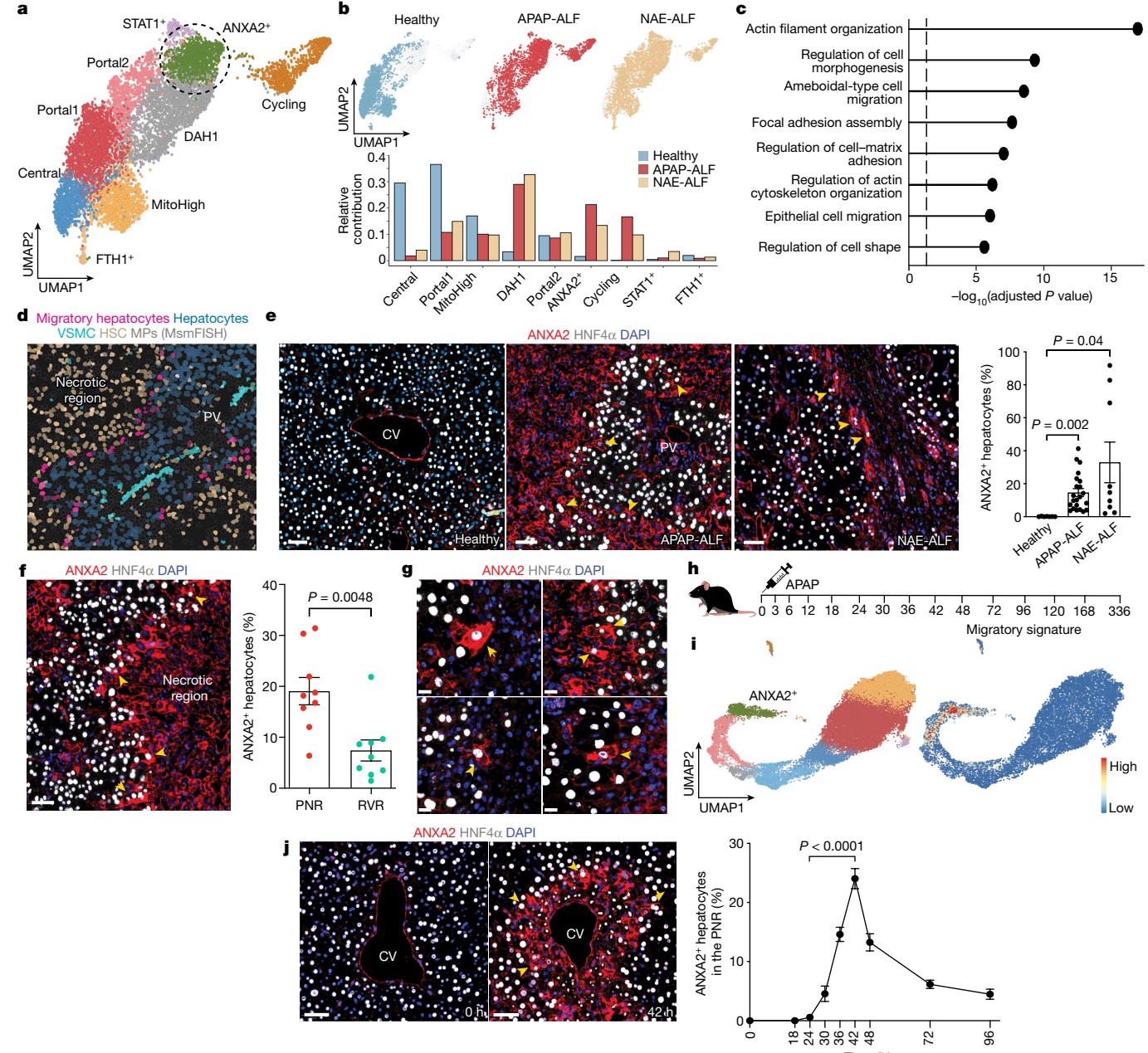

**Fig. 2 | Migratory hepatocytes in regeneration. a**, UMAP of hepatocyte nuclei from healthy, APAP-ALF and NAE-ALF human liver explants, coloured by cluster. DAH1, disease-associated hepatocytes 1. **b**, UMAPs (top) and barplots (bottom) displaying relative contribution of healthy, APAP-ALF and NAE-ALF samples across hepatocyte clusters. **c**, GO terms enriched in human ANXA2+ hepatocytes (Supplementary Table 4). **d**, MsmFISH showing migratory hepatocytes in the APAP-ALF human liver in relation to other cell lineages. HSC, hepatic stellate cell; MP, mononuclear phagocytes; VSMC, vascular smooth muscle cell. **e**, Representative immunofluorescence images of ANXA2 (red), HNF4α (hepatocytes, white) and DAPI (nuclear stain, blue) in healthy, APAP-ALF and NAE-ALF human liver tissue (left; scale bars, 50 μm). Yellow arrowheads denote ANXA2+ hepatocytes. Percentage of ANXA2+ hepatocytes in healthy ($n = 7$), APAP-ALF ($n = 22$) and NAE-ALF ($n = 9$) human livers (right). Two-tailed unpaired Student's $t$-test: APAP-ALF $t = 3.39$, d.f. = 27; NAE-ALF $t = 2.33$, d.f. = 14. Data are mean ± s.e.m. **f**, Representative immunofluorescence images of ANXA2 (red), HNF4α (hepatocytes, white) and DAPI (blue) in the APAP-ALF human liver (left). Yellow arrowheads denote ANXA2+ hepatocytes. Scale bar,

50 μm. Percentage of ANXA2+ hepatocytes present in the PNR and the RVR of the APAP-ALF human liver (right, $n = 9$). Two-tailed paired Student's $t$-test: $t = 3.86$, d.f. = 8. Data are mean ± s.e.m. **g**, Representative immunofluorescence images of ANXA2 (red), HNF4α (hepatocytes, white) and DAPI (blue) in the APAP-ALF human liver ($n = 4$). Yellow arrowheads denote ANXA2+ hepatocytes with migratory morphology. Scale bars, 20 μm. **h**, Schematic of the timepoints processed for snRNA-seq post-APAP-induced liver injury in mice. **i**, UMAP of mouse hepatocyte nuclei from all timepoints post-APAP-induced liver injury, coloured by cluster (left). Application of human migratory hepatocyte gene module to mouse hepatocytes, showing corresponding region enriched in migratory gene signature (right). **j**, Representative immunofluorescence images of ANXA2 (red), HNF4α (hepatocytes, white) and DAPI (blue) in the mouse liver post-APAP-induced liver injury (left). Yellow arrowheads denote ANXA2+ hepatocytes. Scale bars, 50 μm. Percentage of ANXA2+ hepatocytes across timepoints post-APAP-induced liver injury in the PNR (right). Two-way ANOVA, $n = 3$ (0 h and 18 h), $n = 6$ (24–96 h), $F = 44.60$, d.f. = 8,34. Data are mean ± s.e.m.

with controls (Fig. 2e), and increased numbers of ANXA2[+] hepatocytes in the PNR in APAP-ALF (Fig. 2f). ANXA2[+] hepatocytes exhibited a motile morphology with ruffled membranes and extending lamellipodia, characteristic of migratory cells (Fig. 2g and Extended Data Fig. 3j).

We investigated potential ligand interactions with receptors expressed by the migratory hepatocyte subpopulation in the human APAP-ALF snRNA-seq data (Supplementary Table 6), and found that mesenchymal and cholangiocyte subpopulations were the dominant interacting partners (Extended Data Fig. 5a). Multiplex smFISH demonstrated co-location of migratory hepatocytes with myofibroblasts (mesenchyme cluster 1), hepatic stellate cells (mesenchyme cluster 2) and cholangiocytes in the PNR (Extended Data Fig. 5b). Interactome analysis highlighted multiple interactions related to the transforming growth factor-β (TGFβ) signalling pathway (Extended Data Fig. 5c), which has previously been shown to be an important regulator of epithelial cell plasticity and migration[20].

Necrosis, hepatocyte proliferation and hepatocyte ANXA2 expression were also analysed in a cohort of patients with acute severe liver injury who underwent transjugular biopsy and recovered without transplantation. Mean necrotic area was 43.9% (±5.7% s.e.m.), mean hepatocyte proliferation was 21% (±4.2% s.e.m.) and ANXA2[+] hepatocytes were enriched in the PNR (Extended Data Fig. 5d,e). This ANXA2[+] hepatocyte subpopulation was also observed in other causes of human ALF including hepatitis A and hepatitis B, and other drug-induced liver injuries (Extended Data Fig. 5f). Furthermore, we observed ANXA2[+] hepatocytes in multiple aetiologies of human chronic liver disease (Extended Data Fig. 5g) and mouse models of liver injury (Extended Data Fig. 5h,i), demonstrating that ANXA2[+] hepatocytes are present in both peri-central vein and peri-portal liver injury in humans and mice.

To determine whether a corollary migratory hepatocyte subpopulation exists in a mouse model of APAP-induced acute liver injury, we performed snRNA-seq (59,051 nuclei) and spatial transcriptomics on the mouse liver across multiple timepoints (Fig. 2h,i and Extended Data Fig. 6a–d). Applying the human migratory hepatocyte signature to mouse hepatocytes (Fig. 2i and Supplementary Video 1) identified an analogous ANXA2[+] subpopulation (Extended Data Fig. 6e and Supplementary Video 2) emerging in response to APAP-induced mouse liver injury. GO analysis of this analogous ANXA2[+] subpopulation in APAP-induced mouse liver injury demonstrated similar migratory ontology (Extended Data Fig. 6f and Supplementary Table 4). Similar to the human snRNA-seq dataset, *Anxa2* gene expression was also observed in leukocytes, mesenchyme and endothelia during APAP-induced mouse liver injury (Extended Data Fig. 6g–j).

Akin to our findings in human APAP-ALF, spatiotemporal profiling revealed disruption of hepatocyte zonation (Extended Data Fig. 6k) in APAP-induced mouse liver injury. Applying SPATA-derived mouse zonation signatures to the hepatocyte subpopulations demonstrated consensus between the spatial transcriptomics and snRNA-seq datasets (Supplementary Table 2 and Supplementary Video 3). Furthermore, spatial transcriptomics delineated the migratory hepatocyte subpopulation around the necrotic region following APAP-induced mouse liver injury (Extended Data Fig. 6l). Immunofluorescence staining confirmed the presence of ANXA2[+] hepatocytes following APAP-induced mouse liver injury, and these hepatocytes exhibited a similar morphology to the ANXA2[+] hepatocytes observed in human APAP-ALF, with ruffled membranes and lamellipodia (Extended Data Fig. 6m,n). ANXA2[+] hepatocytes increased following APAP-induced acute liver injury in male and female mice (Fig. 2j and Extended Data Fig. 7a) and were specifically enriched in the PNR (Extended Data Fig. 7b,c). ANXA2[+] hepatocytes were less circular than ANXA2[−] hepatocytes in the PNR (Extended Data Fig. 7d). Expression of zonula occludens 1 (ZO-1), a tight junction protein mediating cell–cell contact and cell polarity, was similar in portal-associated and central-associated hepatocytes in uninjured liver (Extended Data Fig. 7e). ANXA2[+] hepatocytes in the PNR expressed ZO-1 following APAP-induced acute liver injury (Extended Data Fig. 7f).

ZO-1 expression in the PNR did not change following APAP-induced acute liver injury, demonstrating maintenance of epithelial sheet connections and hepatocyte polarity during wound closure (Extended Data Fig. 7g). Similar to our findings in human APAP-ALF, interactome analysis of mouse APAP-induced acute liver injury (Supplementary Table 6) highlighted myofibroblasts (mesenchyme clusters 1, 5 and 7) as a potential interacting partner with the migratory hepatocyte subpopulation, via multiple interactions related to the TGFβ signalling pathway (Extended Data Fig. 7h,i).

To determine whether new hepatocytes derive from hepatocytes following APAP-induced mouse liver injury, we lineage-traced hepatocytes using adeno-associated viral AAV8.TBG.Cre injected into *R26R[LSL]tdTomato* mice. AAV8.TBG.Cre injection activated tdTomato expression in 99.8% (±0.1% s.e.m.) of hepatocytes (HNF4α[+]) in healthy mouse liver (Extended Data Fig. 7j). Hepatocyte lineage tracing following APAP-induced mouse liver injury demonstrated that 99.9% (±0.1% s.e.m.; day 7) of new hepatocytes derived from tdTomato[+] hepatocytes (Extended Data Fig. 7j). Furthermore, 100% of ANXA2[+] hepatocytes derived from tdTomato[+] hepatocytes at all timepoints studied (Extended Data Fig. 7k).

As hepatocyte replenishment can occur via conversion of cholangiocytes during severe liver injury[9,10], we performed in silico analysis of the human snRNA-seq dataset and found that hepatocytes and cholangiocytes express distinct genes (Extended Data Fig. 1h and Supplementary Table 3), cluster separately and lack any observable connection in diffusion maps and force-directed graphs (Extended Data Fig. 7l). These data suggest that cholangiocytes are not a major source of hepatocytes in human APAP-ALF and NAE-ALF.

## Wound closure precedes proliferation

Investigating the dynamics of liver regeneration following APAP-induced mouse liver injury, we uncovered a temporal disconnect between wound closure (as assessed by percentage necrotic area) and hepatocyte proliferation. In male mice, peak hepatocyte necrosis occurred at 30 h post-APAP-induced liver injury (22.3 ± 1.3% s.e.m.), with percentage necrotic area decreasing by 30.9% at 42 h (15.4 ± 1.4% s.e.m.) and by 58.3% at 48 h (9.3 ± 1% s.e.m.) (Fig. 3a). Wound closure preceded the onset of hepatocyte proliferation, which peaked at 72 h post-APAP-induced liver injury (Fig. 3a). Following APAP-induced acute liver injury in female mice, hepatocyte necrosis peaked at 36 h (22.2 ± 2.6% s.e.m.), with percentage necrotic area decreasing by 51.8% at 42 h (10.7 ± 1.0% s.e.m.) (Extended Data Fig. 8a).

To assess whether hepatocyte repopulation of the necrotic area immediately adjacent to the central vein is driven by proliferation, we gave 5-bromo-2′-deoxyuridine (BrdU) in drinking water to label all proliferating hepatocytes following APAP-induced mouse liver injury (Fig. 3b). Hepatocyte proliferation had returned to baseline levels by day 14, with complete wound closure (Extended Data Fig. 8b). Glutamine synthetase is expressed exclusively in hepatocytes adjacent to the central vein (Fig. 3c), and in male mice, 75.6% (±9.4% s.e.m.) of glutamine synthetase[+] hepatocytes were BrdU[−] days post-APAP-induced liver injury (Fig. 3c). Similarly, 86.9% (±2.3% s.e.m.) of glutamine synthetase[+] hepatocytes were BrdU[−] in female mice at 14 days post-APAP-induced acute liver injury (Extended Data Fig. 8c). Together, these data demonstrate that the majority of hepatocytes in the area immediately adjacent to the central vein did not arise from hepatocyte proliferation.

Having identified a migratory hepatocyte subpopulation in human and mouse APAP-induced liver injury (Fig. 2 and Extended Data Figs. 3, 5 and 6), and given that hepatocyte proliferation is not the major contributor to wound closure (Fig. 3a,c and Extended Data Fig. 8a,c), we performed 4D intravital microscopy (IVM) to investigate whether hepatocyte migration occurs in vivo following APAP-induced liver injury (Fig. 3d). We used AAV8.TBG.Cre combined with two fluorescent mouse reporter lines to label hepatocytes: *Hep;tdTomato* (single-fluorescent

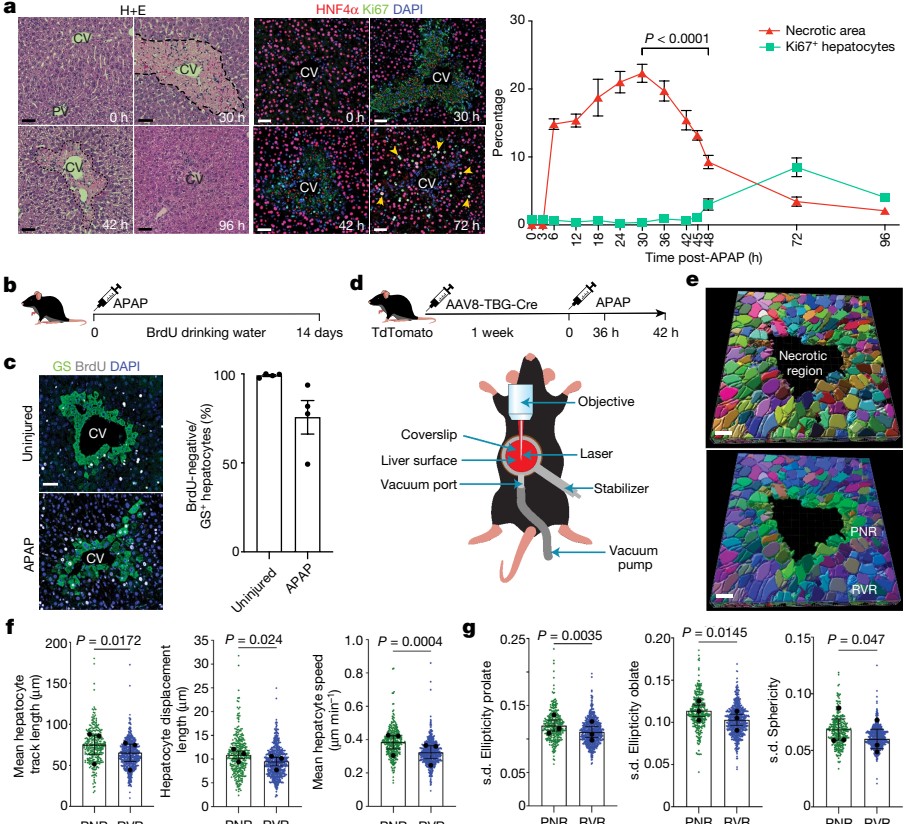

**Fig. 3 | Wound closure precedes proliferation. a**, Representative haematoxylin and eosin (H+E) staining across select timepoints following APAP-induced mouse liver injury (left). Representative immunofluorescence staining: Ki67 (green), HNF4α (hepatocytes, red) and DAPI (nuclear stain, blue) across select timepoints following APAP-induced liver injury (middle). Yellow arrowheads denote Ki67+ hepatocytes. Scale bars, 50 μm. Quantification of the necrotic area (red) and hepatocyte proliferation (Ki67, green) following APAP-induced liver injury (right). Two-way ANOVA, $n = 3$ (0–18 h), $n = 8$ (24–96 h), $F = 40.4$, d.f. = 12.58. Data are mean ± s.e.m. **b**, Schematic depicting BrdU dosing of mice post-APAP-induced liver injury. **c**, Representative immunofluorescence staining (left) and quantification (right) of the percentage of BrdU-negative/glutamine synthetase-positive (GS+) hepatocytes adjacent to

the CV: GS (green), BrdU (white) and DAPI (blue). Scale bar, 50 μm. $n = 4$. Data are mean ± s.e.m. **d**, Schematic depicting IVM experimental protocol (top) and the IVM setup (bottom). **e**, Representative IVM images of Cellpose-segmented hepatocytes (top) and binning of hepatocytes into the PNR (green) or the RVR (blue) (right). Scale bars, 40 μm. **f**, Quantification of cell mobility, mean track length (left; $t = 7.52$, d.f. = 2), mean displacement length (middle; $t = 6.34$, d.f. = 2) and mean speed (right; $t = 47.20$, d.f. = 2) in the PNR (green) and the RVR (blue) of hepatocytes. $n = 3$ mice. **g**, Quantification of cell shape changes over time: ellipticity prolate (left; $t = 16.79$, d.f. = 2), ellipticity oblate (middle; $t = 8.21$, d.f. = 2) and sphericity (right; $t = 4.45$, d.f. = 2). s.d., standard deviation. $n = 3$ mice. In parts **f**,**g**, all data are mean ± s.e.m, and two-tailed paired Student's $t$-test was used for statistical analysis.

reporter mice that express cytoplasmic tdTomato after Cre-mediated recombination) and *Hep;mGFP* (double-fluorescent reporter mice expressing membrane-targeted GFP after Cre-mediated excision). IVM did not affect levels of hepatocyte necrosis, proliferation or hepatocyte ANXA2 expression compared with non-IVM imaged mice at 42 h post-APAP-induced liver injury (Extended Data Fig. 8d–f). 4D IVM of *Hep;tdTomato* reporter mice demonstrated centrilobular hepatocyte necrosis in real time during APAP-induced mouse liver injury (Supplementary Videos 4 and 5).

Owing to the prevalence of ANXA2+ hepatocytes (Fig. 2j and Extended Data Fig. 6l,n) and wound closure activity (Fig. 3a) between 36 h and 42 h post-APAP-induced mouse liver injury, we performed IVM during this timeframe (Fig. 3d). 4D IVM of *Hep;tdTomato* mice demonstrated collective migration of the hepatocyte sheet (Supplementary Videos 6 and 7). Rendering of wound volume (Extended Data Fig. 8g, left) showed 19.4% (±2.8% s.e.m.) reduction in necrosis during the imaging period (Extended Data Fig. 8g, right). Intravital imaging of *Hep;tdTomato* mice identified hepatocytes with a motile morphology, including membrane ruffling and the formation of lamellipodia at the hepatocyte leading edge abutting the wound (Extended Data Fig. 8h and Supplementary Videos 6–14). Using *Hep;mGFP* to clearly segment individual

hepatocytes (Fig. 3e, left), we then classified hepatocytes into the PNR (green) or the RVR (blue) to compare cell mobility and shape between the two regions (Fig. 3e, right). Measures of hepatocyte mobility (mean track length, displacement length and speed) were greater in hepatocytes in the PNR than in hepatocytes in the RVR (Fig. 3f). Assessment of cell shape over time demonstrated that hepatocytes in the PNR displayed greater deviation in ellipticity (prolate and oblate) and sphericity than in hepatocytes in the RVR (Fig. 3g). During the imaging period, there was no change in hepatocyte volume in either the PNR or the RVR (Extended Data Fig. 8i). Together, these data demonstrate that hepatocyte migration is a major mechanism of wound closure following APAP-induced liver injury, and that wound closure is not mediated by hepatocyte hypertrophy.

## Hepatocyte ANXA2 regulates wound closure

ANXA2, which we have identified as a key marker in both human and mouse migratory hepatocytes, has been previously shown to regulate cell migration in other disease settings including carcinogenesis[21–23]. Knockdown of ANXA2 in a human hepatocellular carcinoma cell line (Huh7) decreased wound closure at 72 h in a scratch wound assay

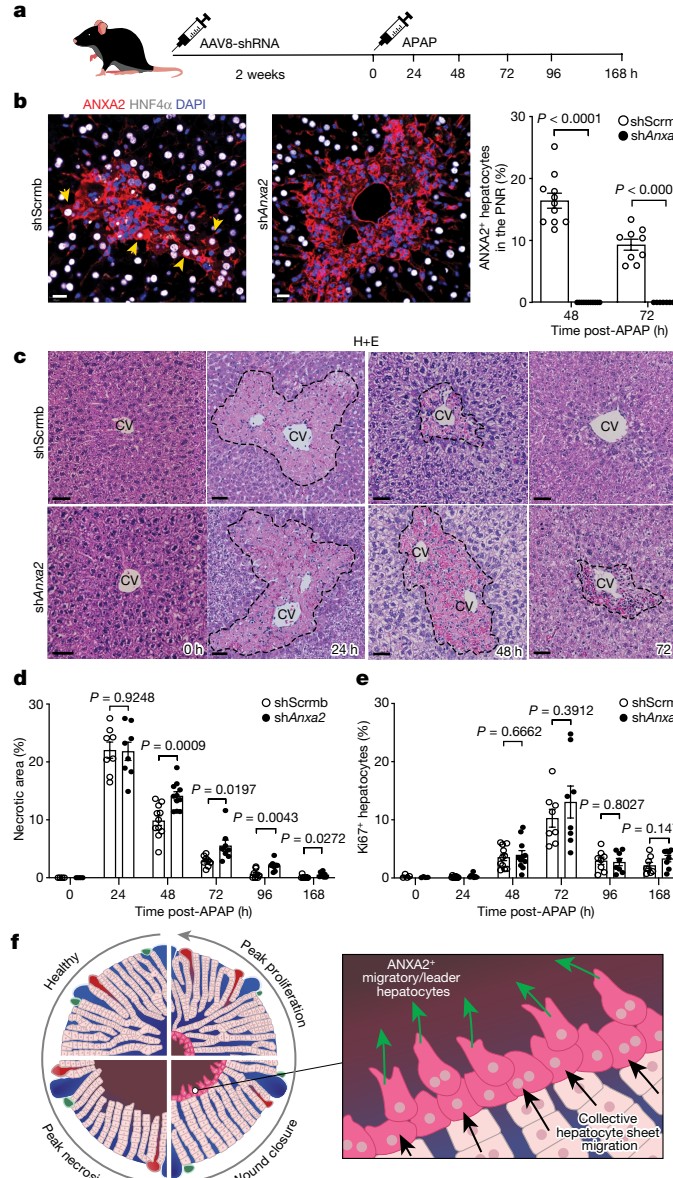

**a**

AAV8-shRNA      APAP

2 weeks      0    24    48    72    96    168 h

**b** ANXA2 HNF4α DAPI

shScrmb

shAnxa2

H+E

**c**

shScrmb

shAnxa2

CV    CV    CV    CV

CV    CV    CV    CV

0 h    24 h    48 h    72 h

**d** ○ shScrmb  ● shAnxa2

$P = 0.9248$
$P = 0.0009$
$P = 0.0197$
$P = 0.0043$
$P = 0.0272$

Necrotic area (%)

Time post-APAP (h)

**e** ○ shScrmb  ● shAnxa2

$P = 0.6662$
$P = 0.3912$
$P = 0.8027$
$P = 0.1478$

Ki67$^+$ hepatocytes (%)

Time post-APAP (h)

**f** Healthy    Peak proliferation

Peak necrosis    Wound closure

ANXA2$^+$ migratory/leader hepatocytes

Collective hepatocyte sheet migration

**Fig. 4 | Hepatocyte ANXA2 regulates wound closure. a**, Schematic depicting the experimental protocol for in vivo hepatocyte *Anxa2* knockdown and APAP-induced mouse liver injury. **b**, Representative immunofluorescence staining of the mouse liver from AAV8-shScrmb-treated or AAV8-sh*Anxa2*-treated mice at 48 h post-APAP-induced liver injury (left). ANXA2 (red), HNF4α (hepatocytes, white) and DAPI (nuclear stain, blue) are shown. Yellow arrowheads denote ANXA2$^+$ hepatocytes. Scale bars, 20 μm. Percentage of ANXA2$^+$ hepatocytes post-APAP-induced liver injury in the PNR of AAV8-shScrmb-treated or AAV8-sh*Anxa2*-treated mice (right). Two-way ANOVA, $n = 11$ (48 h), $n = 9$ (72 h), $F = 20.64$, d.f. = 1. Data are mean ± s.e.m. **c**, Representative H+E staining of mouse livers from AAV8-shScrmb-treated or AAV8-sh*Anxa2*-treated mice across timepoints post-APAP-induced liver injury. Scale bars, 50 μm. **d**, Quantification of the necrotic area following APAP-induced liver injury in AAV8-shScrmb-treated or AAV8-sh*Anxa2*-treated mice. Two-tailed unpaired Student's *t*-test, $n = 4$ (0 h), $n = 8$ (24 h), $n = 11$ (48 h) and $n = 8$ (72–168 h). Data are mean ± s.e.m. **e**, Quantification of Ki67$^+$ hepatocytes following APAP-induced liver injury in AAV8-shScrmb-treated or AAV8-sh*Anxa2*-treated mice. Two-tailed unpaired Student's *t*-test, $n = 4$ (0 h), $n = 8$ (24 h), $n = 11$ (48 h) and $n = 8$ (72–168 h). Data are mean ± s.e.m. **f**, Schematic depicting the temporal disconnect between wound closure and hepatocyte proliferation during APAP-induced liver regeneration.

(Extended Data Fig. 9a,b). Primary hepatocytes from uninjured mouse livers increased *Anxa2* gene expression in response to plating on tissue culture plastic (Extended Data Fig. 9c). Using this in vitro model system, inhibition of *Anxa2* expression in HGF-stimulated primary mouse hepatocytes (Extended Data Fig. 9d) reduced wound closure (Extended Data Fig. 9e); this effect was not mediated by a reduction in hepatocyte proliferation (Extended Data Fig. 9f). *Met* (encoding the HGF receptor) gene expression was similar between Scrmb-siRNA (control) and *Anxa2*-siRNA treated hepatocytes (Extended Data Fig. 9g).

To determine the functional role of hepatocyte ANXA2 during APAP-induced mouse liver injury, we used AAV8-shRNA-*Anxa2* to knockdown *Anxa2* specifically in hepatocytes in vivo (Fig. 4a,b). Knockdown of hepatocyte *Anxa2* abrogated wound closure compared with control (AAV8-shRNA-Scrmb) following APAP-induced mouse liver injury (Fig. 4c,d); this effect was not mediated by a reduction in hepatocyte proliferation or by ongoing hepatocyte injury (Fig. 4e and Extended Data Fig. 9h). Hepatocytes in the PNR of AAV8-shRNA-*Anxa2*-treated mice lacked membrane F-actin and were more circular than those in AAV8-shRNA-Scrmb-treated mice, demonstrating a reduced migratory phenotype (Extended Data Fig. 9i,j). Epithelial sheet connections and hepatocyte polarity were similar between the treatment groups, as determined by ZO-1 expression (Extended Data Fig. 9k). In vitro, phagocytosis of necrotic Scrmb-siRNA and *Anxa2*-siRNA treated hepatocytes by bone marrow-derived macrophages was similar between the two groups (Extended Data Fig. 9l). Treatment with AAV8-shRNA-*Anxa2* did not affect *Anxa2* expression in leukocytes (CD45$^+$), mesenchymal cells and endothelial cells during APAP-induced mouse liver injury compared with control (Extended Data Fig. 10a–c). Furthermore, total numbers of leukocytes, mesenchymal cells and endothelial cells were similar between AAV8-shRNA-Scrmb-treated and AAV8-shRNA-*Anxa2*-treated groups (Extended Data Fig. 10d). In summary, these data demonstrate a temporal disconnect between wound closure and hepatocyte proliferation, and that hepatocyte ANXA2 expression regulates hepatocyte migration and wound closure during APAP-induced liver regeneration (Fig. 4f).

## Discussion

To advance our understanding of human liver regeneration and to help inform design of pro-regenerative therapies, we generated a single-cell, pan-lineage atlas of human liver regeneration. We provide an open-access, interactive browser to allow assessment and visualization of gene expression in all hepatic cell lineages (snRNA-seq and spatial transcriptomics datasets) in healthy and regenerating human and mouse liver (https://liverregenerationatlas.hendersonlab.mvm.ed.ac.uk). Using a cross-species, integrative multimodal approach, we uncovered a novel migratory hepatocyte subpopulation, which is critical in mediating successful wound healing and reconstitution of normal hepatic architecture following liver injury.

Hepatocyte proliferation has previously been considered a major driver of hepatic wound closure following necro-inflammatory liver injury[2,24,25], and recent studies have demonstrated that hepatocytes are primarily maintained by the proliferation of pre-existing hepatocytes during liver homeostasis and regeneration[14–19]. We observed a substantial hepatocyte proliferative response in human APAP-ALF explant livers compared with uninjured human liver, demonstrating that hepatocyte proliferation is robust and relatively unimpeded in human APAP-ALF. However, despite vigorous hepatocyte proliferation, the necrotic wound area in human APAP-ALF explant livers remained notable at time of transplantation. These data suggest that mechanisms and processes other than hepatocyte proliferation are critical to effect successful wound closure following human liver injury.

Furthermore, previous studies in transgenic mice have suggested that hepatocyte proliferation is not a key mechanism regulating wound closure following necro-inflammatory liver injury. Mice with knockout

of the gene encoding plasminogen displayed persistent centrilobular damage and severe impairment of repair following CCl[4]-induced acute liver injury, despite a normal hepatocyte proliferative response compared with control mice[26]. Conditional knockout of the HGF receptor (Met) in mouse hepatocytes followed by CCl[4]-induced acute liver injury impaired centrilobular wound closure and restitution of normal tissue architecture, despite similar levels of hepatocyte proliferation compared with controls[27].

Following our discovery of a novel migratory hepatocyte subpopulation during human and mouse liver regeneration, we used the mouse model of APAP-induced liver injury to investigate the dynamics of liver regeneration. We uncovered a temporal disconnect between centrilobular wound closure and hepatocyte proliferation, demonstrating that wound closure precedes hepatocyte proliferation following APAP-induced liver injury in male and female mice. Furthermore, continual administration of BrdU to label all hepatocytes following APAP-induced mouse liver injury demonstrated that in male and female mice, the majority of hepatocytes in the area immediately adjacent to the central vein did not arise from hepatocyte proliferation. 4D intravital imaging of APAP-induced mouse liver injury identified motile hepatocytes, displaying membrane ruffling and the formation of dynamic protrusions at the leading edge, with collective cell migration of the hepatocyte sheet to effect wound closure. Depletion of hepatocyte ANXA2 expression reduced HGF-induced migration of human and mouse hepatocytes in vitro, and depletion of hepatocyte ANXA2 in vivo abrogated necrotic wound closure following APAP-induced liver injury in mice. In addition, in vivo knockdown of hepatocyte ANXA2 reduced membrane F-actin expression and increased circularity of peri-necrotic hepatocytes compared with control mice. We further observed ANXA2[+] hepatocytes in multiple aetiologies of human chronic liver disease and mouse models of liver injury, demonstrating that ANXA2[+] hepatocytes occur in response to both peri-central and peri-portal liver injury.

Wound healing in the skin is largely driven by keratinocyte migration[28], where rapid reconstitution of the epidermal barrier is vital to stop invasion by pathogens. Our findings point towards a similar mechanism occurring in the liver. Expeditious wound closure in the liver, with restoration of the gut–liver barrier following acute, massive epithelial injury, may be a key determinant of patient outcome in ALF in which sepsis and multi-organ failure are the most common cause of death[3–5]. Recent studies in mice have demonstrated that APAP-induced liver injury causes impairment of intestinal barrier integrity and increased bacterial translocation[29–31], highlighting that APAP toxicity causes barrier breakdown in both the gut and the liver. We propose that the liver prioritizes rapid re-epithelialization via migration of the hepatocyte sheet before hepatocyte proliferation, as rapid restoration of the gut–liver barrier is paramount to survival by preventing bacterial dissemination with subsequent sepsis and multi-organ failure.

In summary, our work dissects unanticipated aspects of human liver regeneration, uncovering a novel migratory hepatocyte subpopulation that mediates wound closure following liver injury. Therapies designed to promote hepatocyte migration with rapid reconstitution of normal hepatic microarchitecture and reparation of the gut–liver barrier may open up a new area of therapeutic discovery in regenerative medicine.

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

## Methods

### Study participants

**University of Edinburgh, UK.** Local approval for procuring human liver tissue for snRNA-seq, spatial transcriptomics and histological analysis was obtained from the Scotland 'A' Research and Ethics Committee (16/SS/0136) and the NRS BioResource and Tissue Governance Unit (study number SR574), following review at the East of Scotland Research Ethics Service (reference 15/ES/0094). Written informed consent was obtained from the participant or a legally authorized representative before enrolment per local regulations. ALF liver tissue was obtained intraoperatively from patients undergoing orthotopic liver transplantation at the Scottish Liver Transplant Unit, Royal Infirmary of Edinburgh. Patient demographics are summarized in Supplementary Table 1 for patients transplanted for APAP-induced ALF and non-A-E ALF. Healthy non-lesional liver tissue was obtained intraoperatively from patients undergoing surgical liver resection for solitary colorectal metastasis at the Hepatobiliary and Pancreatic Unit, Department of Clinical Surgery, Royal Infirmary of Edinburgh. Patients with a known history of chronic liver disease, abnormal liver function tests or those who had received systemic chemotherapy within the past 4 months were excluded from this cohort. For histological assessment of human ALF and chronic liver disease tissue, anonymized unstained formalin-fixed, paraffin-embedded liver tissue sections were provided by the Lothian NRS Human Annotated Bioresource under authority from the East of Scotland Research Ethics Service REC 1 (reference 15/ES/0094).

**US Acute Liver Failure Study Group network.** The Acute Liver Failure Study Group (ALFSG) consortium of US liver centres was established in 1998 to better define causes and outcomes of acute liver injury and ALF. The study protocol was approved by the local institutional review boards of the participating sites: University of Texas Southwestern Medical Center; Baylor University Medical Center, Dallas, TX; Medical University of South Carolina, Charleston, SC; University of Washington, Seattle, WA; Washington University, St. Louis, MO; University of California, San Francisco, and California Pacific Medical Center, San Francisco, CA; University of Nebraska, Omaha, NE; Mount Sinai Medical Center and Columbia University Medical Center, New York, NY; Mayo Clinic, Rochester, MN; University of Pittsburgh, Pittsburgh, PA; Northwestern University, Chicago, IL; Oregon Health Sciences Center, Portland, OR; University of California, Los Angeles, CA; University of Michigan, Ann Arbor, MI; Yale University, New Haven, CT; University of Alabama, Birmingham, AL; Massachusetts General Hospital, Boston, MA; Duke University, Durham, NC; Mayo Clinic, Scottsdale, AZ; Albert Einstein Medical Center and University of Pennsylvania, Philadelphia, PA; Virginia Commonwealth University, Richmond, VA; University of California, Davis, CA; Mayo Clinic, Jacksonville, FL; University of California, San Diego, CA; The Ohio State University, Columbus, OH; University of Kansas Medical Center, Kansas City, KS; Emory University, Atlanta, GA; University of Alberta, Edmonton, Canada. Written informed consent was obtained from the participant or a legally authorized representative before enrolment per local regulations. Sites obtained portions of fresh explanted liver tissue cut into 1 $cm^3$ pieces, placed into individual cryovials and stored at −80 °C until requested for study. The ALFSG was supported by the National Institute of Diabetes and Digestive and Kidney Diseases (NIDDK; grant no. U-01-58369). The samples used in this study were supplied by the NIDDK Central Repositories. This article does not necessarily reflect the opinions or views of the NIDDK Central Repositories or the NIDDK.

**University of Cambridge, UK.** Patients were recruited at Addenbrooke's Hospital, Cambridge, UK with approval from the Health and Social Care Research Ethics Committee A, Office for Research Ethics Committees, Northern Ireland (ORECNI) (16/NI/0196 and 20/NI/0109). Written informed consent was obtained from the participant or a legally authorized representative before enrolment per local regulations. Liver tissue from patients with ALF was derived from explanted livers at the time of transplantation. All tissue samples were snap-frozen in liquid nitrogen and stored at −80 °C in the Human Research Tissue Bank of the Cambridge University Hospitals NHS Foundation Trust.

**University of Birmingham, UK.** Human liver tissue obtained from the University of Birmingham, UK was obtained under approval by South Birmingham Ethics Committee, Birmingham, UK (reference 06/Q2708/11, 06/Q2702/61), and written informed consent was obtained from the participant or a legally authorized representative before enrolment per local regulations. Liver tissue was acquired from explanted livers from patients undergoing orthotopic liver transplantation at the Queen Elizabeth Hospital, Birmingham. All tissue samples were snap-frozen in liquid nitrogen and stored at −80 °C before being processed and shipped by the Birmingham Human Biomaterials Resource Centre (reference 09/H1010/75; 18-319).

**University College, London, UK.** Human liver tissue obtained from University College, London, UK was obtained under local ethical approval (London-Hampstead Research Ethics Committee; reference 07/Q0501/50). Written informed consent was obtained from the participant or a legally authorized representative before enrolment per local regulations. Liver tissue (formalin-fixed, paraffin-embedded) was acquired via transjugular liver biopsy from patients presenting with acute, severe liver injury; these patients spontaneously recovered without liver transplantation.

### Mice

Mice used for all experiments were 8–12 weeks of age and housed in conventional barrier unit facilities with conventional bedding, 12–12-h light–dark cycle, ambient temperature control (21 °C; humidity 40–60%) and access to food and water ad libitum, under pathogen-free conditions at the University of Edinburgh. Male mice were used for all experiments unless described otherwise in the figure legends. Age- and sex-matched mice were randomly assigned to treatment groups. Blinding to the origin of the tissue samples was not performed. The investigators performing the immunofluorescence staining, single-nucleus RNA-seq and RT-qPCR were different from the investigators collecting tissue. The IVM studies were not blinded. All experimental protocols were approved by the University of Edinburgh Animal Welfare and Ethics Board in accordance with UK Home Office regulations. C57BL/6JCrl mice were obtained from Charles River Laboratories (UK). mTmG (Jax 007676; *B6.129(Cg)-Gt(ROSA)26Sor^tm4(ACTB-tdTomato,-EGFP)Luo/J*)[32] and TdTomato (Jax 007914; *B6.Cg-Gt(ROSA)26Sor^tm14(CAG-tdTomato)Hze/J*)[33] reporter mice were obtained from Jackson Laboratories. For the APAP-induced acute liver injury model, mice fasted for 12 h before intraperitoneal injection with 300 mg kg$^{-1}$ (male) or 350 mg kg$^{-1}$ (female) of APAP dissolved in sterile PBS as previously described[34]. Acute and chronic $CCl_4$-induced liver injury was induced as previously described[35]. For bile duct ligation mice were anaesthetized with isoflurane and the common bile duct was surgically ligated. Buprenorphine pain relief was administered postoperatively via subcutaneous injection and animals were maintained at 25 °C for the duration of the study. For DDC (3,5-diethoxycarbonyl-1,4-dihydrocollidine) diet-induced cholestatic liver injury, mice were given 0.1% DDC mixed with normal chow (Special Diet Services) for 21 days.

For hepatocyte-specific AAV8-Cre-mediated reporter gene induction, stock AAV8.TBG.PI.Cre.rBG (AAV8-TBG-Cre; a gift from J. M. Wilson (plasmid #107787, Addgene); stored at −80 °C) was thawed on ice, diluted in sterile PBS to achieve a working titre of $2 \times 10^{12}$ genetic copies per millilitre and was subsequently stored at −20 °C until usage. On the day of injection, the diluted AAV was thawed and each mouse was injected via the tail vein with 100 µl ($2 \times 10^{11}$ genetic

copies per mouse)[36]. Mice were left for 1 week before APAP-induced acute liver injury. For in vivo hepatocyte-specific knockdown of *Anxa2*, mice were intravenously injected with $1 \times 10^{12}$ genetic copies per millilitre AAV8-GFP-U6-mANXA2-shRNA (sh*Anxa2*) or AAV8-GFP-U6-scrmb-shRNA (shScrmb) and left for 2 weeks before APAP-induced acute liver injury. For administration of BrdU in drinking water, BrdU was dissolved in drinking water at a concentration of 0.8 mg ml$^{-1}$.

## Nuclei isolation for snRNA-seq
Human and mouse livers for snRNA-seq were processed as previously described using the TST method[37]. Mouse liver nuclei isolation was performed on $n = 3$ mice per timepoint and nuclei were pooled for snRNA-seq.

## Droplet-based snRNA-seq
Single nuclei were processed through the 10X Genomics Chromium Platform using the Chromium Single Cell 3′ Library and Gel Bead Kit v3 (PN-1000075, 10X Genomics) and the Chromium Single Cell B Chip Kit (PN-1000074, 10X Genomics) as per the manufacturer's protocol, and as previously described[7]. Libraries were sequenced on either an Illumina HiSeq 4000 or NovaSeq 6000.

## Spatial transcriptomics
Unfixed liver tissues were embedded in Tissue-Tek (OCT) and snap-frozen. Samples were then cryosectioned (10 μm) and placed on pre-chilled Visium (10X Genomics) tissue optimization slides or Visium spatial gene expression slides. Spot size was 55 μm, with 100 μm between spot centroids. Tissue sections were processed as per the manufacturer's protocol. On the basis of optimization time course experiments, tissue was permeabilized for 18 min.

## Multiplex smFISH
Unfixed snap-frozen liver tissues were cryosectioned (10 μm) onto Resolve Biosciences slides and sent on dry ice to Resolve Biosciences for processing. Gene probes were designed using Resolve Biosciences proprietary design algorithm. Probe details are provided in Supplementary Table 6. Following sample imaging, spot segmentation, pre-processing, and signal segmentation and decoding, final image analysis was performed in R programming language v3.4.1.

## Liver IVM
Single-colour tdTomato imaging was performed using a Discovery (Coherent) 'multiphoton' laser tuned to 1,050 nm through a ×20 VIS−IR-corrected water immersion objective (NA 1.0) by placing a water drop on top of the coverslip. Dual-colour eGFP−tdTomato imaging was performed using the same setup with the laser tuned to 1,000 nm. Non-descanned GaAsP detectors (GFP NDD filter BP 500−550 nm; tdTomato NDD filter BP 575−610 nm) were used to initially obtain an overview image, after which three peri-central vein fields were selected for timelapse imaging. These three fields were then imaged as *z*-stacks (40−50 μm) every 10 min for 6 h. Following administration of APAP (350 mg kg$^{-1}$) mice were anaesthetized with isoflurane (4% induction, 1−1.5% maintenance) in approximately 95% oxygen (0.8 l min$^{-1}$) produced by an oxygen scavenger (Vettech). The coat above the liver was clipped back, Lacrilube was applied to the eyes, and mice were then placed in a dorsal position on a heated stage on an upright LSM 880 NLO multiphoton microscope (Zeiss). An abdominal incision was made, exposing the surface of the liver; this was then stabilized using a custom coverslip-holding imaging vacuum stabilization armature attached to the stage. The gentlest possible vacuum was applied to the surface in contact with the liver, holding it in place against the coverslip. Mice received subcutaneous fluids every 45 min during imaging. At the end of the imaging session, mice were killed under general anaesthesia by cervical dislocation.

## Liver IVM processing and segmentation
Timelapse image analysis and visualization were performed using Imaris 9.7 (Bitplane). The Imaris 'reference frame' was first used to correct for *xyz* drift. To create 4D rendering of the wound, the hepatocyte channel was inverted and smoothed using a Gaussian filter. The Imaris 'surface' tool was then used to create a surface corresponding to the wound. Object statistics were then exported to analyse the evolution of the volume of the surface over time (expressed as the percentage of initial volume). To create 4D rendering of individual hepatocytes from the mTmG mice, registered images were first imported in Google drive and the online platform ZerocostDL4mic[38] was used to perform Cellpose segmentation[39] on the eGFP (hepatocytes) channel. Cellpose-annotated images were processed in ImageJ using the 'Label to ROI' plugin[40] to create eroded region of interests (ROIs) followed by the 'Mask from ROI' plugin to produce cell masks. To reduce nonspecific segmentation, the tdTomato signal was thresholded and subtracted from the mask channel using the 'channel arithmetics' tool in Imaris. The Imaris 'surface' tool was then used to create surfaces corresponding to the hepatocytes. Cell statistics were then exported to analyse morphodynamic parameters according to their relation to the wound (distance to the wound). Cell behaviour was determined using the track length and speed (indicating cell mobility) and the standard deviation of cell sphericity and ellipticity (indicating changes in cell shape over time).

## Immunohistochemistry and immunocytochemistry staining
Immunohistochemistry and immunofluorescence staining was performed on formalin-fixed, paraffin-embedded liver tissue sections (4 μm). Slides were deparaffinized and immunofluorescently labelled using a Leica Bond RX$_m$ automated robotic staining system. Antigen retrieval was performed using Bond Epitope Retrieval Solution 1 or 2 for 20 min in conjunction with heat application. Sections were then incubated with primary antibodies diluted in 0.1% Triton-X containing PBS. Sections were stained with DAPI (Sigma) and mounted on glass coverslips in ProLong Gold Antifade mounting medium and stored at 4 °C until time of imaging. For in vitro EdU detection, cells were washed in PBS then fixed for 10 min at room temperature with 4% formaldehyde solution in PBS, and cells were then stained according to the manufacturer's protocol. TROMA-III was deposited to the DSHB by Kemler, R. (DSHB Hybridoma Product TROMA-III). A full list of antibodies and conditions is included in Supplementary Table 6.

## Histology image processing and segmentation
Slides were scanned using a Zeiss Axioscan Z1. Images were processed using Zen Blue software (v2.6). For in vitro immunocytochemistry, wells were imaged using an EVOS FL Auto Imaging System. All image analysis was undertaken in QuPath (v0.3.0)[41] with StarDist nuclei detection extension[42].

## Cell culture
Human immortalized hepatocyte cell line (Huh-7; 300156, Cell Lines Service) was cultured using RPMI1640 supplemented with 10% FBS and 2 mM L-glutamine. Huh-7 cells were authenticated using STR profiling and mycoplasma tested by the commercial provider. Primary mouse hepatocytes were isolated and cultured as previously described[43].

## Gene knockdown in hepatocytes
Gene knockdown in Huh7 and primary mouse hepatocytes was performed using siRNA. Cells were plated at 500,000 cells per millilitre (Huh7, Corning Costar; primary mouse hepatocytes, Corning Primaria) followed by serum starvation overnight (in medium without FBS). siRNA duplexes with Lipofectamine RNAiMAX Transfection Reagent were prepared in OptiMEM according to the manufacturer's recommendations and used at a concentration of 50 nM. Cells were exposed to the duplex for 48 h, in antibiotic-free media containing 2% FBS. Cells

were harvested for RNA and RT–qPCR. Gene knockdown efficiency was assessed by RT–qPCR. Cells were treated with control siRNA (1027280, Qiagen), siRNA for *ANXA2* (human, Hs_ANXA2_8, SI02632385, Qiagen) or siRNA for *Anxa2* (mouse, Mm_Anxa2_3, SI00167496, Qiagen).

### Scratch wound assay

The scratch wound assay was performed using the IncuCyte system (Essen Bioscience). Huh7 cells were plated in IncuCyte ImageLock Plates (Essen Bioscience) and treated as above for *ANXA2* gene knockdown. The subconfluent monolayer was then wounded using the IncuCyte Woundmaker. To obtain a confluent monolayer of primary mouse hepatocytes for wound assays, cells were plated as previously described[44], with modifications. In brief, three separate additions of 500,000 cells per millilitre were seeded onto collagen I-coated IncuCyte ImageLock plates at 2-h intervals. Non-adherent hepatocytes were removed between additions with warmed PBS. Cells were then treated as above for gene knockdown before wounding using the IncuCyte Woundmaker. Following wounding, cells were maintained in complete media with the addition of human HGF (100 ng ml$^{-1}$) and 10% FBS for the duration of the assay. EdU (10 µM) was added to the media 24 h before the end of the assay to assess proliferation. For analysis of wound healing, the scratch wound plugin for the IncuCyte Zoom was used. All experiments were performed as quadruple technical replicates; the number of independent experiments is specified in the figure legends.

### In vitro phagocytosis assay

Primary mouse hepatocytes were isolated from uninjured livers as described above and plated into six-well Primaria plates (Corning). Hepatocytes were treated as before for *Anxa2* knockdown and hepatocyte death was induced at 48 h post-knockdown using 10 mM APAP. Scrmb-siRNA (control) or *Anxa2*-siRNA treated dead hepatocytes were then used in an in vitro phagocytosis assay. Bone marrow-derived macrophages (BMDMs) were isolated from mice femurs and differentiated for 7 days in culture with the addition of $10^4$ U ml$^{-1}$ CSF1. Following differentiation, BMDMs were plated into 24-well plates at 250,000 cells per well and cultured overnight. The next day, dead hepatocytes were cultured with CypHer5e NHS Ester (PA15401) for 10 min at room temperature in the dark, and washed three times in PBS before being applied to BMDMs. CypHer5e-stained dead hepatocytes were then cultured with BMDMs at 37 °C for 1.5 h. Non-ingested hepatocytes were then removed by vigorous washing three times with PBS and residual adherent BMDMs were used for subsequent analysis. For flow cytometric analysis, BMDMs were stained with F4/80 (123141, BioLegend; 30 min at 4 °C, 1:100), and a cell viability stain (DAPI; 1:1,000) was performed immediately before acquiring the samples. Data acquisition was performed on a BD LSR Fortessa flow cytometer (Becton Dickinson) using BD FACS Diva software, and data were analysed using FlowJo 10.9.0 software. The gating strategy is outlined in Supplementary Table 6.

### RNA extraction and RT–qPCR

RNA was extracted from primary mouse hepatocytes and Huh-7 cells using the RNeasy Plus Micro Kit, and cDNA synthesis was performed using the QuantiTect Reverse Transcription Kit according to the manufacturer's protocol (Qiagen). Reactions were performed in triplicate in 384-well plate format. RT–qPCR was performed using PowerUp SYBR Green Master Mix. Primers are detailed in Supplementary Table 6. Samples were amplified on an ABI Quantstudio 5 (Applied Biosystems, Thermo Fisher Scientific). The $2^{-\Delta\Delta Ct}$ quantification method, using GAPDH/Gapdh for normalization, was used to estimate the amount of target mRNA in samples, and expression was calculated relative to average mRNA expression levels from control samples.

### Computational analysis

Four computational datasets were analysed: (1) 72,262 human nuclei from healthy ($n = 9$), APAP-ALF ($n = 10$) and NAE-ALF ($n = 12$) livers;

(2) 59,051 mouse nuclei from an APAP-induced acute liver injury timecourse; (3) spatial transcriptomics spots from human liver ($n = 3$ healthy, $n = 2$ APAP-ALF and $n = 2$ NAE-ALF) and mouse liver ($n = 1$ per timepoint); and (4) multiplex smFISH from human liver ($n = 2$ healthy and $n = 2$ APAP-ALF).

### snRNA-seq analysis

We aligned to GRCh38 and mm10 (Ensembl 93) reference genomes (modified to allow intronic and exonic feature alignment), and estimated nuclei-containing partitions and unique molecular identifiers (UMIs), using the CellRanger v3.1.0 Single-Cell Software Suite from 10X Genomics. Further analysis was performed in the R programming language v3.4.1.

To enable reproducible analysis, we developed the SeuratPipe R package v1.0.0 (https://doi.org/10.5281/zenodo.7331092), a pipeline building on existing packages. In brief, we performed analysis as follows: we performed per-dataset quality control in the Seurat[45] R package v4.1.1. We used the Scrublet[46] Python module v0.2.3 to identify potential doublets and the SoupX[47] R package v1.5.2 to automatically calculate and correct for background contamination. Finally, we excluded nuclei that expressed fewer than 1,000 genes, or mitochondrial gene content of more than 5% of the total UMI count.

After merging the individual datasets, we normalized feature counts per nuclei by dividing the total UMI count for that nuclei, then multiplying by a scale factor of 10,000 and natural-log transforming. We corrected for sample bias by obtaining principal component embeddings using the Harmony[48] R package v0.1.0. Furthermore, we downsampled the hepatocyte populations to standardize sample contribution to downstream analysis.

Nucleus clusters were identified using the shared nearest neighbour modularity optimization-based clustering algorithm implemented in Seurat, using Harmony-corrected principal components for the purpose of constructing the shared nearest neighbour graph. We calculated differentially expressed features using a Wilcoxon rank-sum test. To annotate these clusters, we used a curated list of known marker genes per cell lineage in the liver (Supplementary Table 2) to obtain signature scores using the AddModuleScore function in Seurat. Clusters identified as primarily composed of cycling cells were reclustered to split them out into their constituent lineages as above. We then iteratively applied the above workflow for each lineage thus identified, inserting a 'cleansing' step in which we removed clusters displaying an abundance of nuclei previously identified as doublets or overexpressing marker genes of other lineages. We generated a hepatocyte migration gene module (Supplementary Table 2) using the top 25 (by avg_log$_2$FC) differentially expressed features from the human migratory hepatocyte cluster. GO analysis was performed using the clusterProfiler[49] R package v4.8.3. Liver zonation specificity scores were obtained by first scaling central and portal zonation signature scores to a value between 0 and 1, and subsequently setting zonation score = central score/(central score + portal score). We applied quantile thresholding when plotting features of interest.

To determine any confounding effects of age and sex in the APAP-ALF data, Pearson and point-biserial correlation were performed between the variable of interest and the harmony components.

Diffusion maps and force-directed graphs were generated in Scanpy[50] Python module v1.9. Cell cycle effects were regressed before the following dimensionality reductions. Diffusion maps were compiled based on a neighbourhood graph recalculated in Scanpy using harmony components. This neighbourhood was subsequently denoised using the diffusion map coordinates and used as input to partition-based graph abstraction alongside associated sublineage annotations. The partition-based graph abstraction was used in turn to initialize calculation of force-directed graphs.

Interactome analysis was performed on human APAP-ALF and mouse APAP-induced liver injury datasets, using CellChat[51] R package v1.6.1

with default parameters. Annotations from each individual lineage were mapped back to a dataset containing all lineages before performing the analysis.

## Spatial transcriptomics analysis

We aligned to GRCh38 and mm10 (Ensembl 93) reference genomes using the SpaceRanger v1.0.0 Spatial Gene Expression Software Suite from 10X Genomics. Further analysis was performed in the R programming language v3.4.1.

We performed per-dataset quality control in the Seurat R package v4.1.1. We excluded spots expressing fewer than 800 genes or mitochondrial gene content of more than 20% for both human and mouse samples of the total UMI count. We also manually filtered low-quality spots (those isolated from the main tissue section) using the 10X Genomics Loupe browser (v5.0). Similar to snRNA-seq, we computed gene signature scores for hepatocytes (*TTR*, *TF*, *HP*, *CYP2A6*, *CYP2E1*, *CYP3A4* and *HAL*), myofibroblasts (*ACTA2*, *COL1A1*, *COL1A2* and *COL3A1*) and cycling cells (genes listed in Seurat cc.genes.updated.2019). We applied quantile thresholding when plotting features of interest.

Gene expression and tissue topography were used to draw spatial trajectories across healthy and APAP-ALF tissues via the SPATA2 (ref. 52) R package v0.1.0. The trajectory modelling functionality of SPATA2 was used to identify central-associated and portal-associated genes and corresponding modules (Supplementary Table 2) whose expression trajectory followed the underlying spatial model.

## Multiplex smFISH analysis

Nuclei segmentation and expansion were performed using QuPath to demarcate cells. A gene–cell matrix was then obtained, quality control and normalization was applied, and signature scores in tissue were computed using pre-defined cell populations.

## Further statistical analysis

Further statistical analyses were performed using GraphPad Prism. Comparison of changes between two groups was performed using a two-tailed paired Student's *t*-test or unpaired Student's *t*-test. Comparison of changes between groups was performed using a two-way ANOVA with Sidaks multiple comparison test with a single-pooled variance. Pearson's correlation coefficient ($r$) was used to measure the relationship between variables. $P < 0.05$ was considered statistically significant.

## Reporting summary

Further information on research design is available in the Nature Portfolio Reporting Summary linked to this article.

## Data availability

Our snRNA-seq and spatial transcriptomics data are freely available for user-friendly interactive browsing online (https://liverregeneration-atlas.hendersonlab.mvm.ed.ac.uk). All raw and processed sequencing data are deposited in the Gene Expression Omnibus (GEO) under accession number GSE223561. Lists of lineage-specific genes for signature analysis, lists of marker genes from clustering results, lists of GO terms from enrichment analysis and lists of interactome analysis output are available as Supplementary Tables 1–5. Source data are provided with this paper.

## Code availability

All code is available at https://github.com/HendersonLab/LiverRegenerationAtlas.

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

**Acknowledgements** This work was supported by a Wellcome Trust Senior Research Fellowship in Clinical Science (ref. 219542/Z/19/Z) to N.C.H., a Chan Zuckerberg Initiative Seed Network Grant to N.C.H., and a Tenovus Scotland grant (E20-03) to K.P.M., S. J. Wallace and N.C.H. J.R.P. was supported by a Medical Research Council Precision Medicine PhD studentship. C.A.K. was a cross-disciplinary post-doctoral fellow (XDF) supported by funding from the University of Edinburgh and Medical Research Council (MC_UU_00009/2). T.G.B. was funded by the Wellcome Trust (ref. WT107492Z). S.M. was funded by Cancer Research UK core funding to the CRUK Beatson Institute (refs. A17196 and A31287). J.B.G.M. was supported by a CRUK programme grant (A23390). F.F., J.B.G.M. and L.M.C. were supported by CRUK core funding to the Beatson Institute (A31287) and CRUK core funding to L.M.C. (A23983; DRCRPG-Nov22/100007). D.J.M. was supported by a Medical Research Council Senior Clinical Fellowship (MR/P008887/1). P.R. was supported by a Medical Research Council Clinician Scientist Fellowship (MR/N008340/1) and Medical Research Council Senior Clinical Fellowship (MR/W015919/1). M.M.S. was supported by a Peter Samuel Fellowship. We thank the patients who donated liver tissue for this study; J. Davidson, J. Black, C. Ibbotson and A. Baird of the Scottish Liver Transplant Unit and the research nurses of the Wellcome Trust Clinical Research Facility for assistance with consenting patients for this study; the liver transplant coordinators and surgeons of the Scottish Liver Transplant Unit and the surgeons and staff of the Hepatobiliary Surgical Unit, Royal Infirmary of Edinburgh for assistance in procuring human liver samples; the US ALFSG network for assistance in procuring human liver samples; core facilities and services at the Beatson Institute, in particular the Biological Research Unit & the Beatson Advanced Imaging Resource (BAIR); G. Jacquemet for advice on Cellpose segmentation for IVM analysis; R. Insall and L. Machesky for helpful scientific discussions; W. Mungall for technical support; C. Nicol for help with manuscript illustrations; N. Pham for technical assistance with Incucyte; and C. Winchester and R. Li for critical reading of the manuscript. We acknowledge the contribution to this study made by the University of Birmingham's Human Biomaterials Resource Centre, which has been supported through Birmingham Science City–Experimental Medicine Network of Excellence project. This research was funded in whole, or in part, by the Wellcome Trust (Wellcome Trust Senior Research Fellowship in Clinical Science to N.C.H.; ref. 219542/Z/19/Z). This publication is part of the Human Cell Atlas (www.humancellatlas.org/publications).

**Author contributions** K.P.M. performed the experimental design, data generation, and data analysis and interpretation. J.R.W.-K., J.R.P. and C.A.K. performed the computational analyses on the human snRNA-seq data: J.R.P. analysed hepatocytes and J.R.W.-K. analysed non-hepatocyte lineages. J.R.W.-K., J.R.P. and C.A.K. performed the computational analyses on the mouse snRNA-seq data: J.R.P. analysed hepatocytes and J.R.W.-K. analysed non-hepatocyte lineages. C.A.K. performed the computational analyses on the spatial transcriptomics data. C.A.K. and K.P.M. performed the computational analysis on the multiplex smFISH data. C.A.K. developed the SeuratPipe package and generated the interactive online browser. K.P.M., F.F., J.B.G.M., S.M., T.G.B., N.C.H. and L.M.C. performed the experimental design, data generation, data analysis and interpretation for the mouse liver IVM experiments. E.Z., M. Beltran, E.F.S., M. Brice, G.C.W., N.T.Y. and R.D. performed data generation and analysis. D.J.M., G.C.O.,

S. J. Wigmore, S. J. Wallace, L.K., P.R., M.M.S. and R.J. procured human liver tissue. N.O.C. provided support for the cell migration analyses. K.J.S. procured human liver tissue and provided intellectual contribution. T.J.K. and A.Q. provided liver pathology expertise. The ALFSG, J.A.R., W.M.L., M.H. and C.J.W. provided ALF human liver tissue. C.A.V., J.C.M. and S.A.T. provided advice with computational analyses. T.G.B. provided support with hepatocyte lineage tracing, mouse liver IVM experiments and data interpretation. L.M.C. provided expertise with mouse liver IVM experiments and performed data interpretation. K.P.M., J.R.W.-K., J.R.P., C.A.K. and N.C.H. wrote the manuscript. N.C.H. conceived the study, designed the experiments, interpreted data and supervised the study.

**Competing interests** N.C.H. has received research funding from AbbVie, Pfizer, Gilead, Boehringer-Ingelheim and Galecto, and is an advisor or consultant for AstraZeneca, GSK, MSD, Galecto and Pliant Therapeutics. R.J. is an inventor of OPA (licensed to Mallinckrodt Pharma) and founder of Yaqrit Discovery, Hepyx Limited and Cyberliver. All other authors declare no competing interests.

**Additional information**
**Correspondence and requests for materials** should be addressed to N. C. Henderson.

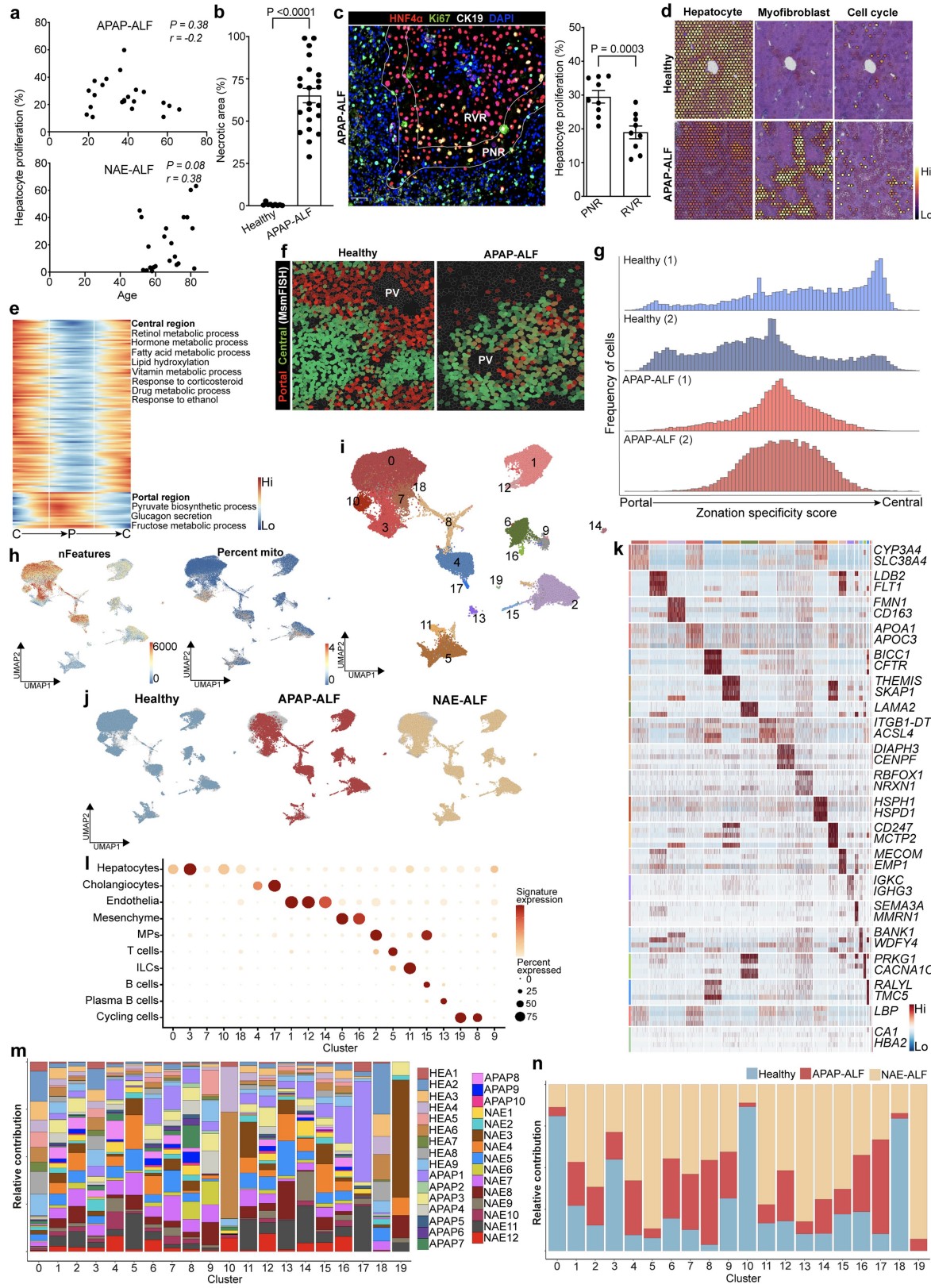

**Extended Data Fig. 1** | See next page for caption.

**Extended Data Fig. 1 | Disruption of zonation and annotation of lineages in human acute liver failure. a**, Pearson's correlation analysis of patient age and hepatocyte proliferation in APAP-ALF (top) and NAE-ALF (bottom). **b**, Quantification of necrotic area in human healthy (n = 9) and APAP-ALF (n = 22) liver tissue. Two-tailed unpaired Student's t-test, t = 8.76, df = 30. Data are mean ± SEM. **c**, Representative immunofluorescence image of HNF4α (hepatocytes, red), Ki67 (green), CK19 (cholangiocytes, white), and DAPI (blue) in human APAP-ALF liver tissue (left). Scale bar 50 µm. Hepatocyte proliferation in the PNR and RVR of human APAP-induced ALF liver tissue (n = 9, right). Two-tailed paired Student's t-test, t = 6.20, df = 8. Data are mean ± SEM. **d**, Spatial expression (ST) of hepatocyte, myofibroblast, and cell cycle signatures across healthy and APAP-ALF liver. n = 3 (healthy), n = 2 (APAP-ALF). **e**, Differential GO terms (Supplementary Table 4) across spatial trajectory analysis from peri-central to peri-portal to peri-central regions in healthy human liver, with exemplar terms labelled (right). **f**, Spatial expression (MsmFISH) of known human zonation gene modules in healthy and APAP-ALF liver tissue. n = 2 (healthy), n = 2 (APAP-ALF). **g**, Distribution of zonation specificity score in healthy (n = 2) and APAP-ALF (n = 2) human liver tissue (MsmFISH). **h**, Quality control metrics (mitochondrial percentage, number of features) across the human snRNA-seq dataset. **i**, UMAP visualisation of 72,262 nuclei from healthy (n = 9), APAP-ALF (n = 10), and NAE-ALF (n = 12) human liver explants. **j**, UMAP visualisation of 72,262 nuclei from healthy (n = 9), APAP-ALF (n = 10), and NAE-ALF (n = 12) human liver explants, annotated by clustering. **k**, Heatmap of marker genes (Supplementary Table 3; colour-coded by cluster) with exemplar genes labelled (right). Columns denote cells, rows denote genes. **l**, Dotplot annotating clusters by lineage signature expression. Circle size indicates cell fraction expressing signature greater than mean; colour indicates mean signature expression. **m**, Stacked barplot denoting relative contribution of human liver samples to each cluster. **n**, Stacked barplot denoting relative contribution of healthy, APAP-ALF, and NAE-ALF samples to each cluster.

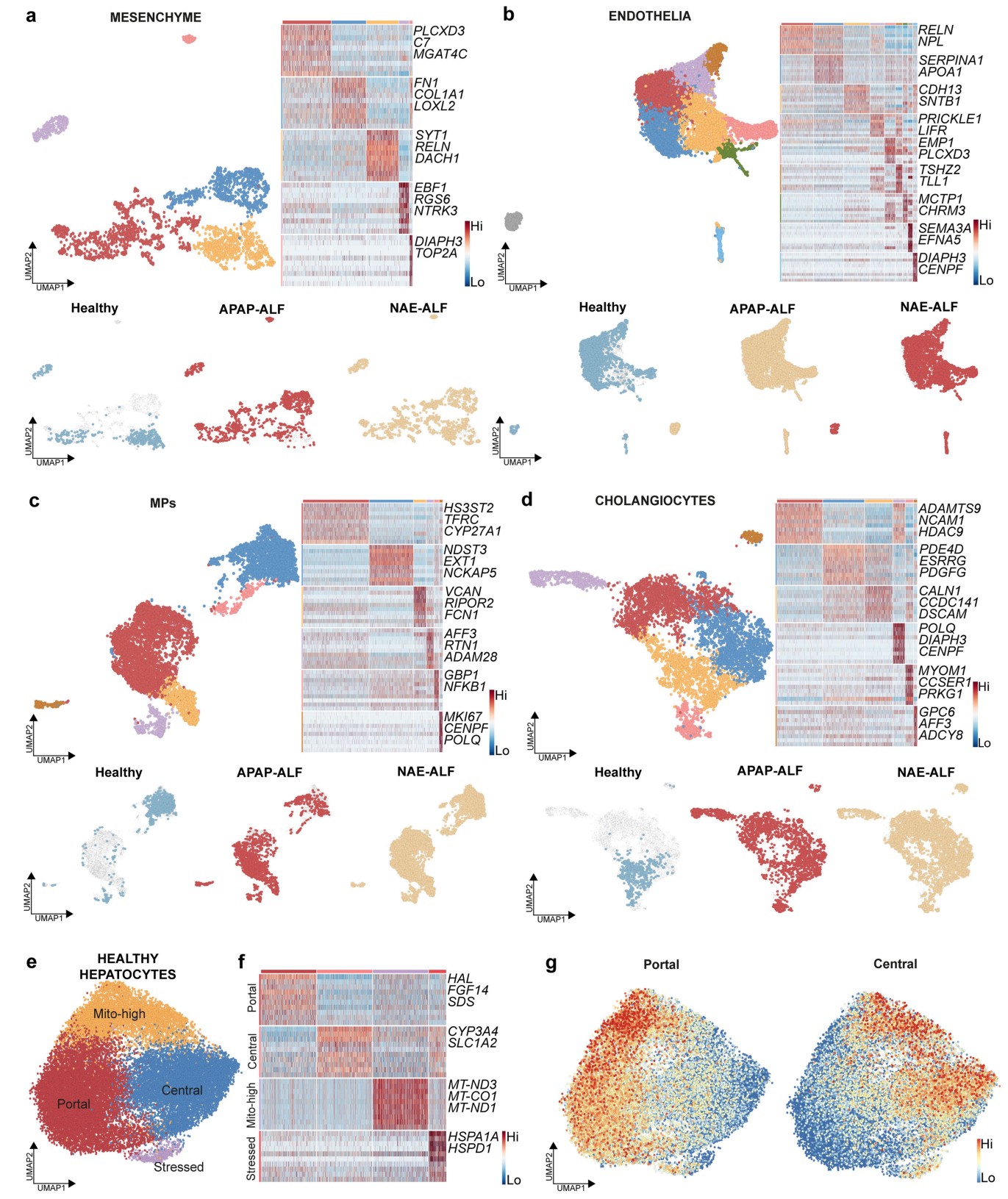

**Extended Data Fig. 2 | Lineage analysis of human liver regeneration atlas.** UMAPs of (**a**) mesenchyme, (**b**) endothelia, (**c**) mononuclear phagocytes, and (**d**) cholangiocytes derived from human liver single nuclei RNA sequencing, coloured by clustering (top left) and aetiology (bottom); heatmap (right) of marker genes (Supplementary Table 3; colour-coded by cluster) with exemplar genes labelled (right). Columns denote cells, rows denote genes. **e**, UMAP of healthy human hepatocyte nuclei, coloured by cluster. **f**, Heatmap of marker genes in healthy hepatocyte clusters (Supplementary Table 3; colour-coded by cluster) with exemplar genes labelled (right). Columns denote cells, rows denote genes. **g**, Human SPATA-derived portal and central region gene modules (Supplementary Table 2) applied to healthy hepatocytes.

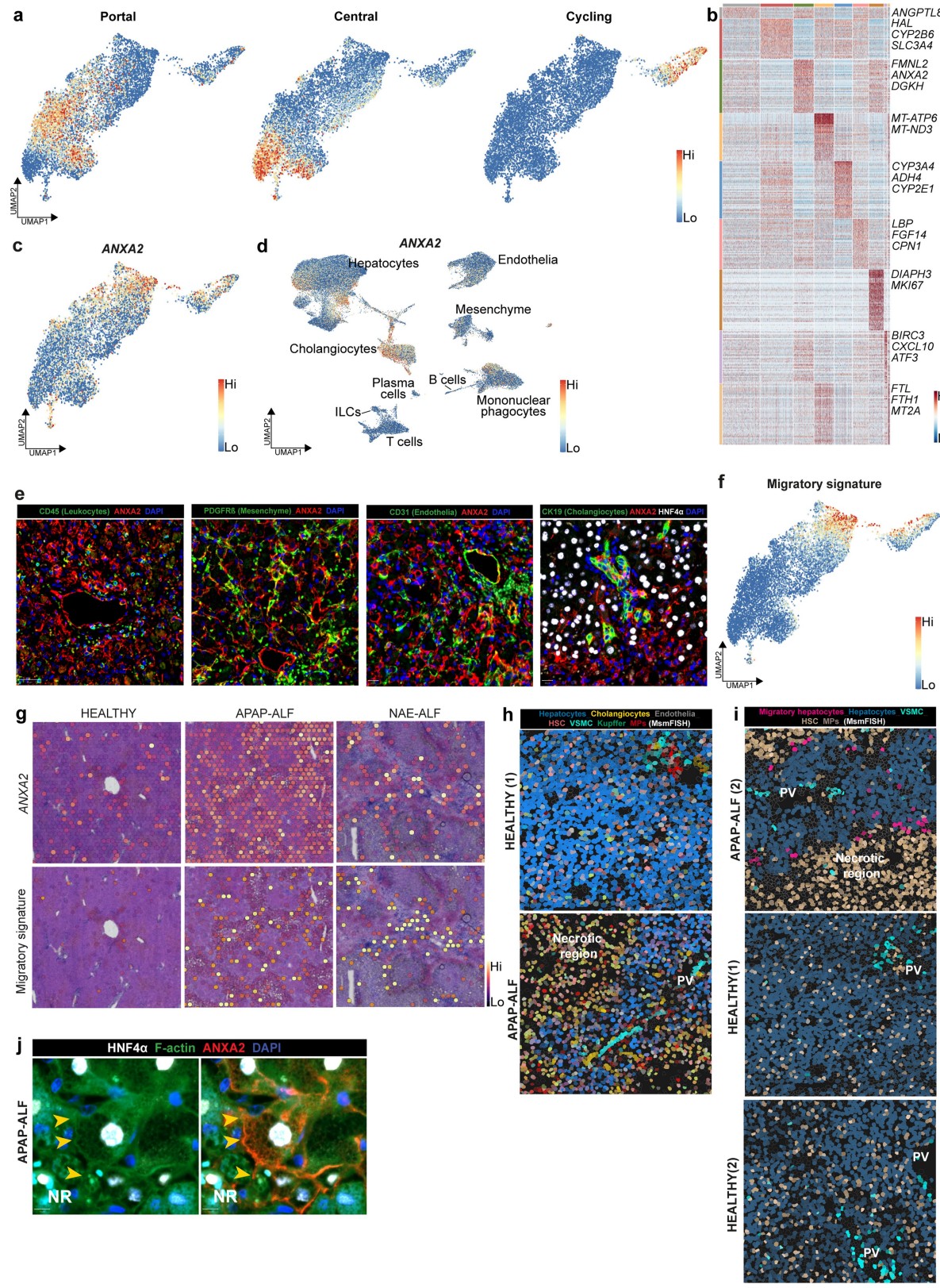

**Extended Data Fig. 3** | See next page for caption.

**Extended Data Fig. 3 | Migratory hepatocytes in human acute liver failure.**
**a**, UMAP of human hepatocyte nuclei, all aetiologies, showing portal, central and cycling gene module (Supplementary Table 2) scores. **b**, Heatmap of marker genes in hepatocyte clusters (Supplementary Table 3; colour-coded by cluster) with exemplar genes labelled (right). Columns denote cells, rows denote genes. **c**, UMAP of human hepatocyte nuclei, all aetiologies, showing *ANXA2* gene expression. **d**, UMAP of human nuclei, all aetiologies, showing *ANXA2* gene expression. **e**, Representative immunofluorescence images of leucocytes (CD45, green; scale bar 50 µm), mesenchyme (PDGFRβ, green), endothelia (CD31, green), or cholangiocytes (CK19, green), along ANXA2 (red), HNF4α (hepatocytes, white), and DAPI (nuclear stain, blue) in human APAP-ALF.

n = 3. Scale bar 20 µm. **f**, UMAP of human hepatocyte nuclei, all aetiologies, showing migratory gene module (Supplementary Table 2) signature. **g**, Spatial expression (MsmFISH) of migratory signature (Supplementary Table 2) and *ANXA2* in healthy, APAP-ALF, and NAE-ALF human liver tissue. **h**, Multiplex smFISH showing cell lineages in healthy (top) and APAP-ALF (bottom) human liver tissue. Gene modules in Supplementary Table 2. **i**, Spatial expression (MsmFISH) of migratory hepatocytes in healthy and APAP-ALF human liver in relation to other cell lineages. Gene modules in Supplementary Table 2. **j**, Representative immunofluorescence images of F-actin (green), ANXA2 (red), HNF4α (hepatocytes, white), and DAPI (blue) in human APAP-ALF liver tissue. NR, necrotic region. n = 3. Scale bar 10 µm.

**a**

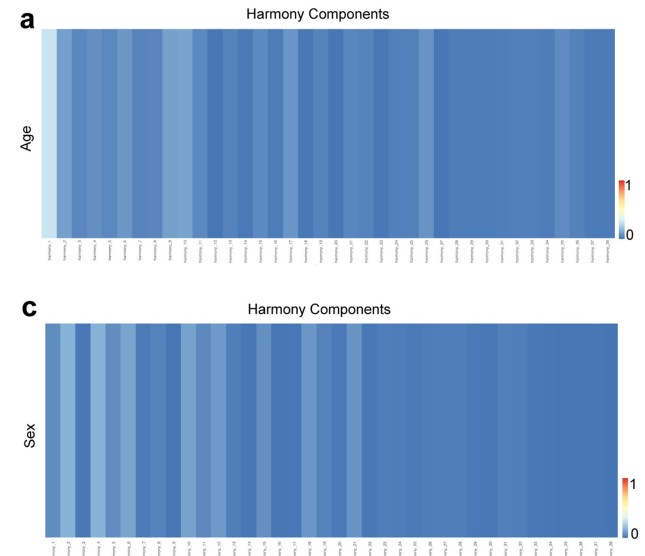

**c**

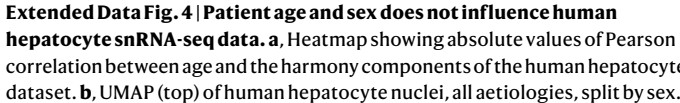

**b**

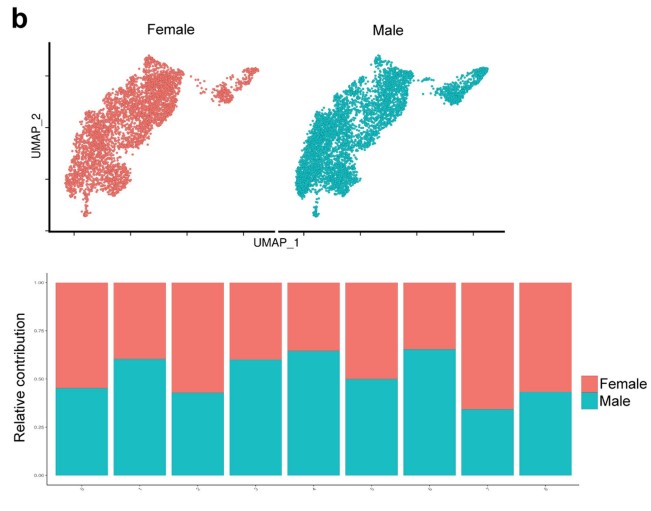

**Extended Data Fig. 4 | Patient age and sex does not influence human hepatocyte snRNA-seq data. a**, Heatmap showing absolute values of Pearson correlation between age and the harmony components of the human hepatocyte dataset. **b**, UMAP (top) of human hepatocyte nuclei, all aetiologies, split by sex.

Barplot (bottom) displaying relative contribution of female and male to each hepatocyte cluster. **c**, Heatmap showing absolute values of point-biserial correlation between sex and the harmony components of the human hepatocyte dataset.

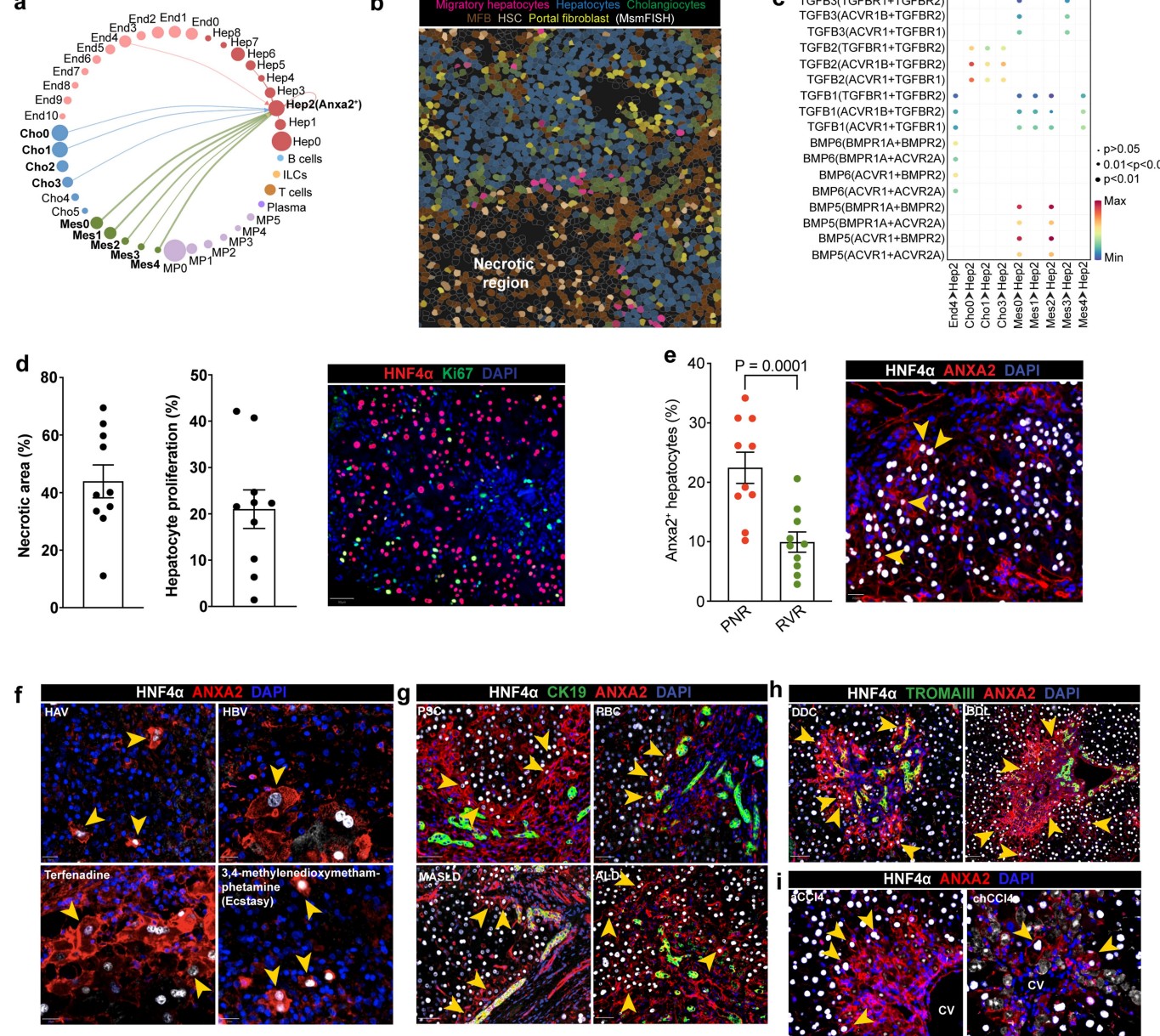

**Extended Data Fig. 5 | Interactome of human ANXA2+ hepatocytes and identification of ANXA2+ hepatocytes in multiple aetiologies of human and mouse liver injury. a**, Circle plot showing the interacting partners of the human migratory hepatocytes (top 20% of interactions). Arrows denote direction from ligand to receptor. Line widths denote scaled interaction strength, dot sizes represent cell number proportions. **b**, Spatial expression (MsmFISH) in APAP-ALF human liver showing lineage interacting partners from (**a**) in relation to migratory hepatocytes. Gene modules in Supplementary Table 2. MFB, myofibroblasts (mes1); HSC, hepatic stellate cells (mes2). **c**, Bubble plots showing TGFβ and BMP ligand-receptor pairs for those interacting partners displayed in (**a**). Dot colour denotes communication probability, dot size denotes significance. Empty spaces show a communication probability of zero. **d**, Quantification of necrotic area (left) and hepatocyte proliferation (right), and representative immunofluorescence of transjugular biopsies from patients with acute, severe liver injury (n = 10) liver tissue. Data are mean ± SEM. HNF4α (hepatocytes, red), Ki67 (green), and DAPI (nuclear stain, blue). Scale bar 50 μm. **e**, Hepatocyte ANXA2 expression in the peri-necrotic region (PNR) and remnant viable region (RVR) of transjugular biopsies from acute, severe liver injury (n = 10, left). Representative immunofluorescence of HNF4α (hepatocytes,

white), ANXA2 (red), and DAPI (blue) in transjugular biopsies from acute, severe liver injury (right). Scale bar 20 μm. Two tailed paired Student's t-test, t = 6.4, df = 9. Data are mean ± SEM. **f**, Representative immunofluorescence images of ANXA2 (red), HNF4α (hepatocytes, white) and DAPI (blue) in hepatitis A-induced ALF (n = 1), hepatitis B-induced ALF, (n = 5) and drug-induced ALF (n = 2). Yellow arrowheads denote ANXA2+ hepatocytes with migratory phenotype. Scale bar 20 μm. **g**, Representative immunofluorescence of ANXA2 (red), HNF4α (hepatocytes, white), CK19 (cholangiocytes, green), and DAPI (blue) in MASLD (metabolic dysfunction-associated steatotic liver disease, n = 5), PBC (primary biliary cholangitis, n = 2), PSC (primary sclerosing cholangitis, n = 5), and ALD (alcohol-induced liver disease, n = 2). Scale bar 50 μm. **h**, Representative immunofluorescence of ANXA2 (red), HNF4α (hepatocytes, white), TROMAIII (cholangiocytes, green), and DAPI (blue) in DDC (3,5-diethoxycarbonyl-1,4-dihydrocollidine) diet-induced (n = 5) and BDL (bile duct ligation) surgical mouse liver injury (n = 3). Scale bar 50 μm. **i**, Representative immunofluorescence images of ANXA2 (red), HNF4α (hepatocytes, white), and DAPI (blue) in acute (aCCl₄, 42hrs post-injection, n = 3) and chronic (chCCl₄) carbon tetrachloride (6 weeks, n = 3) injury. Yellow arrowheads denote ANXA2+ hepatocytes. Scale bar 20 μm.

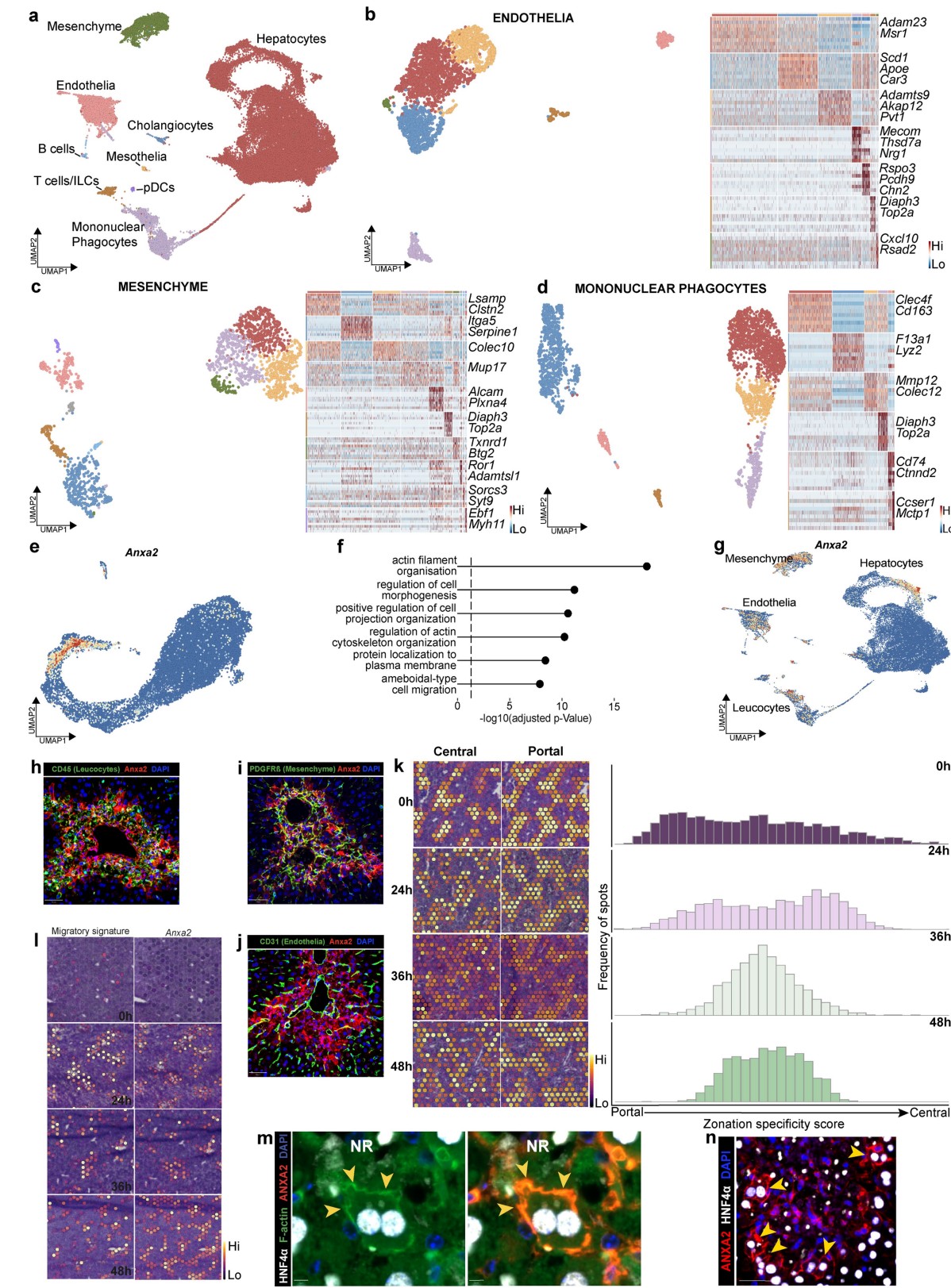

**Extended Data Fig. 6** | See next page for caption.

**Extended Data Fig. 6 | Atlas of mouse APAP-induced acute liver injury.**
**a**, UMAP visualisation of mouse liver nuclei, annotated by lineage inferred using signatures of known lineage markers (Supplementary Table 2). UMAPs of (**b**) of mouse endothelia, (**c**) mesenchyme and (**d**) mononuclear phagocytes derived from mouse liver single nuclei, annotated by clustering (left); heatmap (right) of marker genes (Supplementary Table 3; colour-coded by cluster) with exemplar genes labelled (right). Columns denote cells, rows denote genes. **e**, UMAP of mouse hepatocyte nuclei, all time points, showing *Anxa2* gene expression. **f**, GO terms enriched in mouse migratory hepatocyte cluster (Supplementary Table 4). **g**, UMAP of mouse nuclei, all time points, showing *Anxa2* gene expression. **h**, Representative immunofluorescence image of ANXA2 (red), CD45 (leucocytes, green), and DAPI (nuclear marker, blue) at 48hrs post APAP-induced liver injury. n = 6. Scale bar 50 μm. **i**, Representative immunofluorescence image of ANXA2 (red), PDGFRβ (mesenchyme, green), and DAPI (blue) at 48hrs post APAP-induced liver injury. n = 6. Scale bar 50 μm.

**j**, Representative immunofluorescence image of ANXA2 (red), CD31 (endothelia, green), and DAPI (blue) at 48hrs post APAP-induced liver injury. n = 6. Scale bar 50 μm. **k**, Spatial expression (ST) in selected timepoints post APAP-induced liver injury (left) of mouse liver-derived zonation gene modules (Supplementary Table 2). Distribution of zonation scores across selected timepoints post APAP-induced liver injury (right). **l**, Spatial transcriptomic expression of *Anxa2* (top) and migratory gene signature (bottom; Supplementary Table 2) in mouse liver post APAP-induced liver injury. **m**, Representative immunofluorescence images of F-actin (green), ANXA2 (red), HNF4α (hepatocytes, white), and DAPI (blue) at 42hrs post APAP-induced liver injury. NR, necrotic region. n = 3. Scale bar 5 μm. **n**, Representative immunofluorescence images of ANXA2 (red), HNF4α (hepatocytes, white), and DAPI (blue) in mouse liver at 42hrs post APAP-induced liver injury (left). Yellow arrowheads denote ANXA2[+] hepatocytes. n = 6. Scale bar 50 μm.

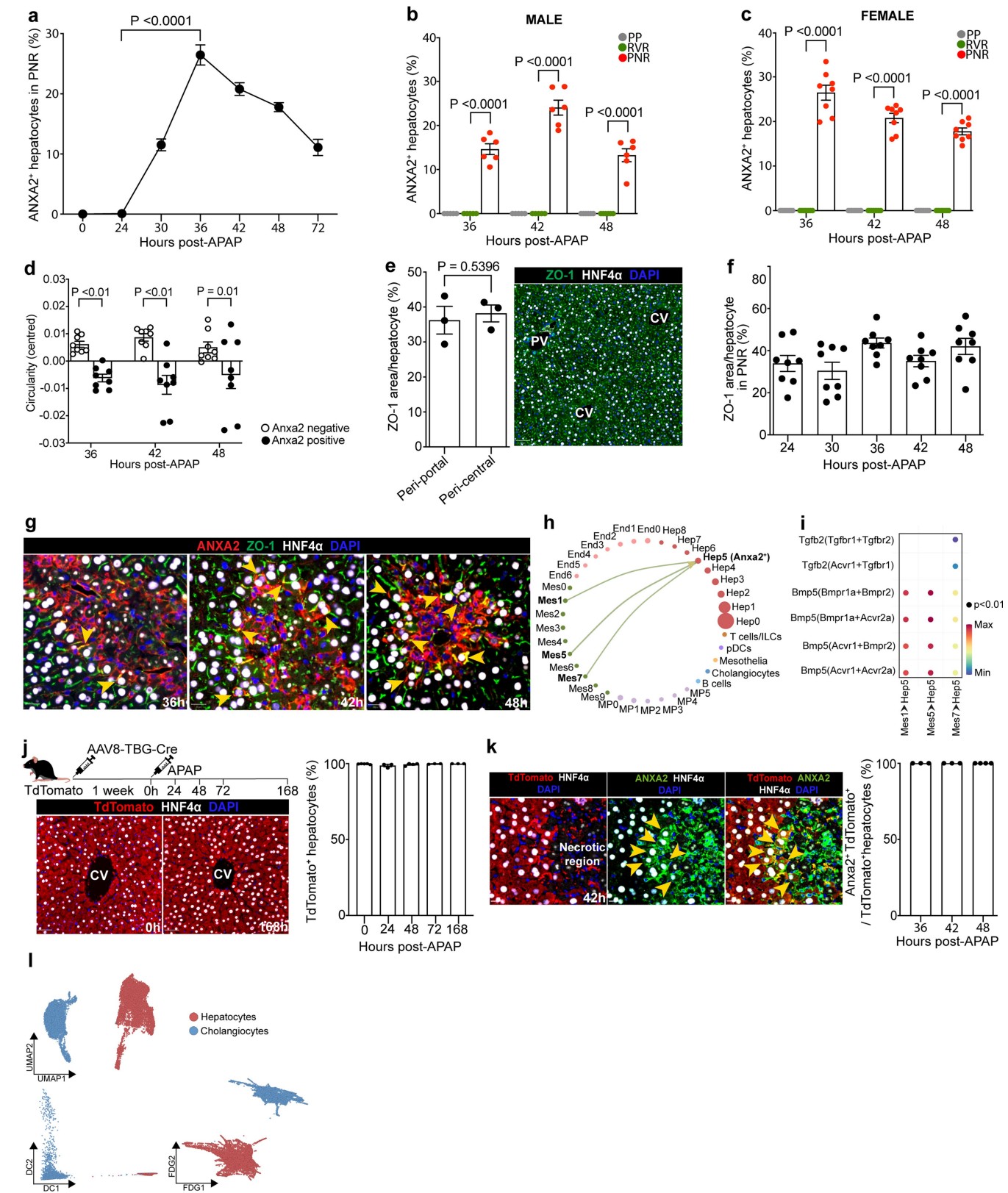

**Extended Data Fig. 7 | See next page for caption.**

**Extended Data Fig. 7 | Characterisation of mouse ANXA2[+] hepatocytes and maintenance of epithelial sheet connections and hepatocyte polarity during wound closure. a**, Percentage ANXA2-positive hepatocytes in the peri-necrotic region (PNR) post APAP-induced liver injury in female mouse liver injury. Two-way ANOVA, n = 4 (0 h), n = 8 (24–72 h), df = 45. Data are mean ± SEM. **b**, Percentage ANXA2[+] hepatocytes in male APAP-induced mouse liver injury. PP, peri-portal; RVR, remnant viable region. Two-way ANOVA, n = 6, $F$ = 15.85, df = 2,15. Data are mean ± SEM. **c**, Percentage ANXA2-positive hepatocytes in female APAP-induced liver injury. Two-way ANOVA, n = 8, df = 7. Data are mean ± SEM. **d**, Circularity of ANXA2-positive and ANXA2-negative hepatocytes in the PNR post APAP-induced liver injury. Two-way ANOVA, n = 8 (36–42 h), df = 42. Data are centred, mean ± SEM. **e**, Quantification (left) of ZO-1 expression in the peri-portal and peri-central regions in uninjured mouse liver. Two-tailed paired Student's t-test, t = 0.73, df = 2. Data are mean ± SEM. Representative immunofluorescence image (right) of ZO-1 (green), HNF4α (hepatocytes, white), and DAPI (nuclear stain, blue) expression in uninjured mouse liver. PV, portal vein, CV, central vein. Scale bar 50 μm. **f**, Quantification of ZO-1 expression in the PNR post APAP-induced liver injury. n = 8 (24–48 h), df = 28. Data are mean ± SEM. **g**, Representative immunofluorescence image of ZO-1 (green), ANXA2 (red), HNF4α (hepatocytes, white), and DAPI (blue) expression across select timepoints following APAP-induced liver injury. Yellow arrowheads denote ANXA2[+] hepatocytes expressing ZO-1. NR, necrotic region. Scale bar 20 μm. **h**, Circle plot showing the interacting partners of the mouse migratory hepatocytes (top 20% of interactions). Arrows denote direction from ligand to receptor. Line widths denote scaled interaction strength, dot sizes represent cell number proportions. **i**, Bubble plot showing TGFβ and BMP ligand-receptor pairs for those interacting partners displayed in (**h**). Dot colour denotes communication probability, dot size denotes significance. Empty spaces show a communication probability of zero. **j**, Schematic depicting experimental protocol for lineage tracing of hepatocytes in AAV8.TBG.Cre-activated *R26^{LSL}tdTomato* mice post APAP-induced liver injury (top). Representative immunofluorescence images of hepatocytes (tdTomato, red), HNF4α (hepatocytes, white), and DAPI (blue) in select timepoints post APAP-induced liver injury (left, scale bar 50 μm). TdTomato[+] hepatocytes (HNF4α[+]) as a percentage of all hepatocytes post APAP-induced liver injury (right). One-way ANOVA, n = 5 (0 h), n = 3 (24 h, 72 h, 168 h), n = 4 (48 h), $F$ = 1.61, df = 4,13. Data are mean ± SEM. **k**, Representative immunofluorescence images of hepatocytes (tdTomato, red), ANXA2 (green), HNF4α (hepatocytes, white), and DAPI (blue) in AAV8.TBG.Cre-activated *R26^{LSL}tdTomato* mice 42hrs post APAP-induced liver injury (left). Yellow arrowheads denote ANXA2[+]TdTomato[+] hepatocytes. Scale bar 20 μm. ANXA2[+]TdTomato[+] hepatocytes (HNF4α[+]) as a percentage of all TdTomato[+] hepatocytes at peak (ANXA2[+] hepatocyte) timepoints post APAP-induced liver injury (right). n = 3 (36 h, 42 h), n = 4 (48 h). Data are mean ± SEM. **l**, Visualisation by UMAP (top), diffusion map (DC, bottom left), and force-directed graph (FDG, bottom right) of human hepatocytes and cholangiocytes from healthy, APAP-ALF, and NAE-ALF human liver explants.

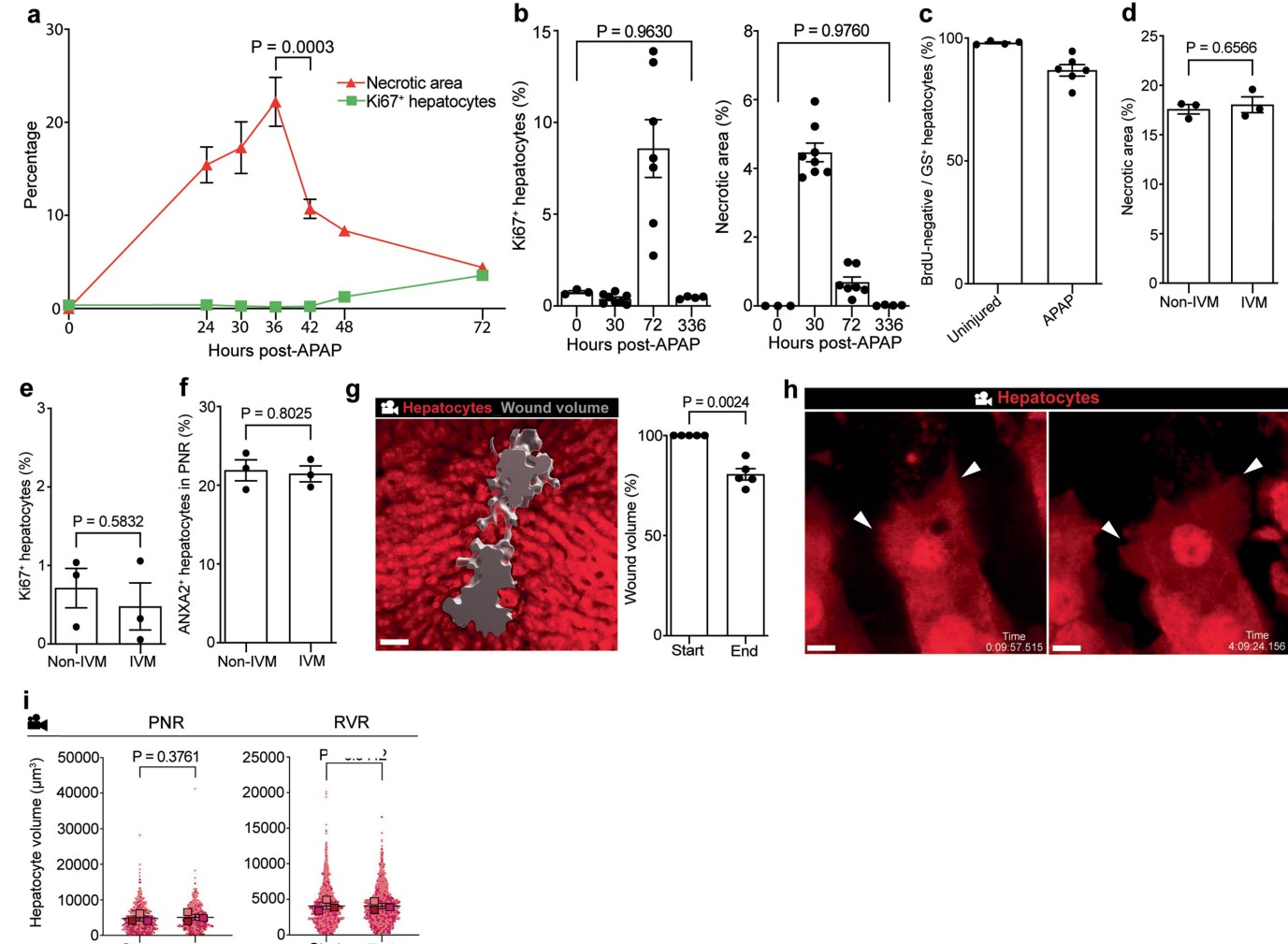

**Extended Data Fig. 8 | Hepatocytes immediately adjacent to the central vein do not arise from hepatocyte proliferation. a**, Quantification of necrotic area (red) and hepatocyte proliferation (green) following APAP-induced liver injury in female mice. One-way ANOVA, n = 4 (0 h), n = 8, df = 45. Data are mean ± SEM. **b**, Quantification of hepatocyte proliferation (left) and necrotic area (right) following APAP-induced liver injury in male mice. Two-way ANOVA, n = 3 (0 h), n = (30 h), n = (72 h), n = 4 (336 h), df = 11. Data are mean ± SEM. **c**, Quantification of percentage BrdU-negative/glutamine synthetase (GS)⁺ hepatocytes adjacent to the central vein following APAP-induced liver injury in female mice. n = 4 (uninjured), n = 6 (APAP). Data are mean ± SEM. Quantification of necrotic area (**d**), Ki67⁺ hepatocytes (**e**), and ANXA2⁺ hepatocytes in the PNR (**f**), following APAP-induced liver injury in IVM and non-IVM (42 h) mice. Two tailed unpaired Student's t-test, n = 3. Data are mean ± SEM. **g**, Representative IVM snapshot of APAP-induced liver injury with volume rendering of wound area (left). Scale bar 50 μm. Quantification of change in wound volume at start and end of IVM imaging session (right). Two-tailed paired Student's t-test, n = 5, t = 6.82, df = 4. Data are mean ± SEM. **h**, Representative IVM snapshots of motile hepatocytes, white arrowheads marking areas of membrane ruffling/lamellipodia formation. Supplementary Videos 6–14. n = 16, three independent experiments. Scale bar 5 μm. **i**, Quantification of hepatocyte volume at start and end of IVM, for each mouse. Two-tailed paired Student's t-test, n = 2 (mouse 1), n = 3 (mouse 2,3) regions per mouse. Data are mean ± SEM.

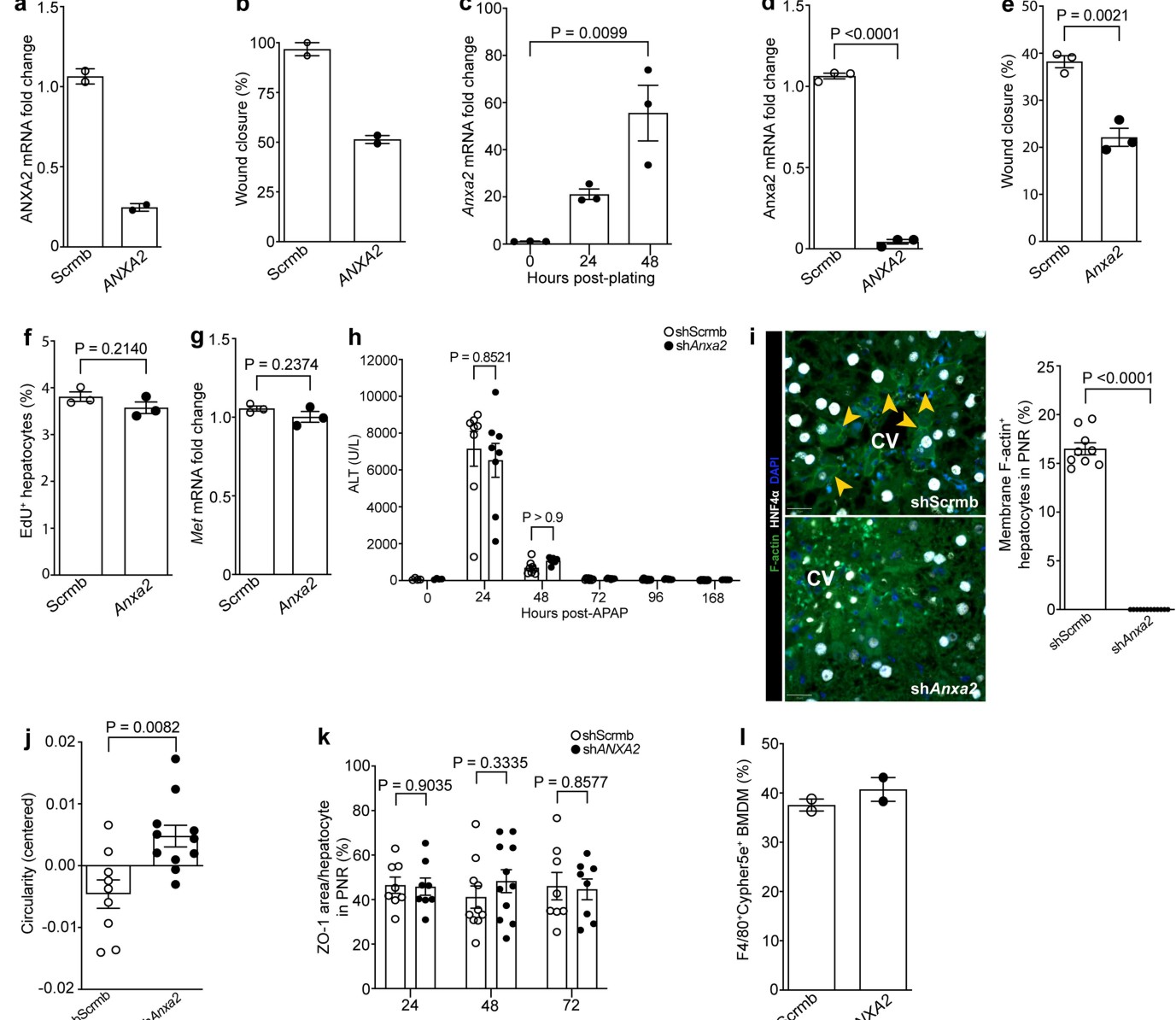

**Extended Data Fig. 9 | Hepatocyte ANXA2 regulates wound closure.**
**a**, *ANXA2* gene expression (RT-qPCR analysis) in Huh7 cells treated with Scrmb (control) or *ANXA2* siRNA. Data are mean ± SEM, two independent experiments. **b**, Percent coverage of scratch wound area 72hrs post-wounding in Huh7 cells (human hepatocyte cell line), treated with Scrmb (control) or *ANXA2*-siRNA. Data are mean ± SEM, two independent experiments. **c**, Timecourse of *Anxa2* gene expression (RT-qPCR analysis) in plated primary mouse hepatocytes. Two-tailed unpaired Student's t-test, t = 4.62, df = 4, three independent experiments. Data are mean ± SEM. **d**, *Anxa2* gene expression (RT-qPCR) in primary mouse hepatocytes treated with Scrmb (control) or *Anxa2* siRNA. Two-tailed unpaired Student's t-test, t = 45.63, df = 4. Data are mean ± SEM, three independent experiments. **e**, Percent coverage of scratch wound area 72hrs post-wounding of primary mouse hepatocytes, treated with Scrmb (control) or *Anxa2*-siRNA. Two-tailed unpaired Student's t-test, t = 7.04, df = 4, three independent experiments. Data are mean ± SEM. **f**, Percent EdU⁺ hepatocytes 72hrs post-wounding of primary mouse hepatocytes, treated with Scrmb (control) or *Anxa2*-siRNA. Two-tailed unpaired Student's t-test, t = 1.48, df = 4, three independent experiments. Data are mean ± SEM. **g**, *Met* gene expression (RT-qPCR analysis) in primary mouse hepatocytes, treated with Scrmb (control) or *Anxa2*-siRNA. Two-tailed unpaired Student's t-test, t = 1.39,

df = 4, three independent experiments. Data are mean ± SEM. **h**, Alanine transaminase (ALT) following APAP-induced liver injury in AAV8-shScrmb or AAV8-sh*Anxa2* treated mice. Two-tailed unpaired Student's t-test, n = 4 (0 h), n = 8 (24–168 h). Data are mean ± SEM. **i**, Representative immunofluorescence images (left) of F-actin (green), HNF4α (hepatocytes, white), and DAPI (nuclear stain, blue) at 48hrs post APAP-induced liver injury in AAV8-shScrmb or AAV8-sh*Anxa2* treated male mice. CV, central vein. Scale bar 20 μm. Quantification of membrane F-actin⁺ hepatocytes in the PNR at 48hrs post APAP-induced liver injury in AAV8-shScrmb or AAV8-sh*Anxa2* treated male mice (right). Two-tailed unpaired Student's t-test, n = 9 (shScrmb), n = 11 (sh*Anxa2*), t = 29.84, df = 18. Data are mean ± SEM. **j**, Circularity of hepatocytes in the peri-necrotic region (PNR) at 48hrs following APAP-induced liver injury in AAV8-shScrmb or AAV8-sh*Anxa2* treated male mice. Two-tailed unpaired Student's t-test, n = 9 (shScrmb), n = 11 (sh*Anxa2*), t = 3.29, df = 18. Data are mean ± SEM. **k**, Quantification of ZO-1 expression in the PNR post APAP-induced liver injury in AAV8-shScrmb or AAV8-sh*Anxa2* treated male mice. Two-tailed unpaired Student's t-test. n = 8 (24 hr,72 hr), n = 11 (48 hr). Data are mean ± SEM. **l**, Percentage of bone marrow-derived macrophages (BMDM) that have phagocytosed control (Scrmb) compared to *Anxa2*-siRNA treated primary mouse hepatocytes. Data are mean ± SEM, two independent experiments.

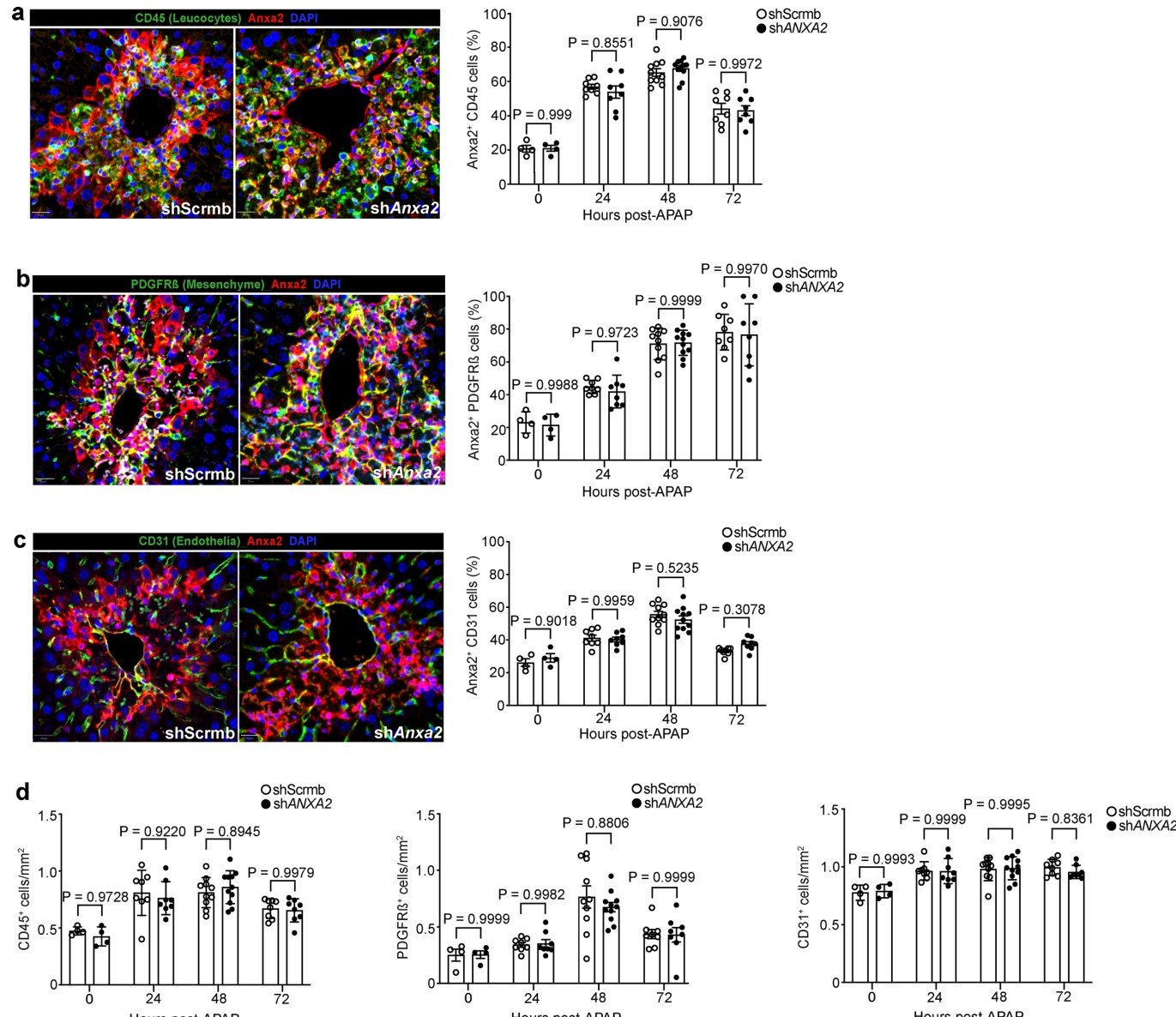

**Extended Data Fig. 10 | AAV8-shRNA-*Anxa2* does not affect ANXA2 expression in non-hepatocyte lineages. a**, Representative immunofluorescence images of CD45 (leucocytes, green), ANXA2 (red), and DAPI (blue) in AAV8-shRNA-Scrmb or AAV8-shRNA-*Anxa*2 treated mice 48hrs post APAP-induced liver injury (left). Scale bar 20 μm. Quantification of CD45⁺/ANXA2⁺ cells following APAP-induced liver injury in AAV8-shScrmb or AAV8-sh*Anxa*2 treated mice (right). Two-way ANOVA, n = 4 (0 h), n = 8 (24 h), n = 11 (48 h), n = 8 (72 h). Data are mean ± SEM. **b**, Representative immunofluorescence staining of PDGFRβ (mesenchyme, green), ANXA2 (red), and DAPI (blue) in AAV8-shRNA-Scrmb or AAV8-shRNA-*Anxa*2 treated mice 48hrs post APAP-induced liver injury (left). Scale bar 20 μm. Quantification of PDGFRβ⁺/ANXA2⁺ cells following APAP-induced liver injury in AAV8-shScrmb or AAV8-sh*Anxa*2 treated mice (right). Two-way ANOVA, n = 4 (0 h), n = 8 (24 h), n = 11 (48 h), n = 8 (72 h). Data are mean ± SEM. **c**, Representative immunofluorescence staining of CD31 (endothelia, green), ANXA2 (red), and DAPI (blue) in AAV8-shRNA-Scrmb or AAV8-shRNA-*Anxa*2 treated mice 48 hrs post APAP-induced liver injury (left). Scale bar 20 μm. Quantification of CD31⁺/ANXA2⁺ cells following APAP-induced liver injury in AAV8-shScrmb or AAV8-sh*Anxa*2 treated mice (right). Two-way ANOVA, n = 4 (0 h), n = 8 (24 h), n = 11 (48 h), n = 8 (72 h). Data are mean ± SEM. **d**, Quantification of PDGFRβ⁺, CD45⁺, and CD31⁺ cells/mm² following APAP-induced liver injury in AAV8-shScrmb or AAV8-sh*Anxa*2 treated mice. Two-way ANOVA, n = 4 (0 h), n = 8 (24 h), n = 11 (48 h), n = 8 (72 h). Data are mean ± SEM.

| | |
|---|---|

# Reporting Summary

## Statistics

For all statistical analyses, confirm that the following items are present in the figure legend, table legend, main text, or Methods section.

| n/a | Confirmed | |
|---|---|---|
| ☐ | ☒ | The exact sample size (*n*) for each experimental group/condition, given as a discrete number and unit of measurement |
| ☐ | ☒ | A statement on whether measurements were taken from distinct samples or whether the same sample was measured repeatedly |
| ☐ | ☒ | The statistical test(s) used AND whether they are one- or two-sided<br>*Only common tests should be described solely by name; describe more complex techniques in the Methods section.* |
| ☐ | ☒ | A description of all covariates tested |
| ☐ | ☒ | A description of any assumptions or corrections, such as tests of normality and adjustment for multiple comparisons |
| ☐ | ☒ | A full description of the statistical parameters including central tendency (e.g. means) or other basic estimates (e.g. regression coefficient) AND variation (e.g. standard deviation) or associated estimates of uncertainty (e.g. confidence intervals) |
| ☐ | ☒ | For null hypothesis testing, the test statistic (e.g. *F*, *t*, *r*) with confidence intervals, effect sizes, degrees of freedom and *P* value noted<br>*Give P values as exact values whenever suitable.* |
| ☒ | ☐ | For Bayesian analysis, information on the choice of priors and Markov chain Monte Carlo settings |
| ☒ | ☐ | For hierarchical and complex designs, identification of the appropriate level for tests and full reporting of outcomes |
| ☐ | ☒ | Estimates of effect sizes (e.g. Cohen's *d*, Pearson's *r*), indicating how they were calculated |

*Our web collection on statistics for biologists contains articles on many of the points above.*

## Software and code

Policy information about availability of computer code

| | |
|---|---|
| Data collection | Initial processing of single-nuclei RNA-sequencing data was performed using the commercial CellRanger pipeline (10X Genomics, version 3.1.0 (see methods)). Initial processing of spatial transcriptomics data was performed using the commerical SpaceRanger pipeline (10X Genomics, version 1.0.0 (see methods)). Multiplex smFISH data was processed via Resolve Biosciences. Subsequent analyses were performed using the open-source R programming language (version 3.4.1).<br>Fluorescent and brightfield microscopy images were acquired using Zen Blue software (Zeiss v2.6) on an Axioscan.Z1 instrument (Zeiss) or EVOS FL Auto 2. Flow cytometry data was acquired using a 5laser Fortessa cytometer (Becton Dickinson). Intravital microscopy images were acquired using LSM 880 NLO multiphoton microscope (Zeiss). Scratch wound data was acquired on an IncuCyte ZOOM live cell analysis system (Essen biosciences). RT-qPCR data was acquired on ABI Quantstudio 5 PCR system (Applied Biosystems). |
| Data analysis | Immunofluorescent and brightfield images were acquired using Zeiss Axioscan slide scanner (Z1) and Zen Blue software (v2.6) and analysed using QuPath (version 0.3.0) for automated cell counting and necrotic area analysis.<br>Scratch wound data were analysed on the IncuCyte proprietary scratch wound analysis software (version 2018A).<br>Intravital microscopy data were processed using Imaris 9.7 (Bitplane) and analysed using Imaris 9.7 (Bitplane) and ZerocostDL4mic, see Methods).<br>Flow cytometry data was analysed using Flowjo 10.9.0. RT-qPCR data were processed and analysed using Design & Analysis software (2.6.0, Quantstudio). Statistical analysis was performed using GraphPad Prism software version 9.4.1.<br>Single-nuclei sequencing analysis was performed in R (version 3.4.1), based around the following packages: Seurat R package 4.1.1, SeuratPipe R package 1.0.0, Scrublet python module 0.2.3, SoupX R package 1.5.2, Harmony R package 0.1.0. Spatial transcriptomics analysis was performed in R (version 3.4.1) using Seurat R package 4.1.1 and SPATA2 R package 0.1.0. Gene Ontology enrichment analysis was performed using the clusterProfiler R package 4.8.3. Interactome analysis was performed using CellChat R package v1.6.1. Diffusion maps and force- |

directed graphs were generated in Scanpy python module v1.9. All code is made available at https://github.com/HendersonLab/LiverRegenerationAtlas.

For manuscripts utilizing custom algorithms or software that are central to the research but not yet described in published literature, software must be made available to editors and reviewers. We strongly encourage code deposition in a community repository (e.g. GitHub). See the Nature Portfolio guidelines for submitting code & software for further information.

## Data

Policy information about availability of data

All manuscripts must include a data availability statement. This statement should provide the following information, where applicable:
- Accession codes, unique identifiers, or web links for publicly available datasets
- A description of any restrictions on data availability
- For clinical datasets or third party data, please ensure that the statement adheres to our policy

Our expression data will be freely available for user-friendly interactive browsing online at www.LiverRegenerationAtlas.hendersonlab.mvm.ed.ac.uk. All raw sequencing data have been deposited in the Gene Expression Omnibus (GEO Accession GSE223561). We make available as Supplementary Tables: lists of lineage-specific genes for signature analysis, lists of marker genes from clustering results, lists of gene ontology terms from enrichment analysis, lists of interactome analysis output. Source data for mouse experiments are also provided with this paper as a Supplementary Table.

The Ensembl 93 human (GRCh38) and mouse (mm10) reference genomes used in this study, originally available at https://www.ensembl.org/info/website/archives/index.html, were built by 10X Genomics and hosted at https://www.10xgenomics.com/support/software/cell-ranger/downloads#reference-downloads.

## Human research participants

Policy information about studies involving human research participants and Sex and Gender in Research.

| Reporting on sex and gender | Gender and age information of patients used for single nuclei-RNA sequencing are reported in SI Table 1. |
|---|---|
| Population characteristics | Specimens were obtained from the Scottish Liver Transplant Unit at the Royal Infirmary of Edinburgh, Edinburgh, UK; United States Acute Liver Failure Study Group network; Addenbrookes Hospital, Cambridge, UK; Queen Elizabeth Hospital, Birmingham, UK and University College London, London, UK. Patient gender and age for the snRNAseq datasets are reported in SI Table 1. |
| Recruitment | Participants were recruited based on meeting criteria for liver transplantation for acute liver failure. Healthy non-lesional liver tissue was obtained from patients undergoing surgical liver resection for solitary colorectal metastasis at the Hepatobiliary and Pancreatic Unit, Department of Clinical Surgery, Royal Infirmary of Edinburgh. Patients with a known history of chronic liver disease, abnormal liver function tests or those who had received systemic chemotherapy within the last four months were excluded from this cohort. |
| Ethics oversight | University of Edinburgh, UK: Local approval for procuring human liver tissue for single-nuclei RNA sequencing, spatial transcriptomics, and histological analysis was obtained from the Scotland 'A' Research and Ethics Committee (16/SS/0136) and the NRS BioResource and Tissue Governance Unit (study number SR574), following review at the East of Scotland Research Ethics Service (reference15/ES/0094). Written informed consent was obtained from the subject or a legally authorised representative prior to enrolment per local regulations. Acute liver failure liver tissue was obtained intraoperatively from patients undergoing orthotopic liver transplantation at the Scottish Liver Transplant Unit, Royal Infirmary of Edinburgh. Patient demographics are summarized in SI Table 1 for patients transplanted for APAP-induced ALF and nonA-E ALF. Healthy non-lesional liver tissue was obtained intraoperatively from patients undergoing surgical liver resection for solitary colorectal metastasis at the Hepatobiliary and Pancreatic Unit, Department of Clinical Surgery, Royal Infirmary of Edinburgh. Patients with a known history of chronic liver disease, abnormal liver function tests or those who had received systemic chemotherapy within the last four months were excluded from this cohort. For histological assessment of human ALF and chronic liver disease tissue, anonymized unstained formalin-fixed paraffin-embedded liver tissue sections were provided by the Lothian NRS Human Annotated Bioresource under authority from the East of Scotland Research Ethics Service REC 1, reference15/ES/0094.

United States Acute Liver Failure Study Group (ALFSG) network: This consortium of U.S. liver centers was established in 1998 to better define causes and outcomes of acute liver injury and ALF. The study protocol was approved by the local institutional review boards of the participating sites: University of Texas Southwestern Medical Center; Baylor University Medical Center, Dallas, TX; Medical University of South Carolina, Charleston, SC; University of Washington, Seattle, WA; Washington University, St. Louis, MO; University of California, San Francisco, and California Pacific Medical Center, San Francisco, CA; University of Nebraska, Omaha, NE; Mount Sinai Medical Center and Columbia University Medical Center, New York, NY; Mayo Clinic, Rochester, MN; University of Pittsburgh, Pittsburgh, PA; Northwestern University, Chicago, IL; Oregon Health Sciences Center, Portland, OR; University of California, Los Angeles, CA; University of Michigan, Ann Arbor, MI; Yale University, New Haven, CT; University of Alabama, Birmingham, AL; Massachusetts General Hospital, Boston, MA; Duke University, Durham, NC; Mayo Clinic, Scottsdale, AZ; Albert Einstein Medical Center and University of Pennsylvania, Philadelphia, PA; Virginia Commonwealth University, Richmond,VA; University of California, Davis, CA; Mayo Clinic, Jacksonville, FL; University of California, San Diego, CA; The Ohio State University, Columbus, OH; University of Kansas Medical Center, Kansas City, KS; Emory University, Atlanta, GA; University of Alberta, Edmonton, Canada. Written informed consent was obtained from the subject or a legally authorized representative prior to enrolment per local regulations. Sites obtained portions of fresh explanted liver tissue cut into 1cm3 pieces, placed into individual cryovials and stored at −80°C until requested for study. The ALFSG was supported by the National Institute of Diabetes and Digestive and Kidney Diseases (NIDDK; grant no.: U-01-58369). The samples used in this study were supplied by the NIDDK Central Repositories. This article |

does not necessarily reflect the opinions or views of the NIDDK Central Repositories or the NIDDK.

University of Cambridge, UK: Patients were recruited at Addenbrooke's Hospital, Cambridge, UK with approval from the Health and Social Care Research Ethics Committee A, Office for Research Ethics Committees, Northern Ireland (ORECNI) (16/NI/0196 & 20/NI/0109). Written informed consent was obtained from the subject or a legally authorised representative prior to enrolment per local regulations. Liver tissue from patients with ALF was derived from explanted livers at the time of transplantation. All tissue samples were snap-frozen in liquid nitrogen and stored at −80°C in the Human Research Tissue Bank of the Cambridge University Hospitals NHS Foundation Trust.

University of Birmingham, UK: Human liver tissue obtained from the University of Birmingham, UK was obtained under approval by South Birmingham Ethics Committee, Birmingham, UK (reference 06/Q2708/11, 06/Q2702/61), and written informed consent was obtained from the subject or a legally authorised representative prior to enrolment per local regulations. Liver tissue was acquired from explanted livers from patients undergoing orthotopic liver transplantation at the Queen Elizabeth Hospital, Birmingham. All tissue samples were snap-frozen in liquid nitrogen and stored at −80°C before being processed and shipped by the Birmingham Human Biomaterials Resource Centre (reference 09/H1010/75; 18-319).

University College, London, UK: Human liver tissue obtained from University College, London, UK was obtained under local ethical approval (London-Hampstead Research Ethics Committee, reference 07/Q0501/50). Written informed consent was obtained from the subject or a legally authorised representative prior to enrolment per local regulations. Liver tissue (formalin-fixed, paraffin-embedded) was acquired via transjugular liver biopsy from patients presenting with acute, severe liver injury, and these patients spontaneously recovered without liver transplantation.

Note that full information on the approval of the study protocol must also be provided in the manuscript.

# Field-specific reporting

Please select the one below that is the best fit for your research. If you are not sure, read the appropriate sections before making your selection.

☒ Life sciences  ☐ Behavioural & social sciences  ☐ Ecological, evolutionary & environmental sciences

For a reference copy of the document with all sections, see nature.com/documents/nr-reporting-summary-flat.pdf

# Life sciences study design

All studies must disclose on these points even when the disclosure is negative.

| | |
|---|---|
| Sample size | Sample size for every experiment performed and image acquired is presented in figure legends. Supplementary Table 1 also provides more detail on sample size in patient groups.<br>For imaging data, images in the manuscript are representative of a minimum of 4 samples within an experiment. In one experiment (Extended Data Fig. 5f) one of the human ALF aetiologies studied (HAV-induced) was n=1 (Extended Data Fig. 5f). Human sample size was determined by tissue availability.<br>For quantitative analyses no sample size calculation was performed, but the sample size / replicate number was chosen in order to provide sufficient data points for the statistical analyses.<br><br>Statistical significance of reported results was assessed by statistical tests as indicated in Methods section. Statistical significance is stated in each figure legend. |
| Data exclusions | Described in detail in Methods. Exclusion criteria were determined following initial assessment and QC of the data. Low gene expression (fewer than 1000 genes) or mitochondrial gene content <5% of the total UMI count, are indicators of outlier low quality cells and were excluded. At each stage of the analysis signature analysis was used to identify and exclude potential doublet clusters. |
| Replication | All experimental findings reported here were successfully replicated across multiple biological samples ('n' reported in each figure legend). All mouse experimental immunofluorescence and histology analyses were performed on a minimum of 3 liver samples to identify representative images. |
| Randomization | One group of randomly selected healthy liver samples and two groups of randomly selected acute liver failure samples were analysed in this study. All subsequent analyses were performed in randomly selected healthy or acute liver failure samples. For mouse experiments, age- and sex-matched mice were randomly assigned to treatment groups. |
| Blinding | Blinding to the origin of the tissue samples was not performed. The investigators performing the immunofluorescence staining, snRNA-seq, and RT-qPCR were different from the investigators harvesting tissue. The IVM studies were not blinded. |

# Reporting for specific materials, systems and methods

We require information from authors about some types of materials, experimental systems and methods used in many studies. Here, indicate whether each material, system or method listed is relevant to your study. If you are not sure if a list item applies to your research, read the appropriate section before selecting a response.

## Materials & experimental systems

| n/a | Involved in the study |
|---|---|
| ☐ | ☒ Antibodies |
| ☐ | ☒ Eukaryotic cell lines |
| ☒ | ☐ Palaeontology and archaeology |
| ☐ | ☒ Animals and other organisms |
| ☒ | ☐ Clinical data |
| ☒ | ☐ Dual use research of concern |

## Methods

| n/a | Involved in the study |
|---|---|
| ☒ | ☐ ChIP-seq |
| ☐ | ☒ Flow cytometry |
| ☒ | ☐ MRI-based neuroimaging |

## Antibodies

| Antibodies used | All antibodies used in this work - application, supplier and product code - are listed in Supplementary Table 6 or provided in the methods section. |
|---|---|

Validation

Antibodies listed in Supplementary Table 6 are widely used commercially available antibodies, and are validated by the companies with publications listed on the company websites. Below we provide links to these publication lists.

Validation statements from manufacturers websites:
-HNF4alpha (R&D Systems PP-H1415-00): https://www.rndsystems.com/products/human-hnf-4alpha-nr2a1-antibody-h1415_pp-h1415-00#product-citations
-HNF4alpha (Abcam ab41898): https://www.abcam.com/products/primary-antibodies/hnf-4-alpha-antibody-k9218-ab41898.html
-ANXA2 (Annexin A2) (Cell Signalling Technology 8235): https://www.cellsignal.com/products/primary-antibodies/annexin-a2-d11g2-rabbit-mab/8235
-Ki67 (Abcam ab15580): https://www.abcam.com/en-kr/products/primary-antibodies/ki67-antibody-ab15580
-Ki67 (Dako M724029-2): https://www.agilent.com/en/product/immunohistochemistry/antibodies-controls/primary-antibodies/ki-67-antigen-%28concentrate%29-76646
- CK19 (Abcam ab220193): https://www.abcam.com/products/primary-antibodies/cytokeratin-19-antibody-krt19800-ab220193.html
-RFP (Rockland Inc 600-401-379):
- https://www.rockland.com/categories/primary-antibodies/rfp-antibody-pre-adsorbed-600-401-379/
-CK18 (Abcam ab181597): https://www.abcam.com/products/primary-antibodies/cytokeratin-18-antibody-epr17347-ab181597.html
-HAL (Atlas Antibodies HPA038548):https://www.atlasantibodies.com/products/primary-antibodies/triple-a-polyclonals/anti-hal-antibody-hpa038548/
-CYP3a4 (Abcam ab3572): https://www.abcam.com/products/primary-antibodies/cytochrome-p450-3a4cyp3a4-antibody-ab3572.html
- Glutamine Synthetase (Abcam ab73593): https://www.abcam.com/products/primary-antibodies/glutamine-synthetase-antibody-ab73593.html
- BrdU (Thermo Fisher B35141): https://www.thermofisher.com/antibody/product/BrdU-Antibody-clone-MoBU-1-Monoclonal/B35141
- CD45 (Dako M0701): https://www.agilent.com/en/product/immunohistochemistry/antibodies-controls/primary-antibodies/cd45-leucocyte-common-antigen-%28concentrate%29-76507
- CD45 R&D Systems AF114): https://www.rndsystems.com/products/mouse-cd45-antibody_af114
- CD31 (Abcam ab182981): https://www.abcam.com/products/primary-antibodies/cd31-antibody-epr17259-ab182981.html
- PDGFRbeta (Abcam ab32570) https://www.abcam.com/products/primary-antibodies/pdgfr-alpha--pdgfr-beta-antibody-y92-c-terminal-ab32570.html)
- ZO-1 (Abcam ab251568): https://www.abcam.com/products/primary-antibodies/zo1-tight-junction-protein-antibody-epr19945-296-bsa-and-azide-free-ab251568.html
- F-actin (Bioss/Thermo Fisher bs-1571R): https://www.thermofisher.com/antibody/product/F-Actin-Antibody-Polyclonal/BS-1571R
- TROMA-III (DSHB ab2133570): https://dshb.biology.uiowa.edu/TROMA-III
- F4/80 BV785 (Biolegend 123141):https://www.biolegend.com/en-us/search-results/brilliant-violet-785-anti-mouse-f4-80-antibody-9919?GroupID=BLG5319&gclid=CjwKCAiAi6uvBhADEiwAWiyRdl_M3LH7PDKDh0FMChQ38p9JXsNzH8WfUR5bSbfwBkzUhgrvs7KXGhoCa04QAvD_BwE

## Eukaryotic cell lines

Policy information about cell lines and Sex and Gender in Research

| Cell line source(s) | Huh7 cells were procured from Cell Lines Service. Huh7 is an immortalised hepatoma cell line from a 57 year old Japanese male with well differentiated hepatocellular carcinoma. |
|---|---|
| Authentication | Huh7 cells were authenticated by the commercial supplier using STR profiling. |
| Mycoplasma contamination | Huh7 cells tested negative for mycoplasma contamination. |
| Commonly misidentified lines (See ICLAC register) | The cell line used, Huh7, is not registered by ICLAC as commonly misidentified. |

## Animals and other research organisms

Policy information about studies involving animals; ARRIVE guidelines recommended for reporting animal research, and Sex and Gender in Research

| | |
|---|---|
| Laboratory animals | Male and female mice aged 8 to 12 weeks were used for the relevant experiments. Mice were housed in conventional barrier unit facilities with conventional bedding, 12:12-hour light:dark cycle, ambient temperature control (21°C; humidity 40-60%) and access to food and water ad libitum, under pathogen–free conditions at the University of Edinburgh. Male mice were used for all experiments unless described otherwise in the figure legends. All experimental protocols were approved by the University of Edinburgh Animal Welfare and Ethics Board in accordance with UK Home Office regulations. C57BL/6JCrl mice were obtained from Charles River Laboratories (UK). mTmG (Jax 007676; B6.129(Cg)-Gt(ROSA)26Sortm4(ACTB-tdTomato,-EGFP)Luo/J) and TdTomato (Jax 007914; B6.Cg-Gt(ROSA)26Sortm14(CAG-tdTomato)Hze/J) reporter mice were obtained from Jackson Laboratories. |
| Wild animals | Study did not involve wild animals. |
| Reporting on sex | Male and female mice were used in this study. |
| Field-collected samples | The study did not involve samples collected from the field. |
| Ethics oversight | All experiments were performed in accordance with UK Home Office regulations. |

Note that full information on the approval of the study protocol must also be provided in the manuscript.

## Flow Cytometry

### Plots

Confirm that:

☒ The axis labels state the marker and fluorochrome used (e.g. CD4-FITC).

☒ The axis scales are clearly visible. Include numbers along axes only for bottom left plot of group (a 'group' is an analysis of identical markers).

☒ All plots are contour plots with outliers or pseudocolor plots.

☒ A numerical value for number of cells or percentage (with statistics) is provided.

### Methodology

| | |
|---|---|
| Sample preparation | Primary mouse hepatocytes were isolated using in vivo perfusion digestion. Primary mouse bone marrow-derived macrophages (BMDMs) were isolated from mice femurs and differentiated for 7 days in culture with the addition of 10^4U/ml CSF1 (colony stimulating factor-1). |
| Instrument | BD LSR Fortessa cytometer, Becton Dickinson |
| Software | BD FACS Diva software was used for flow cytometry on BD LSR Fortessa equipment. Data was analysed using Flowjo v10.9.0. |
| Cell population abundance | n/a |
| Gating strategy | Differentiated primary mouse BMDMs were confirmed using flow cytometry and classed as F4/80+. F4/80+ cells that had phagocytosed dead hepatocytes were identified as Cypher-5E+. Gates and boundaries were defined by comparison to FMO and unstained samples. Representative plots of the gating strategy are available in Supplementary Table 6. |

☒ Tick this box to confirm that a figure exemplifying the gating strategy is provided in the Supplementary Information.

