## [Peer Review File · Nature]

Manuscript Title: Multimodal decoding of human liver regeneration

Reviewer Comments & Author Rebuttals

Reviewer Reports on the Initial Version:

Referees' comments:

Referee #1 (Remarks to the Author):

Authors used a combination of snRNAseq, ST, smFISH and IF to identify gene modules differentially expressed across the lobule in the APAP-ALF and non-A-E human liver samples. Using these approaches, the authors demonstrated loss of hepatocyte portal-central zonation and emergence of hepatocytes with both portal and central characteristics. These data emphasize the hepatocyte plasticity in response to loss of pericentral hepatocytes. These data are impactful as they are obtained from patient samples and provide a robust data set that can serve as a valuable resource for new target and pathway investigations. A potential limitation is that because these samples were obtained at the time of liver transplantation, they represent the most severe cases and are not able to assess the pattern of resolution.

The second portion of the manuscript focuses on mechanistic studies to assess the function of the ANXA2⁺ subpopulation of hepatocytes detected in patient samples from both APAP and non-A-E patient samples. ANXA2⁺ hepatocytes have a migratory phenotype based on Gene ontology characteristics and morphology. The authors note that these ANXA2⁺ hepatocytes are located in the peri-necrotic region in the APAP-ALF livers but are not as localized in the non-A-E ALF livers. The authors then proceed to demonstrate that wound closure precedes hepatocyte proliferation in mice, and that motile hepatocytes at the edge of the necrotic area show collective migration of hepatocyte sheets to effect wound closure. They further show, using knockdown of ANXA2⁺ hepatocytes in cell models and in vivo in the APAP mouse model, that depletion of hepatocyte ANXA2 expression reduces HGF induced human and mouse hepatocyte migration in vitro and abrogates necrotic wound closure following APA-induced mouse liver injury. They also provide data that the ANXA2⁺ hepatocytes effect wound closure preceded the onset of hepatocyte proliferation. The mechanism by which these ANXA2⁺ hepatocytes effect wound closure has not been elucidated in the current study. While this part of the manuscript is a nice extension of previous work, showing how peri-wound hepatocytes repair the pericentral zone, it is missing depth based on more accurate lineage tracing models and detailed characterization of ANXA2⁺ hepatocytes as well as the role of this protein during liver regeneration. However, the manuscript is very well written with beautiful illustrations and provides a valuable resource for future research in an important medical field.

Major points:

The finding that peri-wound hepatocytes promote liver regeneration following loss of zone3 by moving to the pericentral niche and acquiring a zone3 signature is not new. First, the Nusse lab

showed that peri-wound hepatocytes upregulate AXIN2 and replenish the zone3 within days (PMID: 30762896, not cited). In addition, it was shown already that the GS+ hepatocytes in the regenerated zone3 did not, for the most part (~75%), proliferate (PMID: 31866224, reference 12), using a very similar experimental setting (EdU vs. BrdU labeling and DTA-mediated vs APAP-mediated zone3 hepatocyte ablation, respectively). Another study (PMID: 35659879, reference 19) extended this work by profiling the peri-wound hepatocytes in more detail, revealing their transient reprogramming during regeneration. Moreover, it was previously shown that wound closure and auxiliary hepatocyte proliferation in other zones precedes proliferation in zone3 (PMID: 31866224, reference 12). Together, these studies already established the concept that peri-injury hepatocytes move towards the central vein and acquire GS expression (zone3 identity). To add novelty to this manuscript beyond the data resource, the authors should characterize ANXA2+ hepatocytes and the role of ANXA2 in liver regeneration in more detail and discuss their findings in relation to the previous work. Specific points are listed below:

- 1) The AAV-based tracing system the authors used is suitable to exclude contribution of BEC-to-hepatocyte transdifferentiation to the repair process but does not allow to conclude which hepatocytes are new, whether they derived from ANXA2+ hepatocytes or not, and where these hepatocytes migrate. Lineage tracing should be performed using an ANXA2-CreERT2 model with a confetti reporter, instead of using pan-lobular unspecific labeling with an AAV-based system.
- 2) The roles of ANXA2 and ANXA2+ hepatocytes remain unclear. Could ANXA2 simply be required for phagocytosis of hepatocytes? This would fit the ANXA2 staining in the injury area with HNF4a-negative/presumably dead hepatocytes, as well as the abnormal morphology of ANXA2+ hepatocytes. This could further explain the delayed wound healing process following ANXA2-downregulation using siRNAs. While the intravital imaging resolution is impressive, the “active” migration of hepatocytes cannot be seen clearly. It seems that the whole hepatocyte front is moving towards the central vein, possibly pushed by the auxiliary proliferation in other zones. More mechanistic insights into how ANXA2 regulates migration of hepatocytes would be important. Also, more detailed analysis of the impaired wound healing process in absence of ANXA2 is important.
- 3) What are the signals that may be coming from the T cells, macrophages or other cells in the wound that stimulate hepatocyte migration mediated by ANXA2?
- 4) What are the other genes defining the ANXA2+ cluster? Usually, clusters are defined by multiple genes that are differentially expressed in comparison to the complement clusters. Why did the authors pick ANXA2 for follow up?
- 5) When looking at the data in the shiny-app, the ANXA2+ ST spots do not seem to be restricted to peri-wound sites but are present across the lobes of patient livers. They also do not cluster in the UMAP plots along zone3 markers. In fact, the strongest expression seems to be within the wound itself. This should be clarified.

Minor points:

5. page 6, line 245-246: Hepatocyte proliferation had returned to baseline levels by day 14, with complete wound closure (Extended Data Fig. 5a). This data seems missing in the Extended Data Fig.5). In addition, this Figure shows peak proliferation at the same time as peak necrosis. This is confusing given that the authors suggest that wound closure precedes proliferation.

6. While it is reasonable that more diffuse necrosis (i.e. in the non-A-E hepatitis) utilizes the same mechanism of wound closure as in the ALF-APAP, it is also possible that this mechanism is not successful or the principal mechanism for resolution of necrosis when the necrosis is more diffuse/severe (i.e. other regenerative mechanisms may be more important). This should be discussed as a potential limitation of the current study.

7. While it is acknowledged that obtaining truly “healthy” liver may not be feasible, patients who have undergone surgical resection for solitary colorectal metastases may not be truly healthy. This caveat should be included in the discussion.

8. It would be great if the shiny-app could be improved: a) Gene selection should ideally be maintained when switching between samples. b) In addition, the value of the app would be increased if comparisons could be made by automatically generating violin plots (or other graphs) after selecting healthy and diseased human samples and genes.

Referee #2 (Remarks to the Author):

In this study, Matchett et al aim to shed further lights on the mechanisms of liver regeneration upon Acute Liver Failure (ALF).

For this purpose, they take an integrative multimodal (mainly snRAN combined with spatial profiling, with additional 4D intravital imaging) approach and initially compare two ALF groups of patients named as Acetaminophen-induced (APAP-ALF), and nonA-E hepatitis (NAE-ALF) with each other and additionally with a third control healthy group (HEA).

By deconstructing human liver regeneration, they report emergence of a new ANXA2+ migratory hepatocyte subpopulation during human liver regeneration. This observation is accompanied by identification of a corollary migratory subpopulation in an acetaminophen (APAP-)induced mouse model, in which they manage to monitor the necrotic wound closure and hepatocyte proliferation across various time points . This leads them to the observation that wound closure precedes hepatocyte proliferation. They further show that in vitro depletion of ANXA2 expression results in abolishing of AGF-induced in both humans and mice and abrogates necrotic wound closure in APAP-mouse.

This is a comprehensive and data-rich study, exemplifying an interesting systems immunology data-driven approach, leading to immunological hypotheses with potentially great medical implications in therapies and generative medicine. They also provide a link to an interactive shiny app that helps basic interrogation of their compressive multimodal data.

There is obviously a lot to like about this study,

Major concerns:

- One of the key objectives of this study is to investigate which human liver disease exhibit a substantial hepatocyte proliferative response. Proliferative arrest has been previously reported as aging hallmark in hepatic compartments. The three groups they have used are significantly different by age distribution, 41, 53, and 64 for APAP, NAE and HEA respectively. Yet, I do not see any effort in this study to explain what the effect of this cross-groups aging is in their results/observations?
- Similarly, HEA is dominated by Female individuals and whereas APAP and NAE are mostly Male patients! Dose this gender imbalance between the control and the two disease categories will have any confounding effect on the analyses?
- Here, Healthy control is referred to 'non-lesional liver' from patients of with solitary colorectal metastasis. I personally wouldn't call this as Health, but rather the 'Control' group with clear definition in where it is first introduced. If I assume that this is kind of personal tase, still I would be keen that what the impact of medicine or drugs that these patients have been taking on their analysis.
- The current view on snRNA compared to scRNA is that the former is more sensitive and more powerful in identification of rare subpopulations. If this is a key factor in identification of ANXA2 population, ie, can they track this subpopulation other liver data from scRNA technologies?

Minor concerns

- In abstract, the abbreviation 'HGF-induced' is used before definition, which is difficult to understand.
- Line 168, Sfig1i, has too many colors, difficult to track any pattern, in particular for color-blinded readers! Wouldn't be better to collapse this at least to the three categories of HEA, NAE and APAP?
- FigSF1f, I had difficulties spotting cluster 17, not sure if it is the print I have, or needs to be addressed!
- Lines 582 to 586, they refer to four datasets, but the numeric '4)' is missing.
- Line 619, the hepatocyte migration module is defined using top 25 genes. How robust this definition is to top N genes?
- Line 780, How do you know that what you see here is not an effect of ageing?
- Line 800, Perhaps I would remove Fig2j into Supplementary, as I don't see this as informative because using this module score, technically very similar genes will be detected in mouse!
- Line 827, the Figure title and the legend title differs! Better to be the same!
- Line 856, as to my previous comment, inconsistent figure and legend title. I think there are a couple more that I wount be repeating.
- Line 899, I know you have this strategy all along the figure, but for SFig1e, I was keen to find out how 'Hi' is your Hi in the colorbar?
- Line 906, as to one of my previous comments, again I feel too many colors with SF1i, that the whole thing becomes uninformative if not useless.

Hashem Koohy

Referee #3 (Remarks to the Author):

The Manuscript- multimodal decoding of human liver- is well written and a valuable and timely resource for the regeneration/liver community.

The authors do not over interpret data and provide functional data to back up any claims with acknowledgment of any future caveats/future work.

The application of cross-species discovery and direct intervention significantly improves the application of this work for the field. Moreover, the multi-modal aspect of pan lineage paired snRNA/ST/smFISH and functional 4D functional intravital imaging should be applauded.

The data sets generated are of wide applications to many in the field and as such the author (correctly) generated an open-access, interactive portal to allow the community to assess and visualise gene expression in multiple lineages in healthy/APA-ALF and NAE-ALF ST datasets.

The timing of closure/proliferation and uncoupling of wound healing partially driven by ANXA2 that this study reveals to rapidly repair damage before the integrity of the liver/gut barrier induces unrepairable damage is vital for future regenerative medicine application.

With some further work this MS is ready for publication.

Q for fig 1a where the author shows that in APAP-ALF and NAE-ALF they see a marked increase in hepatocytes proliferation can the author – at a high level is fine- describe (if known) what other cells types are activated in the other insults tested? I am aware this could be used for other future manuscripts focusing on other insult types and cell types but would help highlight the hepatocytes specificity of this work while other insults focus in diff cell type/repair mechanisms

Q Can the authors co-stain samples with actin/phalloidin markers to reveal lamellipodia/ruffles in more detail during closure etc throughout MS?

Q similarly above the MS focuses on collective migration and as such it would be good to see via IF or an appropriate staining in any cell-cell contact properties change or are maintain during closure ? simple E-cadherin etc could work here. On this note when the author KD ANXA2 and reduce closure can the author co-stain in this setting for actin/lamella/collective migration marker deregulation to help reveal what ANAX2 is doing here? ie is ruffling of migratory phenotypes reduced or is cell-cell contact changedetc etc..cell polarity cell shape -?

Q in intravital movieswhile collective movement is evident...can the authors score or discuss single cell migration also ? many movies have leader cell movement this should be noted and discussed ...especially in the context of polarity persistence etc..

In fig 2 b the pattern of contribution for cycling while smaller is similar to ANXA2..did authors assess if this was not spatially regulated at necrotic wound like anax2 ? if so this can be discussed to show diff repair options..

Referee #4 (Remarks to the Author):

This is a very elegant study from a well-known group of researchers.

1. The exceptional value of this study is the analysis of a (fairly large) number of liver samples subjected to snRNA-seq from human patients with acute liver failure (and controls). The challenge is, though, to understand liver regeneration from samples that are exclusively obtained from individuals with a failure of recovery (otherwise, a transplant would not been needed). What is really lacking is a group of patients that were biopsied during acute liver injury (ideally matching the peak ALT levels of the APAP-ALF / NAE-ALF cohorts), but then recovered completely (and spontaneously, i.e. without specific treatment such as steroids for AIH or antivirals for HBV).

2. The “hepatocyte proliferation” data in Fig.1 are a bit difficult to interpret. The samples are all from explants – and ALF patients have more Ki-67+ hepatocytes. But apparently, this proliferative activity does not lead to functional replacement of hepatocytes. Does this mean that Ki-67+ cannot be used as a surrogate for proliferation? Or that the proliferation is not high enough? 3. How specific is the existence of ANXA2+ hepatocytes to ALF? Would these (ANXA2+ migratory) hepatocytes also be present in chronic human liver diseases (e.g. NASH) and/or in liver cirrhosis? – From a clinical point of view, this would be highly relevant, as the loss of such a motile hepatocyte population could explain the irreversibility of certain chronic liver disease conditions (or their reduced regenerative potential).

4. The authors did not comment on a recent paper by Tom Bird’s group (May S, et al. J Hepatol. 2023, PMID: 36702176), which could be very relevant for the current study. In this work, the authors find no evidence of predominant expansion of the pericentral hepatocyte population during liver homeostatic regeneration (unlike prior work), and raised some general concerns about peculiarities of mouse models. Moreover, they observed marked differences between male and female mice. To me, this mandates to study liver regeneration in mice of both sexes in order to validate the key observations obtained solely with male mice (as, unfortunately, many studies have done in the past).

5. As a cellular source during liver regeneration, cholangiocyte-to-hepatocyte transition has been described / suggested. What is the evidence in favor or against this in ALF? Do some of the hepatocyte populations (and if so, maybe exclusively in the periportal regions?) repopulation from cholangiocytes / ductular progenitor cells?

6. What is the role of stellate cells in ‘wound closure’ after ALF? The activation of hepatic stellate cells in ALF has been described quite some time ago in human liver biopsies (Dechêne A, et al. Hepatology. 2010; 52(3):1008-16. PMID: 20684020). The authors now propose the uncoupling of wound closure and hepatocyte proliferation, but this may simply represent the task diversification between two populations and this may still be highly orchestrated. From the snRNA-seq data, can the authors comment on HSC-hepatocyte interactions (in a subset-, spatial- and time-resolved manner)?

7. The tissue samples originate from various sources / centers. I am sure the authors have ensured quality of the samples and the validity of the transcriptomic data. What is currently missing, but would tremendously add to the translational value, is a correlation between “molecular findings” and the clinical phenotype (etiology, ALT levels, liver function, hyperacute vs. acute vs. subacute ALF, grade of HE etc).

8. From a conceptual point of view, I would find it important to demonstrate that the hepatocyte migration as a mechanism of wound closure (mouse experiments in Fig. 3) is not specific to APAP-injury, but would also occur in different types of liver injury, e.g., ischemia-reperfusion-injury. Validating this principal finding in a second (independent) ALF model would be very valuable.

Author Rebuttals to Initial Comments:

- Further detail the role of ANXA2+ hepatocytes, and ANXA2 in repair (Reviewer #1)
- Include additional characterization of the wound healing process in the absence of ANXA2 (Reviewer #1)
- Examine the roles of other populations (Reviewers #1, #3, #4)
- Address concerns regarding controls used (Reviewers #1, #2, #4)
- Account for age disparities, sex differences between groups studied as well as drug regimens in controls (Reviewer #2)
- Identify ANXA2+ populations in other datasets (Reviewers #2, #4)
- Include additional detailed characterization of migration (Reviewers #1, #3)
- Assess the possible role of transdifferentiation (Reviewer #4)
- Relate findings to clinical presentation (Reviewer #4)

We thank the reviewers for their very helpful and thoughtful comments, and we feel that the new data we have generated in response to their comments has very much enhanced our manuscript.

Referees' comments:

Referee #1 (Remarks to the Author):

Authors used a combination of snRNAseq, ST, smFISH and IF to identify gene modules differentially expressed across the lobule in the APAP-ALF and non-A-E human liver samples. Using these approaches, the authors demonstrated loss of hepatocyte portal-central zonation and emergence of hepatocytes with both portal and central characteristics. These data emphasize the hepatocyte plasticity in response to loss of pericentral hepatocytes. These data are impactful as they are obtained from patient samples and provide a robust data set that can serve as a valuable resource for new target and pathway investigations. A potential limitation is that because these samples were obtained at the time of liver transplantation, they represent the most severe cases and are not able to assess the pattern of resolution.

The second portion of the manuscript focuses on mechanistic studies to assess the function of the ANXA2+ subpopulation of hepatocytes detected in patient samples from both APAP and non-A-E patient samples. ANXA2+ hepatocytes have a migratory phenotype based on Gene ontology characteristics and morphology. The authors note that these ANXA2+ hepatocytes are located in the peri-necrotic region in the APAP-ALF livers but are not as localized in the non-A-E ALF livers. The authors then proceed to demonstrate that wound closure precedes hepatocyte proliferation in mice, and that motile hepatocytes at the edge of the necrotic area show collective migration of hepatocyte sheets to effect wound closure. They further show, using knockdown of ANXA2+ hepatocytes in cell models and in vivo in the APAP mouse model, that depletion of hepatocyte ANXA2 expression reduces HGF induced human and mouse hepatocyte migration in vitro and abrogates necrotic wound closure following APA-induced mouse liver injury. They also provide data that the ANXA2+ hepatocytes effect wound closure preceded the onset of hepatocyte proliferation. The mechanism by which these ANXA2+ hepatocytes effect wound closure has not been elucidated in the current study. While this part of the manuscript is a nice extension of previous work, showing how peri-wound hepatocytes repair the pericentral zone, it is

missing depth based on more accurate lineage tracing models and detailed characterization of ANXA2+ hepatocytes as well as the role of this protein during liver regeneration. However, the manuscript is very well written with beautiful illustrations and provides a valuable resource for future research in an important medical field.

Major points:

We thank the reviewer for their helpful comments regarding our manuscript. Given the multifaceted nature of the following comment, we have partitioned our responses below:

The finding that peri-wound hepatocytes promote liver regeneration following loss of zone3 by moving to the pericentral niche and acquiring a zone3 signature is not new. First, the Nusse lab showed that peri-wound hepatocytes upregulate AXIN2 and replenish the zone3 within days (PMID: 30762896, not cited).

While PMID: 30762896 (which we have now cited in the manuscript) provided important new information regarding our understanding of liver regeneration, we do not agree with the statement that this manuscript established the concepts that we present in our study.

Firstly, we have uncovered a novel migratory hepatocyte subpopulation which mediates wound closure following liver injury, and secondly we have uncovered a temporal disconnect between wound closure (as assessed by percentage necrotic area) and hepatocyte proliferation, demonstrating that wound closure precedes hepatocyte proliferation.

In PMID: 30762896 the authors show in Fig. 3 that 'Peri-injury Axin2+ hepatocytes proliferate to repair damaged tissue' (Figure 3 caption), then proceed to explore the role of Wnt signalling in liver repair. This manuscript focusses on hepatocyte proliferation as the main driver of hepatic wound repair with the authors concluding that:

'Our findings show that following a localized injury (i.e., CCl₄), only select hepatocytes, specifically those closest to the damaged tissue, are more likely to become activated and undergo proliferation, while hepatocytes farther from the injury site remain quiescent'.

In contrast we use a comprehensive time course to investigate the dynamics of liver regeneration following APAP-induced mouse liver injury, uncovering a temporal disconnect between wound closure (as assessed by percentage necrotic area) and hepatocyte proliferation. In male mice peak hepatocyte necrosis occurred at 30h post APAP-induced liver injury (22.3%±1.3%SEM), with percentage necrotic area decreasing by 30.9% at 42h (15.4%±1.4%SEM) and by 58.3% at 48h (9.3%±1%SEM) (Fig. 3a). In female mice following APAP-induced acute liver injury, hepatocyte necrosis peaked at 36h (22.2%±2.6%SEM), with percentage necrotic area decreasing by 51.8% at 42h (10.7%±1.0%SEM) (Extended Data Fig. 8b). In both sexes wound closure preceded the onset of hepatocyte proliferation.

In addition, it was shown already that the GS+ hepatocytes in the regenerated zone3 did not, for the most part (~75%), proliferate (PMID: 31866224, reference 12), using a very similar experimental setting (EdU vs. BrdU labeling and DTA-mediated vs APAP-mediated zone3 hepatocyte ablation, respectively).

Regarding the data presented in PMID: 31866224, we appreciate the similarities in experimental approach. However, our utilisation of this experiment was to assess whether hepatocyte repopulation of the necrotic area immediately adjacent to the central vein is driven by hepatocyte proliferation following APAP-induced liver injury. Our results demonstrate that in male mice 75.6% ($\pm 9.4\%$ SEM) of hepatocytes in the area immediately adjacent to the central vein did not arise from hepatocyte proliferation following APAP-induced acute liver injury (Fig. 3c). In female mice 86.9% ($\pm 2.3\%$ SEM) of GS+ hepatocytes were BrdU-negative at 14 days post APAP-induced acute liver injury (Extended Data Fig 8d).

Given the differences in experimental models we felt it important to conduct this experiment specifically in our injury setting (APAP-induced acute liver injury), as opposed to pericentral hepatocyte ablation via DTA-mediated ablation of AXIN2+ pericentral hepatocytes.

Another study (PMID: 35659879, reference 19) extended this work by profiling the peri-wound hepatocytes in more detail, revealing their transient reprogramming during regeneration.

PMID: 35659879 shows transient reprogramming of peri-injury hepatocytes during regeneration, finding an onco-fetal gene expression program at 48h post-APAP injury. This study also proposes hepatocyte proliferation to be the major driver of hepatocyte repopulation of zone 3 following APAP-induced liver injury, concluding that:

‘We find that hepatocytes proliferate throughout the liver lobule, creating the mitotic pressure required to repopulate the necrotic pericentral zone rapidly.’

In contrast, we uncovered a temporal disconnect between centrilobular wound closure and hepatocyte proliferation, demonstrating that wound closure precedes hepatocyte proliferation following APAP-induced mouse liver injury (Fig. 3a, Ext. Data Fig. 3b). Furthermore, continual administration of BrdU to label all hepatocytes following APAP-induced mouse liver injury in male mice demonstrated that 75.6% ($\pm 9.4\%$ SEM) of hepatocytes in the area immediately adjacent to the central vein did not arise from hepatocyte proliferation (Fig. 3c). In female mice 86.9% ($\pm 2.3\%$ SEM) of GS+ hepatocytes were BrdU-negative at 14 days post APAP-induced acute liver injury (Extended Data Fig 8d). Furthermore, 4-D intravital imaging of APAP-induced mouse liver injury identified motile hepatocytes, displaying membrane ruffling and the formation of dynamic protrusions at the leading edge, with collective cell migration of the hepatocyte sheet to effect wound closure.

Collectively our data show that mechanisms and processes other than hepatocyte proliferation are critical to effect rapid and successful wound closure following liver injury.

Moreover, it was previously shown that wound closure and auxiliary hepatocyte proliferation in other zones precedes proliferation in zone3 (PMID: 31866224, reference 12).

PMID: 31866224 demonstrates differences in proliferation across the liver zones following allyl alcohol-induced injury, partial hepatectomy and DTA-induced ablation of peri-central hepatocytes. However, this manuscript does not quantify wound closure temporal dynamics with sufficient time points to investigate whether wound closure precedes hepatocyte proliferation.

In contrast, one of our novel findings is a temporal disconnect between wound closure (as assessed by percentage necrotic area) and hepatocyte proliferation following APAP-induced acute liver injury in both male and female mice (Fig. 3a, Extended Data Fig. 8b). In both sexes wound closure preceded the onset of hepatocyte proliferation.

Together, these studies already established the concept that peri-injury hepatocytes move towards the central vein and acquire GS expression (zone3 identity). To add novelty to this manuscript beyond the data resource, the authors should characterize ANXA2+ hepatocytes and the role of ANXA2 in liver regeneration in more detail and discuss their findings in relation to the previous work.

We respectfully disagree with the comment that “Together, these studies already established the concept that peri-injury hepatocytes move towards the central vein and acquire GS expression (zone3 identity).”

As mentioned above, recent work (PMID: 35659879) has proposed that ‘mitotic pressure’ mediated by hepatocyte proliferation in zone 1 and 2 drives repopulation of zone 3 (pericentral hepatocytes). Other recent work (PMID: 31866224) states that ‘compensatory proliferation from all liver zones’ drives repopulation of peri-central hepatocytes. We disagree that the concept of active hepatocyte migration has already been established, either by these studies or in the field.

Our work dissects unanticipated aspects of liver regeneration, demonstrating an uncoupling of wound closure and hepatocyte proliferation and uncovering a novel migratory hepatocyte subpopulation which mediates wound closure following liver injury. Furthermore, 4-D intravital imaging of APAP-induced mouse liver injury identified motile hepatocytes for the first time, displaying membrane ruffling and the formation of dynamic protrusions at the leading edge, with collective cell migration of the hepatocyte sheet to effect wound closure.

Specific points are listed below:

- 1) The AAV-based tracing system the authors used is suitable to exclude contribution of BEC-to-hepatocyte transdifferentiation to the repair process but does not allow to conclude which hepatocytes are new, whether they derived from ANXA2+ hepatocytes or not, and where these hepatocytes migrate. Lineage tracing should be performed using an ANXA2-CreERT2 model with a confetti reporter, instead of using pan-lobular unspecific labeling with an AAV-based system.

We thank the reviewer for this comment, however we are unable to mark an ANXA2-positive hepatocyte subpopulation at baseline due to the absence of hepatocyte ANXA2 expression in healthy liver tissue (Fig. 2m). Therefore an ANXA2-CreERT2 approach would not allow us to lineage trace ANXA2-positive hepatocytes as we would not be able to incorporate a 3-week tamoxifen washout period.

2) The roles of ANXA2 and ANXA2⁺ hepatocytes remain unclear. Could ANXA2 simply be required for phagocytosis of hepatocytes? This would fit the ANXA2 staining in the injury area with HNF4a-negative/presumably dead hepatocytes, as well as the abnormal morphology of ANXA2⁺ hepatocytes. This could further explain the delayed wound healing process following ANXA2-downregulation using siRNAs.

We thank the reviewer for this helpful comment. Regarding ANXA2 staining in the injury area: snRNAseq of human and mouse liver during APAP-induced injury demonstrated that multiple non-hepatocyte lineages (endothelia, leucocytes and mesenchymal cells) express ANXA2 mRNA (Extended Data Fig. 3d; Extended Data Fig. 6f). Furthermore, multiplex immunofluorescence confirmed that ANXA2 is expressed on subsets of leucocytes, mesenchyme, endothelia, and cholangiocytes (Extended Data Fig. 3e; Extended Data Fig. 6g-i).

Regarding the potential role of hepatocyte ANXA2 in mediating phagocytosis of hepatocytes we performed an *in vitro* phagocytosis assay. Primary mouse hepatocytes were treated with Scrmb-siRNA (control) or Anxa2-siRNA for 48hrs. Hepatocyte death was induced with APAP and the necrotic hepatocytes were co-incubated with bone marrow-derived macrophages (BMDMs). Phagocytosis of necrotic Scrmb-siRNA (control) and Anxa2-siRNA treated hepatocytes by BMDMs was similar between the two groups (Extended Data Fig. 9f).

While the intravital imaging resolution is impressive, the “active” migration of hepatocytes cannot be seen clearly. It seems that the whole hepatocyte front is moving towards the central vein, possibly pushed by the auxiliary proliferation in other zones. More mechanistic insights into how ANXA2 regulates migration of hepatocytes would be important. Also, more detailed analysis of the impaired wound healing process in absence of ANXA2 is important.

We agree that our data show collective migration of the hepatocyte front to effect wound closure. 4-D intravital imaging of APAP induced mouse liver injury identified motile hepatocytes, displaying membrane ruffling and the formation of dynamic protrusions at the leading edge, with collective cell migration of the hepatocyte sheet to effect wound closure (Fig. 3e-j, SI Videos 6-13). Movement of the hepatocyte front is not mediated by auxiliary hepatocyte proliferation in other zones as we observed a lack of hepatocyte proliferation before 48hr post APAP-induced acute liver injury (Fig. 3a, Extended Data Fig. 8b). Furthermore, the IVM imaging (which showed collective migration of the hepatocyte sheet) was performed between 36 hours and 42 hours post APAP-induced acute liver injury (Fig. 3d), which is before hepatocyte proliferation occurs.

Expression of zonula occludens-1 (ZO-1), a tight junction protein mediating cell-cell contact and cell polarity, was similar in portal and central-associated hepatocytes in uninjured liver (Extended Data Fig. 7d). ANXA2⁺ hepatocytes in the peri-necrotic region (PNR) expressed ZO-

1 following APAP-induced acute liver injury (Extended Data Fig. 7e), and ZO-1 expression in the PNR did not change following APAP-induced acute liver injury, demonstrating maintenance of epithelial sheet connections and hepatocyte polarity during wound closure (Extended Data Fig. 7f). ANXA2-positive hepatocytes were less circular than ANXA2-negative hepatocytes in the PNR (Extended Data Fig. 7c). We also observed membrane F-actin staining in lamellipodia of ANXA2⁺ hepatocytes in human APAP-ALF (Extended Data Fig. 3j) and in ANXA2⁺ hepatocytes post APAP-induced acute liver injury in mice (Extended Data Fig. 6k). Epithelial sheet connections and hepatocyte polarity were similar between AAV8-shRNA-Scrbp and AAV8-shRNA-Anxa2 treated mice following APAP-induced acute liver injury (Extended Data Fig. 9e). However, hepatocytes in the PNR of AAV8-shRNA-Anxa2-treated mice lacked membrane F-actin and were more circular than those in AAV8-shRNA-Scrbp-treated mice, demonstrating a reduced migratory phenotype in ANXA2 negative hepatocytes (Extended Data Fig. 9c,d).

These data demonstrate that hepatocyte sheet migration is a major mechanism of wound closure following APAP-induced liver injury, and we have included a schematic summarising our proposed model of liver regeneration following APAP-induced liver injury (Fig. 4l).

3) What are the signals that may be coming from the T cells, macrophages or other cells in the wound that stimulate hepatocyte migration mediated by ANXA2?

To investigate potential ligands interacting with receptors expressed by the migratory hepatocyte subpopulation, we applied interactome analysis using the CellChat R package to the human APAP-ALF dataset. We found mesenchymal and cholangiocyte subpopulations to be the dominant interacting partners with migratory hepatocytes (Extended Data Fig. 5a). Multiplex smFISH demonstrated co-location of migratory hepatocytes with myofibroblasts (mesenchyme cluster 1), hepatic stellate cells (mesenchyme cluster 2) and cholangiocytes in the PNR (Extended Data Fig. 5b). CellChat analysis highlighted multiple interactions related to the transforming growth factor beta (TGF β) signalling pathway (Extended Data Fig. 5c) which has previously been shown to be an important regulator of epithelial cell plasticity and migration (PMID: 33756119). These interactions and pathways were further demonstrated in mouse APAP-induced acute liver injury (Extended Data Fig. 7g,h).

4) What are the other genes defining the ANXA2⁺ cluster? Usually, clusters are defined by multiple genes that are differentially expressed in comparison to the complement clusters. Why did the authors pick ANXA2 for follow up?

Further genes defining the ANXA2⁺ cluster are listed in the differential gene expression results (SI Table 3). A multiplex smFISH panel incorporating multiple differential genes expressed by the migratory hepatocyte cluster was used to delineate this population in tissue (Fig. 2d; Extended Data Fig. 3i; SI Table 2).

ANXA2 was chosen for follow up for the following reasons: 1) ANXA2 expression is enriched in the migratory hepatocyte cluster in both the human and mouse hepatocyte datasets (Extended Data Fig. 3c, SI Video 2); 2) Previous studies have demonstrated a role for ANXA2 in epithelial cancer cell migration (PMID: 23558678, PMID 28634253, PMID 20079732) which suggested to us that ANXA2 may be important in hepatocyte migration.

5) When looking at the data in the shiny-app, the ANXA2⁺ ST spots do not seem to be restricted to peri-wound sites but are present across the lobes of patient livers. They also do not cluster in the UMAP plots along zone3 markers. In fact, the strongest expression seems to be within the wound itself. This should be clarified.

The spatial transcriptomic data was generated using 10X Visium technology, where mRNA is captured on 55um spots set 100um apart; therefore, each spot is comprised of transcriptomic data from more than one cell type. The shiny application displays the raw gene expression, whereas the manuscript plots are generated using quantile thresholding to reduce background expression, allowing clearer visualisation of ST spots that express high levels of ANXA2; we have now stated this in our methods.

Furthermore, we show that ANXA2 expression is enriched in the migratory hepatocyte subpopulation and demonstrate (using immunofluorescence staining) that ANXA2⁺ hepatocytes are enriched in the PNR (Fig. 2d-f). We also show that ANXA2 is present on other cell types including endothelia, leucocytes, and mesenchyme in both humans and mice following APAP-induced liver injury (Extended Data Fig. 3d,e. Extended Data Fig. 6f-i).

The ANXA2⁺ migratory hepatocyte subpopulation is distinct enough from zone 3 hepatocytes that they do not cluster together using non-spatially resolved snRNA-seq data. Harnessing both immunofluorescence staining and spatial transcriptomics to determine the location of this population we demonstrate that ANXA2⁺ migratory hepatocytes reside in the peri-necrotic region in human and mouse APAP-induced acute liver injury (Fig. 2e,f,k,n, Extended Data Fig. 3g,i, Extended Data Fig. 7b). Furthermore, we observed ANXA2⁺ migratory hepatocytes in multiple aetiologies of human chronic liver disease and mouse models of liver injury, demonstrating that this hepatocyte phenotype occurs in response to both peri-portal and peri-central liver injury (Extended Data Fig. 5g-i).

Minor points:

5. page 6, line 245-246: Hepatocyte proliferation had returned to baseline levels by day 14, with complete wound closure (Extended Data Fig. 5a). This data seems missing in the Extended Data Fig.5). In addition, this Figure shows peak proliferation at the same time as peak necrosis. This is confusing given that the authors suggest that wound closure precedes proliferation.

We thank the reviewer for this helpful comment and apologise for any confusion caused. This hepatocyte proliferation and wound closure data is now referred to on line 316-318, and is presented in Extended Data Fig. 8c. The intention of this figure is to demonstrate that hepatocyte proliferation has returned to baseline levels by day 14, with complete wound closure. We have now included data from extra time points on these graphs which, alongside showing return to baseline of hepatocyte proliferation and necrotic area, highlight the temporal disconnect between hepatocyte proliferation and wound closure.

6. While it is reasonable that more diffuse necrosis (i.e. in the non-A-E hepatitis) utilizes the same mechanism of wound closure as in the ALF-APAP, it is also possible that this mechanism is not successful or the principal mechanism for resolution of necrosis when the necrosis is more diffuse/severe (i.e. other regenerative mechanisms may be more important). This should be discussed as a potential limitation of the current study.

We thank the reviewer for this helpful comment, and we have amended the text in the discussion section to highlight this point.

7. While it is acknowledged that obtaining truly “healthy” liver may not be feasible, patients who have undergone surgical resection for solitary colorectal metastases may not be truly healthy. This caveat should be included in the discussion.

We have added text in the results section clarifying the origin of our human liver healthy control tissue to address this caveat.

8. It would be great if the shiny-app could be improved: a) Gene selection should ideally be maintained when switching between samples. b) In addition, the value of the app would be increased if comparisons could be made by automatically generating violin plots (or other graphs) after selecting healthy and diseased human samples and genes.

We thank the reviewer for this helpful comment. The web application now maintains gene selection between samples and generates a violin plot grouped by cluster and split by condition to allow users to make the comparisons suggested by the reviewer.

Referee #2 (Remarks to the Author):

In this study, Matchett et al aim to shed further lights on the mechanisms of liver regeneration upon Acute Liver Failure (ALF).

For this purpose, they take an integrative multimodal (mainly snRAN combined with spatial profiling, with additional 4D intravital imaging) approach and initially compare two ALF groups of patients named as Acetaminophen-induced (APAP-ALF), and nonA-E hepatitis (NAE-ALF) with each other and additionally with a third control healthy group (HEA).

By deconstructing human liver regeneration, they report emergence of a new ANXA2+ migratory hepatocyte subpopulation during human liver regeneration. This observation is accompanied by identification of a corollary migratory subpopulation in an acetaminophen (APAP-)induced mouse model, in which they manage to monitor the necrotic wound closure and hepatocyte proliferation across various time points . This leads them to the observation that wound closure precedes hepatocyte proliferation. They further show that in vitro depletion of ANXA2 expression results in abolishing of AGF-induced in both humans and mice and abrogates necrotic wound closure in APAP-mouse.

This is a comprehensive and data-rich study, exemplifying an interesting systems immunology data-driven approach, leading to immunological hypotheses with potentially great medical implications in therapies and generative medicine. They also provide a link to an interactive shiny app that helps basic interrogation of their compressive multimodal data. There is obviously a lot to like about this study,

Major concerns:

- One of the key objectives of this study is to investigate which human liver disease exhibit a substantial hepatocyte proliferative response. Proliferative arrest has been previously reported as aging hallmark in hepatic compartments. The three groups they have used are significantly different by age distribution, 41, 53, and 64 for APAP, NAE and HEA respectively. Yet, I do not see any effort in this study to explain what the effect of this cross-groups aging is in their results/observations?

We thank the reviewer for this helpful comment regarding age distribution in the three groups (healthy, APAP-ALF and NAE-ALF). The mean age of each group is an accurate representation of the demographics of these ALF cohorts, as APAP-ALF occurs in a younger population than NAE-ALF. Generating Pearson correlations between patient age and the harmony components of the hepatocyte dataset shows that age did not influence variability in the snRNA-seq data (Extended Data Fig. 4a). We also examined hepatocyte proliferation versus age within each ALF aetiology (APAP-ALF and NAE-ALF) in our immunofluorescence staining data and found no correlation (Extended Data Fig. 1a).

- Similarly, HEA is dominated by Female individuals and whereas APAP and NAE are mostly Male patients! Does this gender imbalance between the control and the two disease categories will have any confounding effect on the analyses?

We thank the reviewer for this comment and acknowledge that there is a slight sex imbalance in some of the conditions. However, as shown in SI Table 1 our healthy/control samples contain more male individuals than female (7:3). APAP-ALF contains 4 male samples and 6 female samples, and NAE-ALF has 3 male patients and 9 female patients. Therefore, in our liver injury datasets the sex imbalance leans towards the female, not the male side.

Visualising the sex split on a UMAP and comparing per-sex contribution to the hepatocyte clusters (Extended Data Fig. 4b), we see no sex bias in any of our hepatocyte subpopulations. Generating point-biserial correlations between patient sex and the harmony components of the hepatocyte dataset shows that patient sex did not influence variability in the snRNA-seq data (Extended Data Fig. 4c).

- Here, Healthy control is referred to 'non-lesional liver' from patients of with solitary colorectal metastasis. I personally wouldn't call this as Health, but rather the 'Control' group with clear definition in where it is first introduced. If I assume that this is kind of personal tase, still I would be keen that what the impact of medicine or drugs that these patients have been taking on their analysis.

We have added text in the results section clarifying the origin of our human liver healthy control tissue (where it is first introduced) to address this. The healthy control tissue was sourced from the Biobank in Edinburgh, and although we were provided with basic clinical metadata, information regarding the medications patients had been taking was not available.

- The current view on snRNA compared to scRNA is that the former is more sensitive and more powerful in identification of rare subpopulations. If this is a key factor in identification

of ANXA2 population, ie, can they track this subpopulation other liver data from scRNA technologies?

We have used single nuclei RNA sequencing (snRNA-seq) in this study as it allowed us to generate high quality hepatocyte data from frozen tissue from biobanks around the world. We and others (PMID 34792289) have found that single-cell RNA-seq of hepatocytes results in very poor data. To our knowledge, no other group has generated scRNA-seq datasets from human acute liver failure samples.

Minor concerns

- In abstract, the abbreviation 'HGF-induced' is used before definition, which is difficult to understand.

This has been amended.

- Line 168, Sfig1i, has too many colors, difficult to track any pattern, in particular for color-blinded readers! Wouldn't be better to collapse this at least to the three categories of HEA, NAE and APAP?

We thank the reviewer for this helpful comment. In addition to visualising the contribution per sample to each cluster (Extended Data Fig. 1j), we have now included a figure showing the contribution of HEA, APAP and NAE samples to each cluster (Extended Data Fig. 1k).

- FigSF1f, I had difficulties spotting cluster 17, not sure if it is the print I have, or needs to be addressed!

This has been amended.

- Lines 582 to 586, they refer to four datasets, but the numeric '4' is missing.

This has been amended.

- Line 619, the hepatocyte migration module is defined using top 25 genes. How robust this definition is to top N genes?

When we vary the number of top N genes we see that the signatures produced in both the human and mouse datasets consistently identify the migratory hepatocyte population (Rebuttal Fig. 1a,b). Furthermore, training a scAnnotatR classifier on the human dataset and applying it to our mouse populations, we can see that the predicted migratory score again correctly identifies the migratory hepatocyte (Rebuttal Fig. 2). Therefore, varying the number of top N genes robustly identifies the migratory hepatocyte population.

Our approach of using a gene signature allows readers to probe their datasets for a similar population. 25 genes provides the best balance between complexity, accuracy, sensitivity, and specificity, after generating confusion matrices across a range of N between cells that belong to the migratory hepatocyte cluster and cells that are defined as migratory based on the thresholded migration signature score (Table 1).

- Line 780, How do you know that what you see here is not an effect of ageing?

We examined hepatocyte proliferation versus age within each ALF aetiology (APAP-ALF and NAE-ALF) in immunofluorescence staining data, and found no correlation (Extended Data Fig. 1a). Generating Pearson correlations between patient age and the harmony components of the hepatocyte dataset demonstrated that age did not influence variability in the snRNA-seq data (Extended Data Fig. 4a).

- Line 800, Perhaps I would remove Fig2j into Supplementary, as I don't see this as informative because using this module score, technically very similar genes will be detected in mouse! We have clarified in the results text that the GO terms in Fig. 2j were independently identified from the mouse ANXA2⁺ hepatocyte cluster, the marker genes of which were similar but not equivalent to the genes listed in the human migratory score.

- Line 827, the Figure title and the legend title differs! Better to be the same! We thank the reviewer for this helpful comment. This has been amended and all figure titles have been checked.

- Line 856, as to my previous comment, inconsistent figure and legend title. I think there are a couple more that I would be repeating. We thank the reviewer for highlighting this. This has been amended and all figure titles checked.

- Line 899, I know you have this strategy all along the figure, but for SFig1e, I was keen to find out how 'Hi' is your Hi in the colorbar? The numerical range for nFeatures and percent mito have been added to this figure.

- Line 906, as to one of my previous comments, again I feel too many colors with SF1i, that the whole thing becomes uninformative if not useless. We thank the reviewer for this helpful comment. As per our previous comment, in addition to visualising the contribution per sample to each cluster (Extended Data Fig. 1j) we have now included the contribution of HEA, APAP-ALF and NAE-ALF samples to each cluster (Extended Data Fig. 1k).

Hashem Koohy

Referee #3 (Remarks to the Author):

The Manuscript- multimodal decoding of human liver- is well written and a valuable and timely resource for the regeneration/liver community.

The authors do not over interpret data and provide functional data to back up any claims with acknowledgment of any future caveats/future work.

The application of cross-species discovery and direct intervention significantly improves the application of this work for the field. Moreover, the multi-modal aspect of pan lineage paired snRNA/ST/smFISH and functional 4D functional intravital imaging should be applauded.

The data sets generated are of wide applications to many in the field and as such the author (correctly) generated an open-access, interactive portal to allow the community to assess and visualise gene expression in multiple lineages in healthy/APA-ALF and NAE-ALF ST datasets.

The timing of closure/proliferation and uncoupling of wound healing partially driven by ANXA2 that this study reveals to rapidly repair damage before the integrity of the liver/gut barrier induces unreparable damage is vital for future regenerative medicine application.

With some further work this MS is ready for publication.

Q for fig 1a where the author shows that in APAP-ALF and NAE-ALF they see a marked increase in hepatocytes proliferation can the author – at a high level is fine- describe (if known) what other cells types are activated in the other insults tested? I am aware this could be used for other future manuscripts focusing on other insult types and cell types but would help highlight the hepatocytes specificity of this work while other insults focus in diff cell type/repair mechanisms

We have added additional text in the introduction section of the manuscript to highlight how recent single-cell human liver studies are largely focussed on chronic liver disease (which have emphasised non-parenchymal cell populations) rather than acute liver failure. This underlines the hepatocyte specificity of our work, compared to the recent published studies using single cell genomics to investigate mechanisms regulating chronic liver disease.

Q Can the authors co-stain samples with actin/phalloidin markers to reveal lamellipodia/ruffles in more detail during closure etc throughout MS?

We thank the reviewer for this helpful comment. We demonstrate membrane F-actin staining in lamellipodia of ANXA2⁺ hepatocytes in human APAP-ALF (Extended Data Fig. 3j) and in ANXA2⁺ hepatocytes post APAP-induced acute liver injury in mice (Extended Data Fig. 6k). Moreover, hepatocytes in the peri-necrotic region (PNR) of AAV8-shRNA-*Anxa2*-treated mice lacked membrane F-actin unlike AAV8-shRNA-*Scrm1* (control) mice, demonstrating a reduced migratory phenotype (Extended Data Fig. 9c).

Q similarly above the MS focuses on collective migration and as such it would be good to see via IF or an appropriate staining in any cell-cell contact properties change or are maintain during closure ? simple E-cadherin etc could work here. On this note when the author KD ANXA2 and reduce closure can the author co-stain in this setting for actin/lamella/collective migration marker deregulation to help reveal what ANXA2 is doing here? Is ruffling of migratory phenotypes reduced or is cell-cell contact changedetc etc..cell polarity cell shape -?

We thank the reviewer for these very helpful suggestions. We performed E-cadherin immunohistochemistry, however this protein is not expressed on zone 3 hepatocytes in mice (Rebuttal Fig. 3), which precludes its use for analysis of hepatocytes in this region following

APAP-induced liver injury. Instead, we performed immunofluorescence staining for zonula occludens-1 (ZO-1), a tight junction protein mediating cell-cell contact and cell polarity in hepatocytes.

ZO-1 was similar in portal and central-associated hepatocytes in uninjured liver (Extended Data Fig. 7d). ANXA2⁺ hepatocytes in the PNR expressed ZO-1 following APAP-induced acute liver injury (Extended Data Fig. 7e), and ZO-1 expression in the PNR did not change following APAP-induced acute liver injury, demonstrating maintenance of epithelial sheet connections and hepatocyte polarity during wound closure (Extended Data Fig. 7f). ANXA2-positive hepatocytes were less circular than ANXA2-negative hepatocytes in the PNR (Extended Data Fig. 7c). Epithelial sheet connections and hepatocyte polarity were similar between AAV8-shRNA-ScrmB and AAV8-shRNA-*Anxa2* treated mice following APAP-induced acute liver injury (Extended Data Fig. 9e). Hepatocytes in the PNR of AAV8-shRNA-*Anxa2*-treated mice lacked membrane F-actin and were more circular than those in AAV8-shRNA-ScrmB-treated mice, demonstrating a reduced migratory phenotype in ANXA2-negative hepatocytes (Extended Data Fig. 9c,d).

Q in intravital movieswhile collective movement is evident...can the authors score or discuss single cell migration also ? many movies have leader cell movement this should be noted and discussed ...especially in the context of polarity persistence etc..

We thank the reviewer for this helpful comment. The data presented is an analysis of the change in shape and movement of individual hepatocytes over the time course of wound closure (Fig. 3h-j). As the reviewer highlighted, the intravital movies show collective cell migration, and also leader cell movement which we have noted and discussed (lines 343-345, SI Videos 6-13).

We have further added new data analysing the maintenance of hepatocyte polarity and cell-cell contact during APAP-induced liver regeneration (Extended Data Fig. 7d-f, 9e).

In fig 2 b the pattern of contribution for cycling while smaller is similar to ANXA2..did authors assess if this was not spatially regulated at necrotic wound like anax2 ? if so this can be discussed to show diff repair options..

We have analysed the topographical distribution of hepatocyte proliferation in human APAP-ALF and show that hepatocyte proliferation is increased in the PNR (peri-necrotic region) compared to the RVR (remnant viable region) in human APAP-ALF (Fig. 1d). We used the mouse model of APAP-induced liver injury to investigate the temporal dynamics of hepatocyte proliferation versus wound repair (Fig. 3a, Extended Data Fig. 8b), demonstrating an uncoupling of wound closure and hepatocyte proliferation, and uncovering a novel migratory hepatocyte subpopulation which mediates wound closure following liver injury. These data demonstrate that hepatocyte migration (rather than hepatocyte proliferation) is a major mechanism of repair following APAP-induced liver injury.

Referee #4 (Remarks to the Author):

This is a very elegant study from a well-known group of researchers.

1. The exceptional value of this study is the analysis of a (fairly large) number of liver samples subjected to snRNA-seq from human patients with acute liver failure (and controls). The challenge is, though, to understand liver regeneration from samples that are exclusively obtained from individuals with a failure of recovery (otherwise, a transplant would not be needed). What is really lacking is a group of patients that were biopsied during acute liver injury (ideally matching the peak ALT levels of the APAP-ALF / NAE-ALF cohorts), but then recovered completely (and spontaneously, i.e. without specific treatment such as steroids for AIH or antivirals for HBV).

We thank the reviewer for this helpful and thoughtful comment. We have procured FFPE sections (obtained via transjugular liver biopsy) from a rare cohort of patients (n=11) at University College, London. These patients presented with acute, severe liver injury and underwent transjugular liver biopsy and recovered spontaneously without specific treatments or transplantation.

We analysed necrosis, hepatocyte proliferation and hepatocyte ANXA2 expression in transjugular biopsies from these patients. Mean necrotic area was 43.9% ($\pm 5.7\%$ SEM), mean hepatocyte proliferation was 21% ($\pm 4.2\%$ SEM), and ANXA2⁺ hepatocytes were enriched in the PNR (Extended Data Fig. 5d,e), demonstrating that the ANXA2⁺ hepatocyte subpopulation also expands in patients presenting with acute, severe liver injury who recovered spontaneously.

2. The “hepatocyte proliferation” data in Fig.1 are a bit difficult to interpret. The samples are all from explants – and ALF patients have more Ki-67⁺ hepatocytes. But apparently, this proliferative activity does not lead to functional replacement of hepatocytes. Does this mean that Ki-67⁺ cannot be used as a surrogate for proliferation? Or that the proliferation is not high enough?

Ki67 is a widely and extensively used marker to identify proliferating cells. We observed a substantial hepatocyte proliferative response in human APAP-ALF explant livers compared to control human liver, demonstrating that hepatocyte proliferation is robust and relatively unimpeded in human APAP-ALF. However, despite vigorous hepatocyte proliferation observed in the livers of this patient cohort (who have fulfilled King’s criteria for liver transplantation), the necrotic wound area in human APAP-ALF explant livers remained substantial at time of transplantation (Fig 1c).

3. How specific is the existence of ANXA2⁺ hepatocytes to ALF? Would these (ANXA2⁺ migratory) hepatocytes also be present in chronic human liver diseases (e.g. NASH) and/or in liver cirrhosis? – From a clinical point of view, this would be highly relevant, as the loss of such a motile hepatocyte population could explain the irreversibility of certain chronic liver disease conditions (or their reduced regenerative potential).

We thank the reviewer for this helpful comment. In addition to observing the ANXA2⁺ hepatocyte subpopulation in other causes of human ALF (including hepatitis A and B, and other drug-induced liver injuries), we now demonstrate their presence in the following human chronic liver diseases – Metabolic dysfunction-Associated Steatotic Liver Disease (MASLD),

Primary Sclerosing Cholangitis (PSC), Primary Biliary Cholangitis (PBC), and Alcohol-related Liver Disease (ALD) (Extended Data Fig. 5g).

In addition, we have demonstrated the presence of ANXA2⁺ hepatocytes in other mouse models of liver injury including acute and chronic CCl₄ (carbon tetrachloride), and the following biliary fibrosis models: surgical BDL (bile duct ligation) and the DDC (3,5-Diethoxycarbonyl-1,4-Dihydrocollidine) diet model (Extended Data Fig. 5h,i).

Together these data demonstrate that the ANXA2⁺ hepatocyte subpopulation occurs in response to both peri-central vein and peri-portal liver injury in humans and mice.

4. The authors did not comment on a recent paper by Tom Bird's group (May S, et al. J Hepatol. 2023, PMID: 36702176), which could be very relevant for the current study. In this work, the authors find no evidence of predominant expansion of the pericentral hepatocyte population during liver homeostatic regeneration (unlike prior work), and raised some general concerns about peculiarities of mouse models. Moreover, they observed marked differences between male and female mice. To me, this mandates to study liver regeneration in mice of both sexes in order to validate the key observations obtained solely with male mice (as, unfortunately, many studies have done in the past).

We thank the reviewer for this helpful comment. In the human snRNA-seq data, visualising the sex split on a UMAP, and comparing per-sex contribution to the hepatocyte clusters (Extended Data Fig. 4b), we see no sex bias in any of our hepatocyte subpopulations. Generating point-biserial correlations between patient sex and the harmony components of the hepatocyte dataset shows that patient sex did not influence variability in the snRNA-seq data (Extended Data Fig. 4c).

To assess this in mice we investigated the dynamics of liver regeneration following APAP-induced mouse liver injury in female C57/BL6 mice. Female mice injured with APAP also demonstrated the emergence of an ANXA2⁺ hepatocyte subpopulation that was enriched in the peri-necrotic region (Extended Data Fig. 7a,b). Similar to male mice, we found a temporal disconnect between wound closure (as assessed by percentage necrotic area) and hepatocyte proliferation in female mice post APAP-induced acute liver injury (Extended Data Fig. 8b).

To assess in female mice whether hepatocyte repopulation of the necrotic area immediately adjacent to the central vein is driven by proliferation, we gave BrdU in drinking water to label all proliferating hepatocytes following APAP-induced mouse liver injury (Fig. 3b). Glutamine synthetase (GS) is expressed exclusively in hepatocytes adjacent to the central vein, and in female mice following APAP-induced liver injury 86.9% ($\pm 2.3\%$ SEM) of GS-positive hepatocytes were BrdU-negative at 14 days post-injury (Extended Data Fig. 8d). Similar to male mice, these data demonstrate that the majority of hepatocytes in the area immediately adjacent to the central vein do not arise from hepatocyte proliferation.

5. As a cellular source during liver regeneration, cholangiocyte-to-hepatocyte transition has been described / suggested. What is the evidence in favor or against this in ALF? Do some of

the hepatocyte populations (and if so, maybe exclusively in the periportal regions?) repopulation from cholangiocytes / ductular progenitor cells?

We have investigated the potential contributions of other cell lineages to hepatocytes in the mouse APAP liver injury model (using the AAV8-based lineage tracing system) which demonstrated that 99.9% of hepatocytes during APAP-induced acute liver injury derive from hepatocytes (Fig. 2o).

We have also investigated whether we can observe a cholangiocyte-to-hepatocyte transition in human ALF using the snRNAseq dataset. Hepatocytes and cholangiocytes express distinct genes (Extended Data Fig. 1h, SI Table 3), cluster separately, and lack any observable connection in diffusion maps and force-directed graphs (Extended Data Fig. 8a). These data suggest that cholangiocytes are not a major source of hepatocytes in human APAP-ALF and NAE-ALF.

6. What is the role of stellate cells in ‘wound closure’ after ALF? The activation of hepatic stellate cells in ALF has been described quite some time ago in human liver biopsies (Dechêne A, et al. *Hepatology*. 2010; 52(3):1008-16. PMID: 20684020). The authors now propose the uncoupling of wound closure and hepatocyte proliferation, but this may simply represent the task diversification between two populations and this may still be highly orchestrated. From the snRNA-seq data, can the authors comment on HSC-hepatocyte interactions (in a subset-, spatial- and time-resolved manner)?

We thank the reviewer for this helpful comment. We applied interactome analysis using the CellChat R package to investigate potential ligand interactions with receptors expressed by the migratory hepatocyte subpopulation in the human APAP-ALF snRNA-seq data and found that mesenchymal and cholangiocyte subpopulations were the dominant interacting partners (Extended Data Fig. 5a). Multiplex smFISH demonstrated co-location of migratory hepatocytes with myofibroblasts (mesenchyme cluster 1), hepatic stellate cells (mesenchyme cluster 2), and cholangiocytes in the peri-necrotic region (PNR) (Extended Data Fig. 5b). The interactome analysis highlighted multiple interactions related to the transforming growth factor beta (TGF β) signalling pathway (Extended Data Fig. 5c), which has previously been shown to be an important regulator of epithelial cell plasticity and migration (PMID: 33756119).

Similar to our findings in human APAP-ALF, analysis of the mouse APAP-induced acute liver injury datasets using CellChat highlighted myofibroblasts (mesenchyme clusters 1,5,7) as a potential interacting partner with the migratory hepatocyte subpopulation, again via multiple interactions related to the TGF β signalling pathway (Extended Data Fig. 7g,h).

7. The tissue samples originate from various sources / centers. I am sure the authors have ensured quality of the samples and the validity of the transcriptomic data. What is currently missing, but would tremendously add to the translational value, is a correlation between “molecular findings” and the clinical phenotype (etiology, ALT levels, liver function, hyperacute vs. acute vs. subacute ALF, grade of HE etc).

We agree that a correlation between molecular findings and clinical phenotypes would represent an ideal future goal, however to achieve this would require a specifically-designed, prospective study including a much larger patient cohort and full clinical metadata for every patient throughout the study. In contrast, our study used retrospectively collected samples which had limited associated clinical metadata, obtained from collaborations with multiple centres around the world, to accrue an adequate sample size which enabled this exploratory study of human liver regeneration at single cell resolution. Although we were unable to access full clinical metadata for every case, we do show age, sex, and aetiology of acute liver failure (SI Table 1).

Despite the above noted limitations, our work dissected unanticipated aspects of human liver regeneration, and uncovered a novel migratory hepatocyte subpopulation which mediates wound closure following liver injury. We hope these findings will stimulate a new area of therapeutic discovery in regenerative medicine.

8. From a conceptual point of view, I would find it important to demonstrate that the hepatocyte migration as a mechanism of wound closure (mouse experiments in Fig. 3) is not specific to APAP-injury, but would also occur in different types of liver injury, e.g., ischemia-reperfusion-injury. Validating this principal finding in a second (independent) ALF model would be very valuable.

We thank the reviewer for this helpful comment. We have performed immunofluorescence staining for ANXA2⁺ hepatocytes in multiple mouse models of liver injury. We found that ANXA2⁺ hepatocytes with a motile morphology (membrane ruffling, lamellipodia) are present in the following mouse models of liver injury (Extended Data Fig. 5h,i):

1. Acute and chronic carbon tetrachloride-induced liver injury
3. DDC diet-induced biliary fibrosis
4. Surgical bile duct ligation (BDL)-induced biliary fibrosis

Furthermore, the ANXA2⁺ hepatocyte subpopulation was also observed in other causes of human ALF including hepatitis A and B, and other drug-induced liver injuries (Extended Data Fig. 5f). As described above, we have also added new data demonstrating that ANXA2⁺ hepatocytes are present in multiple aetiologies of chronic human liver diseases, including primary sclerosing cholangitis (PSC), primary biliary cholangitis (PBC), metabolic-associated steatotic liver disease (MASLD) and alcohol-induced liver disease (ALD) (Extended Data Fig. 6g). These data demonstrate that the ANXA2⁺ migratory hepatocyte phenotype occurs in response to both peri-portal and peri-central vein liver injury in both humans and mice.

Figure 1. **a**, Migratory signature constructed from the top N differentially expressed genes of the migratory hepatocyte cluster in human hepatocytes. **b**, Migratory signature constructed from the top N differentially expressed genes of the human migratory hepatocyte cluster in mouse hepatocytes.

Figure 2. Migratory hepatocyte probability score of mouse hepatocytes as predicted by a scAnnotatR classifier trained on the human hepatocyte dataset.

Figure 3. Representative immunofluorescence staining in healthy mouse liver showing zonation of E-cadherin. Scale bar 100μm.

Table 1. Confusion matrix results (accuracy, sensitivity and specificity) of the human migratory hepatocyte cluster compared to predicted human migratory hepatocytes based on migration signature score thresholding. A threshold of 0.5 was applied to each top N signature score with all cells above this threshold labelled as a (TRUE) migratory cell in the human hepatocyte dataset.

Top N	5	10	25	50	100
Accuracy	0.8719	0.8877	0.9062	0.9085	0.9124
Sensitivity	0.6167	0.6167	0.5808	0.5835	0.5664
Specificity	0.9076	0.9256	0.9518	0.9541	0.9609

Reviewer Reports on the First Revision:

Referees' comments:

Referee #1 (Remarks to the Author):

The authors discuss similarities with previous work and how their model advanced the field in the response letter. However, this information is missing in the discussion section of the revised manuscript and should be added in shortened form. Other than that the authors addressed my comments and should be congratulated to a great paper.

Referee #2 (Remarks to the Author):

The authors have fully addressed my comments and suggestions. I have no further concern. I would like to take this opportunity to congratulate the authors for this interesting and comprehensive study.

Referee #3 (Remarks to the Author):

The MS is much improved and my question are answered

Referee #4 (Remarks to the Author):

The authors have done a great job, and the additional data strongly support the original and novel findings from this manuscript.

My biggest struggle with the revised version is the extended figure 5, i.e. the claim that these processes are relevant in multiple aetiologies of acute and chronic liver injury in humans (and mice). On the one hand, this important claim should be integrated into the main text / figures (and maybe even the abstract). On the other hand, more statistically sound analyses would be needed to justify this claim. The authors are showing some pictures from different settings, but there is no statistical analysis summarizing the data. How many samples were analyzed? Is the number / pattern of ANXA2+ hepatocytes different in the settings (acute vs chronic, stage of fibrosis, MASLD vs PSC)?

Author Rebuttals to First Revision:

Referees' comments:

Referee #1 (Remarks to the Author):

The authors discuss similarities with previous work and how their model advanced the field in the response letter. However, this information is missing in the discussion section of the revised manuscript and should be added in shortened form. Other than that the authors addressed my comments and should be congratulated to a great paper.

We thank the reviewer for their very positive comments regarding our manuscript, and we have added text and references (including PMID 30762896) in the discussion section (2nd paragraph) discussing previous work in this area and how our model has advanced the field.

Referee #2 (Remarks to the Author):

The authors have fully addressed my comments and suggestions. I have no further concern. I would like to take this opportunity to congratulate the authors for this interesting and comprehensive study.

We thank the reviewer for their very positive comments regarding our manuscript.

Referee #3 (Remarks to the Author):

The MS is much improved and my question are answered

We thank the reviewer for their very positive comments regarding our manuscript.

Referee #4 (Remarks to the Author):

The authors have done a great job, and the additional data strongly support the original and novel findings from this manuscript.

My biggest struggle with the revised version is the extended figure 5, i.e. the claim that these processes are relevant in multiple aetiologies of acute and chronic liver injury in humans (and mice). On the one hand, this important claim should be integrated into the main text / figures (and maybe even the abstract). On the other hand, more statistically sound analyses would be needed to justify this claim. The authors are showing some pictures from different settings, but there is no statistical analysis summarizing the data. How many samples were analyzed? Is the

number / pattern of ANXA2+ hepatocytes different in the settings (acute vs chronic, stage of fibrosis, MASLD vs PSC)?

We thank the reviewer for this thoughtful comment regarding our manuscript.

To comprehensively investigate the functional role of ANXA2+ hepatocytes in various aetiologies of human chronic liver disease, across fibrosis stages, is a separate and important question and represents a very substantial body of work beyond the scope of this manuscript.

We did not intend to claim that ANXA2+ hepatocytes have similar roles and functions in all aetiologies of liver injury, and to clarify this we have amended the text in our manuscript from:

Furthermore, we observed ANXA2⁺ hepatocytes in multiple aetiologies of human chronic liver disease (Extended Data Fig. 5g) and mouse models of liver injury (Extended Data Fig 5h,i), demonstrating that this hepatocyte phenotype occurs in response to both peri-central vein and peri-portal liver injury in humans and mice.

To:

Furthermore, we observed ANXA2⁺ hepatocytes in multiple aetiologies of human chronic liver disease (Extended Data Fig. 5g) and mouse models of liver injury (Extended Data Fig 5h,i), demonstrating that ANXA2⁺ hepatocytes are present in both peri-central vein and peri-portal liver injury in humans and mice.

We have also added sample numbers to the relevant parts of the figure legends.

We once again thank the reviewer for highlighting this important point and as discussed above we have re-worded this section of the text to clarify this.